# Sharp Concentration Bounds for Bundle-Valued Statistics on Manifolds

**Swagatam Das** [1]  **Václav Snášel** [2]

## Abstract

Many geometric statistics and manifold learning pipelines routinely produce observations—such as tangent vectors or local frames—whose natural home is a varying family of fibers attached to different points of a base manifold, rather than a single shared vector space. Forming empirical averages requires transporting these observations to a common reference fiber, introducing curvature- and holonomy-driven effects absent from classical concentration theory. We develop a non-asymptotic concentration theory for such transported empirical means, deriving finite-sample, dimension-free Hoeffding- and Bernstein-type bounds via sharp Hilbert-space inequalities. When shortest paths to the reference point are non-unique, transport becomes path-dependent and introduces a deterministic holonomy bias; we isolate and quantify this bias through bundle curvature and loop geometry, with sharp closed-form formulas for the tangent bundle of a round sphere. The resulting bias–variance decomposition separates the stochastic fluctuation decaying at the classical $n^{-1/2}$ rate in sample size $n$, from a curvature-driven error floor that no amount of additional data can eliminate; minimax lower bounds confirm both terms are unavoidable. We further establish a robust median-of-means estimator achieving optimal rates under heavy tails, and a central limit theorem in the reference fiber. Controlled experiments on the sphere validate all theoretical predictions.

## 1. Introduction

Consider the problem of representing a global wind map. At every point on the Earth's surface, we must attach a small, flat coordinate system to describe the local wind's speed and direction. In the language of geometry, this collection of "data spaces attached to points" is a *vector bundle*. While we can easily draw a flat grid on a local map of a single city, the Earth's intrinsic curvature makes it impossible to "comb" these local coordinate systems into one single, consistent grid for the entire planet without creating a topological "twist" or a seam.

This geometric tension has a precise algebraic counterpart. A vector bundle $(E, \pi, M)$ formalizes exactly this picture: $M$ is the base manifold (the Earth's surface, in our example), $E$ is the total space collecting all local coordinate systems together, and $\pi : E \to M$ is a smooth surjection — a projection map — that tells you which point of $M$ each local coordinate system is attached to. At each point $x \in M$, the preimage $E_x := \pi^{-1}(x) \cong \mathbb{R}^k$ is a vector space fiber, playing the role of the local coordinate system for wind direction and speed at $x$. A wind map is then a *section* of the bundle — a smooth assignment $s : M \to E$ satisfying $\pi(s(x)) = x$, so that $s(x) \in E_x$ at every point, meaning each location on Earth is assigned a wind vector living in its own local coordinate system. The impossibility of a globally consistent grid reflects the fact that $E$ may be globally "twisted," precluding a single coordinate system even though it locally resembles a product space. This is the natural language for geometric ML pipelines encompassing manifolds, graphs, and Lie groups, where one wishes to capture coordinate-invariant features such as directions or local frames (Bronstein et al., 2021). It also mirrors the structure of gauge fields in physics, where curvature governs both field strength and, as we shall see, the quantification of statistical uncertainty.

While concentration inequalities are central to non-asymptotic learning (Boucheron et al., 2013), existing Euclidean results do not extend to geometric pipelines such as manifold regression, optimal transport, or gauge-deep learning (Cohen et al., 2019; Bronstein et al., 2021). These tasks require averaging observations from *different* fibers $E_{X_i}$ by first moving them to a reference fiber $E_{x_0}$ via parallel transport. However, transport is path-dependent; non-unique minimizing geodesics introduce a deterministic, curvature-driven ambiguity—quantified by holonomy (Ambrose & Singer, 1953)—that lacks a Euclidean counterpart and remains unaddressed by standard manifold-mean theory

[1]Electronics and Communication Sciences Unit, Indian Statistical Institute, Kolkata 700108, India [2]VSB Technical University of Ostrava, Czech Republic. Correspondence to: Swagatam Das <swagatam.das@isical.ac.in>.

*Proceedings of the 43rd International Conference on Machine Learning*, Seoul, South Korea. PMLR 306, 2026. Copyright 2026 by the author(s).

(Bhattacharya & Patrangenaru, 2003).

Formally, we study i.i.d. samples $X_1, \ldots, X_n \sim \mu$ on $M$ together with a measurable section $s : M \to E$ that produces bundle-valued observations $s(X_i) \in E_{X_i}$. We analyze the estimator obtained by choosing a reference point $x_0 \in M$, transporting each observation to the reference fiber via parallel transport, and averaging in $E_{x_0}$. The central technical goal is to provide explicit, non-asymptotic high-probability bounds on the deviation of the transported mean from its expectation and to understand how these bounds change when the underlying transport is path-dependent. In particular, we expose a simple structure: sampling variability decays with $n$ as in Euclidean concentration (Pinelis, 1994), while geometric ambiguity is controlled by curvature/holonomy and may not decay with additional data.

**Contributions.** This paper develops a compact, non-asymptotic concentration theory for transported bundle-valued means and separates sampling variability from deterministic transport ambiguity:

- We state explicit geometric conditions ensuring a measurable, canonical reduction of bundle-valued samples to a single reference fiber via minimizing-geodesic parallel transport, and clarify how the transported mean differs fundamentally from the Fréchet mean approach (Bhattacharya & Patrangenaru, 2003).

- Under a uniform bound on $\|s(x)\|_{E_x}$, we apply sharp Hilbert-space inequalities (Pinelis, 1994) to obtain dimension-free Hoeffding and Bernstein concentration bounds for the transported mean (Theorems 1–2).

- We quantify the deterministic holonomy bias with curvature/area bounds, provide sharp holonomy formulas on $TS_r^2$, and prove minimax lower bounds (Theorem 4) showing both the $n^{-1/2}$ rate and the holonomy floor are unavoidable.

- We establish a robust median-of-means estimator achieving $\sigma/\sqrt{n}$ under heavy tails (Corollary 2), a CLT in the reference fiber (Theorem 5), and uncertainty quantification recipes for manifold regression, gauge-equivariant GNNs, and Wasserstein tangent spaces.

**Scope and reading guide.** The main body focuses on the estimator in Eq. (1): transport each bundle-valued observation to a common reference fiber and average it. This primitive appears whenever data-dependent objects live in varying local coordinate systems, for instance, tangent-space residuals in intrinsic regression, fiber features in gauge-equivariant message passing, and vector-field measurements on curved domains. Two points are essential for evaluation: (i) once Assumption 1 reduces the problem to i.i.d. variables in a fixed Hilbert space, the stochastic concentration component follows from sharp classical inequalities; and (ii) the geometric novelty is to explicitly separate this stochastic term from a deterministic transport ambiguity term that depends on curvature and holonomy. Readers interested primarily in the geometric component may jump to Section 5; readers interested in minimax optimality may consult Section 5.1; and readers interested in learning-theory style use can consult the confidence radii in Corollary 1, the robustness and CLT results in Section 5.2, and the practical trade-offs discussed in Section 6. The appendix retains the full technical development, including complete proofs and extended worked calculations.

## 2. Related Work

Our concentration results leverage classical non-asymptotic probability in Banach and Hilbert spaces, specifically the sharp, dimension-free martingale and Bernstein-type inequalities developed by Pinelis (1994) and Boucheron et al. (2013); Ledoux & Talagrand (1991). In the field of manifold statistics, large-sample theory has been extensively developed for Fréchet means and intrinsic/extrinsic CLTs (Bhattacharya & Patrangenaru, 2003; 2005; Afsari, 2011; Hotz et al., 2024). Unlike these works, which focus on data points on the base manifold, we study *sections* of vector bundles. While standard manifold estimators often require non-linear optimization and face convergence challenges near cut loci — the set of points where geodesics, the shortest paths on a manifold, cease to be uniquely length-minimizing — (Hundrieser, 2024), our transported mean approach yields $n^{-1/2}$ rates and explicitly isolates curvature-driven error floors. For heavy-tailed settings, we build upon the median-of-means (MoM) framework (Lugosi & Mendelson, 2019), which was recently extended to non-positively curved spaces by Yun & Park (2023). Finally, our work provides a statistical foundation for geometric deep learning architectures that aggregate features via bundle connections (Bronstein et al., 2021; Ashmore et al., 2022; Puechmorel, 2023), and complements recent advances in Wasserstein-type distances for bundle-valued Gaussian mixtures (Wilson et al., 2024) as well as in parallel transport along optimal transport geodesics for distributional dynamics (Saidi et al., 2026). A detailed comparison with the prior works in tabular form can be found in Section 7.

## 3. Method

Let $(E, \pi, M)$ be a smooth real vector bundle of rank $k$ over a complete $d$-dimensional Riemannian manifold $(M, g)$. Assume $E$ carries a bundle metric $\langle \cdot, \cdot \rangle_{E_x}$ and a compatible metric connection $\nabla$. Fix a reference point $x_0 \in M$ and denote the reference fiber by $E_{x_0}$.

Let $X_1, \ldots, X_n \sim \mu$ be i.i.d. samples on $M$ and let $s : M \to E$ be a measurable section. Given a measurable transport rule selecting, for each $x \in \mathcal{S} := \operatorname{supp}(\mu)$, a curve $\gamma_x$ from $x$ to $x_0$, define the induced parallel transport $P_{x \to x_0} := P_{\gamma_x} : E_x \to E_{x_0}$ and the transported observations

$$Y_i := P_{X_i \to x_0} s(X_i) \in E_{x_0}, \qquad \bar{Y}_n := \frac{1}{n} \sum_{i=1}^{n} Y_i. \quad (1)$$

Let $Y := P_{X \to x_0} s(X)$ for $X \sim \mu$ and define the transported mean $m^\star := \mathbb{E}[Y] \in E_{x_0}$.

**Relation to the Fréchet mean.** The transported mean $m^\star = \mathbb{E}[P_{X \to x_0} s(X)] \in E_{x_0}$ is fundamentally different from evaluating the section at the Fréchet mean $\bar{x} = \arg\min_{x \in M} \mathbb{E}[\operatorname{dist}(X, x)^2]$, i.e., from $s(\bar{x}) \in E_{\bar{x}}$. These two objects target different population quantities: the transported mean aggregates fiber-valued observations after alignment, whereas $s(\bar{x})$ evaluates the section at an estimated base point. In general,

$$m^\star \neq P_{\bar{x} \to x_0} s(\bar{x}),$$

and the difference is controlled by the curvature and variability of $s$. Moreover, estimation of the Fréchet mean $\bar{x}$ can converge slower than $n^{-1/2}$ near cut loci and requires solving a nonlinear optimization on $M$. In contrast, our approach reduces to Hilbert-space averaging in $E_{x_0}$, yielding dimension-free $n^{-1/2}$ rates plus an explicit geometric term, at lower computational cost.

### 3.1. Geometric preliminaries

Geodesics generalize straight lines. For $x \in M$ and $v \in T_x M$, let $\gamma_{x,v}$ be the unique geodesic with $\gamma_{x,v}(0) = x$ and $\dot{\gamma}_{x,v}(0) = v$. The *exponential map* at $x$ is $\exp_x(v) = \gamma_{x,v}(1)$ whenever the geodesic exists on $[0,1]$; where $\exp_x$ is locally a diffeomorphism, its inverse $\log_x$ maps a point $y$ back to the tangent vector whose unit-time geodesic reaches $y$. Figure 1 visualizes the relationship between exp and log.

For $x \in M$, the *injectivity radius* $\operatorname{Inj}(x)$ is the largest $r$ such that $\exp_x$ is a diffeomorphism from the Euclidean ball $B(0, r) \subset T_x M$ onto its image. A *normal ball* $B(x, r)$ with $r < \operatorname{Inj}(x)$ is a geodesic ball on which every point is joined to $x$ by a unique minimizing geodesic that depends smoothly on the endpoint; this is the local regime where transport can be chosen canonically.

Let $E \to M$ be a rank-$k$ vector bundle equipped with a bundle metric and a compatible connection $\nabla$. Parallel transport along a curve $\gamma$ induces an isometry $P_\gamma : E_{\gamma(0)} \to E_{\gamma(1)}$. The curvature of $\nabla$ is an $\operatorname{End}(E)$-valued 2-form $\Omega$, measuring how much the bundle twists over infinitesimal parallelograms on $M$; holonomy around a loop $\Gamma$ based at $x$

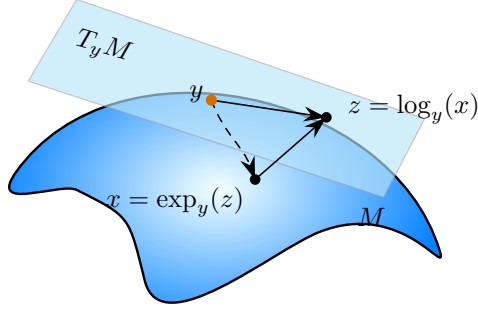

*Figure 1.* Exponential and logarithm maps: $z = \log_y(x) \in T_y M$ and $x = \exp_y(z) \in M$.

is the isometry $P_\Gamma : E_x \to E_x$ encoding the total accumulated twist around $\Gamma$, and the holonomy group is generated by all such loop transports. In our statistical setting, the non-uniqueness of "canonical" paths (e.g., minimizing geodesics) yields deterministic ambiguity in the transported estimator; Appendix C provides full holonomy control via curvature.

Metric compatibility is crucial: because each $P_{x \to x_0}$ is an isometry, moment bounds on $s(X)$ transfer directly to moment bounds on $Y = P_{X \to x_0} s(X)$. In particular, Assumption 2 implies $\|Y\| \leq B$ almost surely, independent of the fiber dimension $k$. Likewise, the Bernstein proxy $\sigma^2 = \|\operatorname{Cov}(Y)\|_{\mathrm{op}}$ uses the operator norm rather than the trace, so the resulting concentration radii do not worsen with $k$ when $Y$ is uniformly bounded in norm.

**Assumption 1** (Geodesic uniqueness / measurable transport). *One of the following holds:*

1. *(**Hadamard case**) $M$ is Cartan–Hadamard (complete, simply connected, and nonpositively curved), so minimizing geodesics are unique.*

2. *(**Normal-ball case**) $\mathcal{S} \subset B(x_0, D)$ for some $D < \operatorname{Inj}(x_0)$, so each $x \in \mathcal{S}$ is joined to $x_0$ by a unique minimizing geodesic depending smoothly on $x$.*

**Assumption 2** (Uniform boundedness). *There exists $B > 0$ such that $\|s(x)\|_{E_x} \leq B$ for all $x \in \mathcal{S}$.*

**Assumption 3** (Variance proxy). *The variance proxy*

$$\sigma^2 := \sup_{\|u\|_{E_{x_0}} = 1} \operatorname{Var}(\langle Y, u \rangle) = \|\operatorname{Cov}(Y)\|_{\mathrm{op}}$$

*is finite.*

**Interpretation of $\sigma^2$.** Once the data are transported to $E_{x_0}$, the variance proxy is a standard Hilbert-space quantity: in an orthonormal basis of $E_{x_0}$, $\operatorname{Cov}(Y)$ is the $k \times k$ covariance matrix of the coordinate representation of $Y$, and

$\sigma^2$ is its largest eigenvalue. This choice is not ad hoc; it is the natural scale parameter in sharp Bernstein inequalities for vector-valued sums. In practice, if $B$ is conservative but $\sigma^2$ is small, Bernstein yields materially tighter radii than Hoeffding. Conversely, when only a hard bound is available, Hoeffding remains applicable without additional estimation.

Assumption 1 guarantees that $x \mapsto P_{x \to x_0}$ (hence $x \mapsto Y(x) := P_{x \to x_0} s(x)$) can be chosen measurably, so $Y_1, \ldots, Y_n$ are i.i.d. in $E_{x_0}$. A complete overview of geometric notation, curvature conventions, and holonomy preliminaries appears in Appendix A.

### 3.2. Canonical transport rules and measurability

Formally, a *transport rule* to $x_0$ is a measurable assignment $x \mapsto \gamma_x$ of a curve from $x$ to $x_0$ (defined at least on $\mathcal{S}$), together with the induced parallel transport $P_{x \to x_0} := P_{\gamma_x}$. Assumption 1 covers two common regimes in which a canonical rule is available:

- In the Hadamard case, minimizing geodesics are unique, so one can take $\gamma_x$ to be the unique minimizing geodesic from $x$ to $x_0$.

- In the normal-ball case, uniqueness holds on $B(x_0, D)$ with $D < \mathrm{Inj}(x_0)$, and $x \mapsto \gamma_x$ varies smoothly with $x$; measurability is immediate.

Outside these regimes, the *cut locus* of $x_0$ can introduce multiple minimizing geodesics, and any choice of $\gamma_x$ becomes a modeling decision that may introduce deterministic ambiguity. Our framework makes this explicit by separating (i) a stochastic deviation term controlled by Banach/Hilbert concentration and (ii) a curvature-driven transport ambiguity term controlled by holonomy (Section 5).

From an algorithmic perspective, when $\gamma_x$ is the unique minimizing geodesic, $P_{x \to x_0}$ can be computed by numerically solving the parallel-transport ODE along $\gamma_x$, or by closed forms in symmetric spaces (e.g., spheres). When the minimizing geodesic is not unique, one can still enforce a deterministic rule (e.g., a lexicographic tie-break among geodesics, or a reference-frame convention). The bounds below remain valid for any such measurable choice, at the price of an explicit holonomy term.

## 4. Main concentration results

Under Assumption 1, the transported variables $Y_i := P_{X_i \to x_0} s(X_i)$ are i.i.d. in the fixed Hilbert space $E_{x_0}$, with $\|Y_i\| \leq B$ almost surely by Assumption 2. This geometric reduction — from bundle-valued observations scattered across different fibers to i.i.d. vectors in a single Hilbert space — is the key step that makes classical concentration machinery applicable. Specifically, it allows us to

apply the sharp martingale inequalities of Pinelis (1994) directly to the transported empirical mean $\bar{Y}_n$, yielding the following dimension-free bounds. The variance proxy $\sigma^2 = \|\mathrm{Cov}(Y)\|_{\mathrm{op}}$ uses the operator norm rather than the trace, which is what keeps the Bernstein bound independent of the fiber dimension $k$: only the largest eigenvalue of the covariance matters, not its full spectrum.

**Theorem 1** (Hoeffding inequality for transported means)**.** *Under Assumptions 1 and 2, for all $\varepsilon > 0$,*

$$\mathbb{P}\big(\|\bar{Y}_n - m^\star\| \geq \varepsilon\big) \ \leq \ 2 \exp\left(-\frac{n\varepsilon^2}{8B^2}\right).$$

This bound depends only on the almost-sure norm bound $B$ and requires no distributional assumptions beyond measurability of the transport rule. When second-moment information is available, the following Bernstein-type bound is strictly tighter in the low-noise regime.

**Theorem 2** (Bernstein inequality for transported means)**.** *Under Assumptions 1–3, for all $\varepsilon > 0$,*

$$\mathbb{P}\big(\|\bar{Y}_n - m^\star\| \geq \varepsilon\big) \ \leq \ 2 \exp\left(-\frac{n\varepsilon^2}{2\big(\sigma^2 + \frac{2B\varepsilon}{3}\big)}\right).$$

*When $\sigma^2 \ll B\varepsilon$, the Bernstein bound is materially tighter than Hoeffding; when only $B$ is available, Hoeffding applies without estimating $\sigma^2$.*

Inverting these tail bounds directly yields explicit confidence sets in the reference fiber $E_{x_0}$.

**Corollary 1** (Confidence radii)**.** *Fix $\delta \in (0, 1)$. Under Theorem 1, with probability at least $1 - \delta$,*

$$\|\bar{Y}_n - m^\star\| \ \leq \ \sqrt{\frac{8B^2 \log(2/\delta)}{n}}.$$

*Under Theorem 2, with probability at least $1 - \delta$,*

$$\|\bar{Y}_n - m^\star\| \ \leq \ \sqrt{\frac{2\sigma^2 \log(2/\delta)}{n}} \ + \ \frac{2B \log(2/\delta)}{3n}.$$

*The $2B\varepsilon/3$ term in the Bernstein radius is conservative (arising from the bound $\|\xi_i\| \leq 2B$); under exact recentering (when $m^\star$ is known), the factor $2B$ may be replaced by $B$, yielding $B \log(2/\delta)/(3n)$.*

Both theorems match standard Euclidean finite-sample rates but apply to transported bundle statistics; the only additional requirement is a well-posed measurable transport reduction, guaranteed by Assumption 1. Complete proofs, a finite-dimensional $\varepsilon$-net derivation, and the sharper constant available under exact recentering are given in Appendix B.

## 4.1. Using the bounds in practice

Corollary 1 gives a radius $r_n(\delta)$ such that $\bar{Y}_n$ lies within $r_n(\delta)$ of $m^\star$ with probability at least $1 - \delta$. Applying it requires (i) a reference point and transport rule, (ii) bounds or estimates of $B$ and $\sigma^2$, and (iii) when transport is non-unique, a holonomy/curvature bound.

**Reference point and transport rule.** $x_0$ may be a fixed landmark (e.g., a node of a graph, an anchor point on a surface) or a data-dependent center, such as the empirical Fréchet mean $\hat{x}_0$ of the base samples. If the support lies in a normal ball, the unique minimizing geodesic is the canonical choice; otherwise, any deterministic measurable convention is valid, at the cost of a nonzero holonomy term.

When $x_0 = \hat{x}_0$ is data-dependent, an additional fluctuation enters: by Lipschitz continuity of $x \mapsto P_{X \to x} s(X)$,

$$\|\bar{Y}_n - m^\star\| \lesssim \frac{B}{\sqrt{n}} + \Delta_{\text{hol}} + L \cdot \text{dist}(\hat{x}_0, x_0),$$

where $L$ is the relevant Lipschitz constant. Since $\text{dist}(\hat{x}_0, x_0) = O_P(n^{-1/2})$ under standard conditions, the dominant rate remains $n^{-1/2}$ in well-behaved regimes; near cut loci, the additional term may dominate, and a fixed $x_0$ should be preferred.

**Computing transported samples.** Evaluate $Y_i = P_{X_i \to x_0} s(X_i)$ by solving the parallel-transport ODE along the selected curve $\gamma_{X_i}$. Closed forms exist on common symmetric spaces (e.g., $S^2$), and numerical transport is standard on meshes via discrete connections (Boissonnat et al., 2018).

**Specifying $B$ and $\sigma^2$.** For intrinsically bounded sections (e.g., unit vectors, clipped residuals, normalized features), $B$ can be set deterministically; otherwise it can be upper-bounded from smoothness or compactness, or handled via robust alternatives (Appendix C). For Bernstein-style bounds, $\sigma^2$ can be estimated from the transported data via the operator norm of the sample covariance of the $Y_i$.

**Transport ambiguity.** If minimizing geodesics are not unique, use the bias–variance decomposition of Theorem 3: stochastic error plus $\Delta(P, \widetilde{P}; s)$. Eq. (3) (defined below in Section 5) provides a curvature-based control in normal balls, while Proposition 1 gives a sharp area-based bound on $S_r^2$.

**Sample-size trade-off.** When $\Delta_{\text{hol}}$ is non-negligible, increasing $n$ reduces only the stochastic radius. To restore $n^{-1/2}$ performance, one should restrict the support to a smaller normal ball, prefer a fixed $x_0$ over a data-dependent one, switch to an intrinsic estimator that avoids global transport, or apply holonomy correction (Appendix C).

## 5. Holonomy ambiguity and bias–variance decomposition

If minimizing geodesics between $x$ and $x_0$ are non-unique, transported statistics depend on the chosen geodesic and thus on the chosen transport rule. Let $P$ and $\widetilde{P}$ be two measurable transport rules to $x_0$. The *canonical* (section-dependent) transport ambiguity is defined as

$$\Delta(P, \widetilde{P}; s) := \sup_{x \in \mathcal{S}} \left\| \left( P_{x \to x_0} - \widetilde{P}_{x \to x_0} \right) s(x) \right\|_{E_{x_0}}. \quad (2)$$

Throughout the main body, $\Delta_{\text{hol}}$ denotes this quantity evaluated at the pair $(P, P^{\text{ref}})$ where $P^{\text{ref}}$ is a fixed canonical transport rule (e.g., the unique minimizing geodesic where available). The appendix uses two further forms that upper-bound $\Delta(P, \widetilde{P}; s)$: a section-uniform version $\Delta_{\text{hol}}^{\text{unif}} := \sup_x \| P_{x \to x_0} - \widetilde{P}_{x \to x_0} \|_{\text{op}} \cdot B$ (Appendix A), and a per-sample operator-norm form used in Theorem 11 (Appendix D). All three upper-bound the bias in the transported mean; the canonical form (2) is the tightest.

**Theorem 3** (Holonomy term in the transported mean). *Let $\bar{Y}_n^{(P)}$ denote the transported mean under rule $P$ and $m^{(P)} := \mathbb{E}[\bar{Y}_n^{(P)}]$. Then, for any $\widetilde{P}$,*

$$\|\bar{Y}_n^{(P)} - m^{(\widetilde{P})}\| \le \|\bar{Y}_n^{(P)} - m^{(P)}\| + \Delta(P, \widetilde{P}; s),$$

*and the stochastic term $\|\bar{Y}_n^{(P)} - m^{(P)}\|$ obeys Theorems 1–2.*

**Curvature control of $\Delta(P, \widetilde{P}; s)$.** Let $\Omega$ be the curvature 2-form of $\nabla$ and assume $\zeta := \sup_{y \in U} \|\Omega_y\|_{\text{op}} < \infty$ on a region $U$ containing $\mathcal{S}$. For two minimizing geodesics $\gamma_1, \gamma_2$ joining $x$ to $x_0$ inside a normal ball of diameter at most $D$, the discrepancy is governed by the holonomy around the loop $\Gamma = \gamma_1 \circ \gamma_2^{-1}$. If additionally $\zeta_1 := \sup_{y \in U} \|\nabla \Omega_y\|_{\text{op}} < \infty$, then the induced transport operators satisfy the explicit estimate

$$\|P_{\gamma_1} - P_{\gamma_2}\|_{\text{op}} \le \zeta D^2 + \tfrac{1}{3} \zeta_1 D^3, \quad (3)$$

and thus $\Delta(P, \widetilde{P}; s) \le (\zeta D^2 + \tfrac{1}{3} \zeta_1 D^3) B$ under Assumption 2. Appendix C contains the geometric proof and discusses how the $D^3$ term can be dropped or refined under stronger structure.

**Bounding curvature norms in practice.** The quantity $\zeta = \sup_{y \in U} \|\Omega_y\|_{\text{op}}$ is the operator norm of the bundle curvature on the region where data live. For the Levi–Civita connection on the tangent bundle, this operator norm is controlled by the Riemannian curvature tensor. It can often be upper-bounded by a sectional-curvature bound on $U$. On a constant-curvature manifold (e.g., the round sphere), $\zeta$ is constant, so Eq. (3) reduces to a simple scaling law in the neighborhood diameter. More generally, if a coarse geometric model or numerical estimate of curvature is available

(for example, via finite-element approximations on a mesh, see (Boissonnat et al., 2018)), it can be inserted directly into Eq. (3) to obtain a conservative but explicit holonomy error bar.

**Proposition 1** (Sphere holonomy (sharp)). *Let $M = S_r^2$ be the round sphere of radius $r$ with its Levi–Civita connection, and let $U \subset B(x_0, \rho)$ with $\rho < \pi r/2$. For any $x \in U$ and two piecewise-smooth paths $\gamma_1, \gamma_2$ in $U$ joining $x$ to $x_0$, let $\Gamma = \gamma_1 \circ \gamma_2^{-1}$ and let $A(\Gamma)$ be the oriented area of a spanning surface. Then*

$$\|P_{\gamma_1} - P_{\gamma_2}\|_{\mathrm{op}} \le 2\sin\big(A(\Gamma)/(2r^2)\big) \le A(\Gamma)/r^2 \le \pi\rho^2/r^2.$$

Proposition 1 yields a concrete deterministic offset scale $\Delta_{\mathrm{hol}} \lesssim (\pi\rho^2/r^2)B$ for tangent-bundle transport on $S_r^2$. Beyond spheres, Eq. (3) provides a conservative but general curvature-based control.

### 5.1. Minimax Lower Bounds

The upper bounds of Theorems 1–3 are tight. The following theorem shows that *both* the $n^{-1/2}$ stochastic term and the $\kappa D^2$ holonomy floor are unavoidable within the class of transport-based estimators, confirming that no alignment-based algorithm can improve upon this bias–variance decomposition.

**Theorem 4** (Minimax lower bounds for transported bundle means). *Let $\mathcal{S}_B$ be the class of measurable sections with $\|s(x)\|_{E_x} \le B$ on $\mathrm{supp}(\mu)$, and let $\hat{m}_n$ range over all transport-based estimators.*

(i) General manifolds. *Let $(M, g)$ be a Riemannian manifold with metric $g$, and assume the sectional curvature satisfies $|K| \le \kappa$ on the geodesic ball $B(x_0, D)$. Assume further either that $E = TM$ with the Levi–Civita connection and $K \ge \kappa_- > 0$ (pinched positive curvature), or that $(M, g)$ has constant sectional curvature $\kappa > 0$ everywhere. Then there exists a universal $c > 0$ such that*

$$\inf_{\hat{m}_n} \sup_{s \in \mathcal{S}_B} \mathbb{E}_s\big[\|\hat{m}_n - m^\star(s)\|\big] \ge c\left(\frac{B}{\sqrt{n}} + B\kappa D^2\right). \quad (4)$$

(ii) Round sphere $S_r^2$. *Let $M = S_r^2$, $E = TS_r^2$, and $\mathrm{supp}(\mu) \subset B(x_0, D)$, $D < \pi r/2$. Then there exists a universal $c > 0$ such that*

$$\inf_{\hat{m}_n} \sup_{s \in \mathcal{S}_B} \mathbb{E}_s\big[\|\hat{m}_n - m^\star(s)\|\big] \ge c\left(\frac{B}{\sqrt{n}} + \frac{BD^2}{r^2}\right). \quad (5)$$

*Proof sketch.* Both bounds follow from Le Cam's two-point method (Le Cam, 1986). For the stochastic term, take $s_0, s_1 \in \mathcal{S}_B$ differing in mean by $\delta = B/\sqrt{n}$; Pinsker's inequality gives $\mathrm{TV}(\mathbb{P}_{s_0}^{(n)}, \mathbb{P}_{s_1}^{(n)}) \le \frac{1}{2}$, yielding the $n^{-1/2}$ contribution. For the geometric term, construct a geodesic

triangle $(x_0, x_1, x_2) \subset B(x_0, D)$ enclosing area $A \asymp D^2$; the Ambrose–Singer theorem (Ambrose & Singer, 1953) gives $\|P_\Gamma - I\|_{\mathrm{op}} \ge c_1\kappa D^2$. Setting $s_0(x) = BP_{x_0 \to x}v$ and $s_1(x) = BP_{x_0 \to x}(P_\Gamma v)$ for a unit $v \in E_{x_0}$ separates the population means by $B\kappa D^2$ while keeping the two laws statistically indistinguishable at small $\kappa D^2$. On $S_r^2$, holonomy is exact ($\kappa = 1/r^2$), giving the sharp $D^2/r^2$ rate in (5). Full proofs are in Appendix C.2–C.3 (Theorems 9–10). □

**Remark 1.** *These lower bounds apply within the class of transport-based estimators (those that align observations to a common fiber before averaging), which is the natural class for geometric ML pipelines. Extrinsic estimators that ignore fiber structure avoid the holonomy floor but sacrifice geometric interpretability. The two terms in (4) are matched to within universal constants by the upper bounds of Corollary 1 and Theorem 3, confirming minimax optimality.*

**Further extensions (Appendix).** Robustness to heavy tails: we adapt the median-of-means framework to achieve sub-Gaussian rates under finite second moment only (Appendix D.4). Full-section estimation: uniform concentration over $L^2(M, \mu; E)$ via Rademacher complexity (Appendix D.3). First-order holonomy correction: a curvature-derived de-biasing step reducing the offset from $O(D^2)$ to $O(D^4)$ (Appendix C.4).

### 5.2. Robustness and Asymptotic Normality

The concentration bounds of Section 4 assume a uniform bound $B$ on the section. We record here the two companion results - heavy-tail robustness and a CLT - that complete the statistical picture. Full proofs are in Appendix D.

**Corollary 2** (Median-of-means robustness). *Suppose only $\mathbb{E}\|Y\|^2 \le \sigma_{L^2}^2 < \infty$ holds (no uniform bound $B$). Partition the $n$ samples into $K = \lceil 2\log(2/\delta) \rceil$ blocks of size $m = \lfloor n/K \rfloor$, form block means $\bar{Y}^{(j)}$ in $E_{x_0}$, and let $\widetilde{Y}_{\mathrm{MoM}}$ be their geometric median. Under Assumption 1, with probability at least $1 - \delta$,*

$$\|\widetilde{Y}_{\mathrm{MoM}} - m^\star\| \le 6\,\sigma_{L^2}\,n^{-1/2},$$

*where $\sigma_{L^2}^2 := \mathbb{E}\|Y\|^2$ is the $L^2$ second moment (distinct from the operator-norm proxy $\sigma_{\mathrm{op}}^2 = \|\mathrm{Cov}(Y)\|_{\mathrm{op}}$ of Assumption 3; one always has $\sigma_{\mathrm{op}}^2 \le \sigma_{L^2}^2$).*

This matches the minimax rate $\sigma/\sqrt{n}$ of Theorem 4(i) without requiring a bounded section, at the cost of a $\log(1/\delta)$ factor in the block count. When $\Delta_{\mathrm{hol}} > 0$, the holonomy bias adds to the MoM error exactly as in Theorem 3; the bias–variance structure is preserved.

**Theorem 5** (CLT in the reference fiber). *Assume Assumption 1 holds so that $Y_1, \ldots, Y_n$ are i.i.d. in $E_{x_0}$ under a fixed canonical transport rule $P^{\mathrm{ref}}$, and let $\mathbb{E}\|Y\|^2 < \infty$. Let $m^\star = \mathbb{E}[Y] \in E_{x_0}$ and $\Sigma = \mathrm{Cov}(Y) \in \mathbb{R}^{k \times k}$. The CLT*

$\sqrt{n}(\bar{Y}_n - m^\star) \xrightarrow{d} \mathcal{N}(0, \Sigma)$ *holds in* $E_{x_0} \cong \mathbb{R}^k$ *for the mean* $m^\star$ *defined by the chosen rule* $P^{\mathrm{ref}}$. *When the population mean is defined without reference to a fixed transport rule (i.e. up to holonomy ambiguity), the additional condition* $\Delta_{\mathrm{hol}} = o(n^{-1/2})$ *(e.g. when* $\mathrm{supp}(\mu) \subset B(x_0, \rho_n)$ *with* $\rho_n^2 n^{1/2} \to 0$*) ensures the holonomy bias is negligible. The CLT holds relative to the common population quantity. Full proof is in Appendix D (Theorem 12); the sphere specialization with shrinking support is Corollary 5.*

Theorem 5 shows that once the holonomy floor is negligible relative to sampling noise, the transported mean is asymptotically Gaussian with the natural fiber covariance $\Sigma$. This complements the finite-sample Bernstein bound: the Bernstein confidence radius shrinks at the same $n^{-1/2}$ rate and the limiting distribution is exactly Gaussian, so both are driven by $\Sigma$. Together, Theorems 1–4 and Corollary 2, and Theorem 5 give a complete non-asymptotic and asymptotic theory for transported bundle-valued means.

# 6. Numerical Validation and Applications

While the preceding developments establish a rigorous analytical framework, this section provides empirical grounding for the derived concentration bounds through numerical instantiation and representative downstream applications. We use the sphere's tangent bundle to quantify the interplay between stochastic sampling error and deterministic holonomy bias. Full numerical experiments are in Appendix H; detailed implementation notes and additional results are also provided in Appendix C.

## 6.1. Manifold regression residual means

Fig. 1). The standard pipeline transports each $r_i$ to a reference tangent space $T_{x_0} M$ and averages, producing a bundle-valued mean $\bar{Y}_n$. Classical analyses treat this pipeline via local Euclidean approximations and predict $O(n^{-1/2})$ error. Our results give the explicit decomposition

$$\|\bar{Y}_n - m^\star\| \lesssim \underbrace{n^{-1/2}}_{\text{sampling}} + \underbrace{\Delta_{\mathrm{hol}}}_{\text{geometric bias}},$$

revealing a curvature-induced term absent from classical analyses.

**Bound constants.** If the regression map and responses remain inside a normal ball of radius $\rho$, then $\|\log_{X_i} \hat{f}(X_i)\| \le \rho$ and one can take $B \le \rho$. The Bernstein variance proxy $\sigma^2$ captures how dispersed the transported residuals are in $T_{x_0} M$; empirically, it can be approximated by the operator norm of the sample covariance of $Y_i$. Corollary 1 therefore yields a non-asymptotic confidence ball in $T_{x_0} M$ for the mean transported residual, which can be mapped back to a manifold-level error bar via the exponential map.

**Holonomy bias and error floor.** When transport non-uniqueness is present, Theorem 3 adds the deterministic offset $\Delta_{\mathrm{hol}}$. On a manifold with sectional curvature $\kappa$ and data supported in $B(x_0, \rho)$,

$$\Delta_{\mathrm{hol}} \lesssim \kappa \rho^2 B, \qquad \text{so} \qquad \|\bar{Y}_n - m^\star\| \lesssim \frac{B}{\sqrt{n}} + \kappa \rho^2 B.$$

Once $n \gtrsim (\kappa \rho^2)^{-2}$, error becomes curvature-dominated: increasing sample size alone cannot reduce it below $\kappa \rho^2 B$. This predicts an *observable error floor* in tangent-space averaging pipelines when data are spread across regions of non-negligible curvature. On $S_r^2$ with data in $B(x_0, \rho)$, Proposition 1 gives the sharp bound $\Delta_{\mathrm{hol}} \le (\pi \rho^2 / r^2) B$, and the stochastic and geometric terms are equated at the crossover sample size

$$n^\times(\delta) \approx \frac{8 \log(2/\delta)}{\pi^2} \left(\frac{r}{\rho}\right)^4. \tag{6}$$

For large caps (large $\rho/r$) holonomy dominates at moderate $n$; in the local regime $\rho \ll r$, the holonomy term is negligible until very large $n$. To restore $n^{-1/2}$ performance, one may restrict to a smaller normal ball, use sample splitting to ensure fixed $\hat{f}$ (making residuals i.i.d.), or apply the first-order holonomy correction of Appendix C.4.

**Independence of residuals.** When $\hat{f}$ is fixed (e.g., estimated on a held-out split), each residual $r_i = \log_{X_i} \hat{f}(X_i)$ is a deterministic function of $X_i \sim \mu$, so the transported variables $Y_i = P_{X_i \to x_0} r_i$ are i.i.d. and Theorems 1–2 apply directly.

## 6.2. Further applications

The bias–variance decomposition of Theorem 3 applies directly to three further pipelines.

**Gauge-equivariant GNNs.** Node features in local frames are aggregated via edge-wise parallel transport maps (Bronstein et al., 2021; Cohen et al., 2019). The geometric term becomes discrete holonomy around cycles, governing gauge inconsistency; small stochastic error alongside transport-dependent instability signals holonomy as the source, not lack of data. See Appendix C, §C.5.1 for the tail bound and $\Delta_{\mathrm{hol}}$ control.

**Diffusion-tensor imaging.** Averaging diffusion directions or SPD tensors over anatomical neighborhoods is a canonical bundle-valued problem (Fletcher & Joshi, 2004; Pennec et al., 2006). Corollary 1 yields a confidence radius for the transported average, while Eq. (3) bounds the holonomy offset when curvature-driven misalignment creates a bias irreducible by additional samples. The full pipeline is in Appendix F.

**Wasserstein tangent spaces.** Under Caffarelli regularity (Caffarelli, 1992) (assuming a compact convex domain $\mathcal{X}$ and densities bounded above and below), the tangent space at a reference measure $\mu_0$ is the Hilbert space $L^2(\mathcal{X}, \mu_0; \mathbb{R}^d)$ (Ambrosio et al., 2008). In this regular regime, averaging tangent representations of observations at different base measures fits our framework *analogously*: the bias–variance decomposition separates the $n^{-1/2}$ stochastic decay from the convention-dependent transport ambiguity. We note that the Wasserstein space is infinite-dimensional and does not carry a finite-rank bundle connection in the sense of Section 3; the analogy therefore holds at the level of the Hilbert-space averaging structure, not as a literal special case. For a discussion of the precise scope and requirements, see the extended results in Appendix C.

### 6.3. Implementation notes: computing transports

In continuous manifolds, parallel transport along a curve is computed by solving a linear ODE defined by the connection. For Levi–Civita transport on embedded manifolds, standard numerical schemes integrate the corresponding Christoffel-symbol system. In triangle meshes and graphs, discrete connections approximate transport via orthogonal transformations between adjacent tangent frames or learned equivariant alignment matrices. These choices fit our framework, provided the rule is deterministic and measurable. Evaluating a single parallel transport requires $O(d^3)$ operations in $d$-dimensional ambient coordinates; these computational considerations are orthogonal to the concentration theory developed above.

### 6.4. Sphere example

On $S_r^2$, the sharp holonomy formula of Proposition 1 gives an explicit and fully computable criterion for whether transport ambiguity is negligible relative to sampling error. For data supported in a geodesic ball $B(x_0, \rho)$ with $\rho < \pi r/2$, Proposition 1 gives $\Delta_{\text{hol}} \leq (\pi\rho^2/r^2)B$, and Corollary 1 gives stochastic radii scaling as $B/\sqrt{n}$ (Hoeffding) or $\sqrt{\sigma^2/n}$ (Bernstein). The two contributions to the total error are therefore

$$\|\bar{Y}_n - m^\star\| \lesssim \underbrace{\sqrt{\frac{8B^2 \log(2/\delta)}{n}}}_{\text{stochastic}} + \underbrace{\frac{\pi\rho^2}{r^2}B}_{\text{geometric bias}} .$$

**Crossover sample size.** As already derived in Section 6.1 (manifold regression), equating the stochastic and geometric terms on $S_r^2$ yields the crossover sample size $n_\times(\delta)$ given in Eq. (6), beyond which holonomy bias dominates and increasing sample size alone cannot reduce the total error further.

**Practical regimes.** Equation (6) makes two qualitative predictions explicit:

- *Large cap* ($\rho$ comparable to $r$): $n_\times(\delta)$ is of moderate size, so holonomy effects dominate already at feasible sample sizes. For example, with $\rho = r/2$ and $\delta = 0.05$, $n_\times \approx 8\ln(40)/\pi^2 \cdot 16 \approx 48$.

- *Local regime* ($\rho \ll r$): $n_\times(\delta) \propto (r/\rho)^4 \to \infty$, so the holonomy term is negligible until extremely large $n$, and classical $n^{-1/2}$ analyses remain accurate.

This explains why curvature-driven transport ambiguity is often invisible in small-sample experiments but becomes a limiting factor in high-data regimes. To restore $n^{-1/2}$ performance beyond $n_\times(\delta)$, one may restrict the support to a smaller geodesic ball (reducing $\rho$), or apply the first-order holonomy correction described in Appendix C.4, which reduces the deterministic offset from $O(\rho^2/r^2)$ to $O(\rho^4/r^4)$ and correspondingly increases $n_\times(\delta)$ by a factor of $(r/\rho)^4$.

## 7. Comparison with Related Works

Table 1 summarizes how our framework relates to established directions in probability and geometry. The nonasymptotic concentration theories of Ledoux & Talagrand (1991) and Boucheron et al. (2013) achieve finite-sample guarantees but operate exclusively in linear spaces, with no mechanism for curvature or holonomy; the manifold-based Fréchet mean results of Bhattacharya & Patrangenaru (2003; 2005) and Fletcher et al. (2004) handle curved geometry but deliver only asymptotic normality with no exponential tail bounds. Our framework occupies the intersection of both desiderata: it operates on vector bundles over Riemannian manifolds and delivers finite-sample Hoeffding and Bernstein bounds, with the holonomy bias $\Delta_{\text{hol}}$ isolated as a quantitatively sharp term via Eq. (3), while minimax lower bounds (Theorem 4) confirm that both the $n^{-1/2}$ stochastic term and the holonomy floor are unavoidable within the class of transport-based estimators. For heavy-tailed sections, Corollary 2 extends the median-of-means framework of Lugosi & Mendelson (2019) from linear Banach spaces to bundle-valued data, retaining the minimax rate $\sigma/\sqrt{n}$, and Theorem 5 establishes asymptotic Gaussianity in the reference fiber once the holonomy floor is negligible. Taken together, these results unify Euclidean, manifold, and bundle-based inference under a single probabilistic structure with explicit finite-sample guarantees throughout.

## 8. Conclusion

We developed non-asymptotic concentration bounds for empirical means of bundle-valued data on Riemannian manifolds. By transporting samples to a common fiber, our

*Table 1.* **Comparison with major frameworks in probabilistic mean estimation and concentration theory.** ✓ = finite-sample bounds available; × = asymptotic only.

| Framework | Setting | Assumptions | Main contributions | Finite-sample? | Limitations |
|---|---|---|---|---|---|
| Ledoux & Talagrand (1991) | Banach/Hilbert-valued RVs | Sub-Gaussian or bounded $\psi_2$ tails | Dimension-free concentration: $\Pr(\|\bar{Y}_n - \mathbb{E}Y\| > t) \leq e^{-cnt^2/\sigma^2}$ | ✓ | Linear spaces only; no curvature or transport. |
| Boucheron et al. (2013) | Euclidean random vectors | Bounded or subexponential tails | Sharp nonasymptotic Hoeffding/Bernstein inequalities | ✓ | No geometric structure; cannot model holonomy. |
| Bhattacharya & Patrangenaru (2003; 2005) | Intrinsic Fréchet means | NPC manifolds; convexity of geodesic balls | Asymptotic CLT on tangent spaces | × | Asymptotic only; no exponential bounds. |
| Fletcher et al. (2004) | Shape/manifold statistics | Bounded-curvature manifold regions | CLT for geodesic and extrinsic means | × | No nonasymptotic concentration bounds. |
| Lugosi & Mendelson (2019) | Banach spaces; heavy tails | Finite second moment | Robust median-of-means; optimal $\sigma/\sqrt{n}$ rate | ✓ | Linear setting; no curvature or bundle effects. |
| **This work** | Vector bundles over Riemannian manifolds | $\|s(X)\| \leq B$; finite $\sigma^2$; curvature bound on $\mathrm{supp}(\mu)$ | (i) Dimension-free Hoeffding & Bernstein bounds in $E_{x_0}$; (ii) holonomy bias $\Delta_{\mathrm{hol}}$ with curvature/area control (3); (iii) minimax lower bounds (Thm. 4; full proofs in Appendix C.2–C.3); (iv) robust MoM for heavy-tailed sections (Cor. 2); (v) CLT in $E_{x_0}$ (Thm. 5) | ✓ | Requires measurable transport; holonomy floor irreducible by data alone when $\Delta_{\mathrm{hol}} > 0$. |

results extend sharp Hilbert-space Hoeffding and Bernstein inequalities to geometric settings, yielding a clean bias–variance decomposition

$$\|\bar{Y}_n - m^\star\| \lesssim \frac{B}{\sqrt{n}} + \Delta_{\mathrm{hol}},$$

that pinpoints when geometry intrinsically limits statistical accuracy. Minimax lower bounds (Theorem 4) confirm both terms are unavoidable within the class of transport-based estimators. Under heavy tails, Corollary 2 retains the $\sigma/\sqrt{n}$ rate, and Theorem 5 establishes Gaussianity in the reference fiber once the holonomy floor is negligible. Controlled experiments on $S^2$ confirm both the $n^{-1/2}$ stochastic decay and the holonomy-induced error floor, matching Proposition 1 to within $3.7\%$ across all tested configurations (Appendix H, Table 5).

**Limitations and outlook.** The framework is intentionally modular: the statistical component requires only i.i.d. transported samples in a fixed Hilbert space, while the geometric component requires a measurable transport rule and curvature control on the data-support region. When global transport is ill-posed, localization to normal balls or intrinsic estimators can avoid holonomy bias. In systems that *learn* a connection — such as equivariant alignment modules — our results identify which geometric quanti-

ties, particularly discrete holonomy around cycles, must be controlled for stable aggregation. Approximate or learned transports enter the bound additively as a third term, preserving the bias–variance structure while capturing deviations from metric-compatibility.

## Impact Statement

This paper advances the theoretical foundations of machine learning by developing non-asymptotic statistical guarantees for geometric and bundle-valued data. The results provide explicit concentration bounds for learning pipelines on manifolds and vector bundles, with potential relevance to applications such as gauge-equivariant graph neural networks, manifold regression, and diffusion tensor imaging. As a theoretical contribution aimed at improving the reliability and understanding of learning in non-Euclidean settings, we do not anticipate direct negative societal impacts arising from this work.

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

**Appendix Roadmap.** The appendix is organized as follows. Section A establishes geometric notation, reviews Riemannian and vector-bundle prerequisites, and collects the probabilistic primitives used throughout. Section B contains the rigorous concentration theory: formal restatements of Theorems 1–3 of the main text with complete proofs (as Theorems 6–8 below), the elementary $\varepsilon$-net derivation (Proposition 2), and the holonomy discrepancy lemmas (Lemmas 3 and 4). Section C develops extended results: exact holonomy control on round spheres (Proposition 3, the appendix proof of Proposition 1), minimax lower bounds confirming the inevitability of both the stochastic and holonomy terms (Theorems 9 and 10, which are the full proofs of the combined Theorem 4 stated in the main body), a first-order holonomy correction reducing the bias from $O(D^2)$ to $O(D^4)$, and worked applications to gauge-equivariant GNNs, manifold regression, and diffusion-tensor imaging. Section D provides refined probabilistic guarantees: a dimension-explicit transported-mean bound (Theorem 11), a central limit theorem in the reference fiber (Theorem 12), $L^2$-section concentration (Theorem 13), and the robust median-of-means estimator (Theorem 14). Auxiliary lemmas supporting the main proofs are collected at the end of each section. Representative application examples—tangent-bundle residuals, Grassmann subspace tracking, hyperbolic embeddings, and full-section estimation—are worked out in Appendix F, Appendix H provides numerical verification of the holonomy error floor on $S^2$, and Appendix I develops an intrinsic formulation of bundle-valued concentration without a fixed reference fiber.

# A. Preliminaries

This section establishes notation, reviews standard geometric facts, and outlines the probabilistic primitives we employ. We use the notation from Berger's book (Berger, 2003). The Hadamard alternative in Assumption 1(1) asks that $(M, g)$ be a Cartan–Hadamard manifold. In particular, Cartan–Hadamard manifolds are CAT(0) spaces in the sense of Alexandrov, so distance functions and squared distance to geodesics are geodesically convex and metric projections onto closed, convex subsets are well-defined and 1-Lipschitz; see, e.g., (Gallot et al., 2004; Lee, 1997; Bridson & Häfliger, 1999; Ballmann et al., 2006). Far from being a restrictive assumption, the Cartan–Hadamard class is extremely rich and covers many model spaces encountered in geometric analysis and manifold-based learning. Besides Euclidean space and real hyperbolic space, it includes all simply connected, complete manifolds with non-positive sectional curvature, in particular symmetric spaces of non-compact type such as $\mathrm{SL}(n, \mathbb{R})/\mathrm{SO}(n)$ and products of such spaces, as well as many pinched negatively curved manifolds (Bridson & Häfliger, 1999; Ballmann et al., 2006). Moreover, every complete Riemannian manifold with non-positive curvature has a Cartan–Hadamard manifold as its universal covering space (Bridson & Häfliger, 1999), so our assumptions can be viewed as working on the natural universal cover of a broad class of non-positively curved models. From an applied point of view, Cartan–Hadamard manifolds already underpin a substantial body of work in statistics and machine learning, and thus form a natural setting for our bundle-valued concentration theory.

## A.1. Notation of Riemannian manifolds and geodesics and vector-bundle basics

We start with a brief reminder of Riemannian geometry. More complete presentations can be found in (Lee, 1997; do Carmo, 1992; Gallot et al., 2004). A *Riemannian manifold* $(M, g)$ of dimension $d$ is a smooth manifold $M$ together with a smoothly varying inner product $g_x : T_xM \times T_xM \to \mathbb{R}$, $\quad x \in M$, on each tangent space $T_xM$. For $u, v \in T_xM$ we write $\langle u, v \rangle_x := g_x(u, v)$ and $\|u\|_x^2 := \langle u, u \rangle_x$. The metric allows one to define the length of a piecewise smooth curve $\gamma : [0, 1] \to M$: $L(\gamma) = \int_0^1 \|\dot\gamma(t)\|_{\gamma(t)} \, \mathrm{d}t$. The *Riemannian distance* $\mathrm{dist}(x, y)$ is the infimum of $L(\gamma)$ over all curves $\gamma$ connecting $x$ to $y$.

A smooth curve $\gamma : I \to M$ is a *geodesic* if it is locally a critical point of the length (or equivalently the energy) functional. Equivalently, $\gamma$ satisfies the geodesic equation $\nabla_{\dot\gamma}\dot\gamma = 0$, where $\nabla$ is the Levi-Civita connection of $g$.

Geodesics generalize straight lines. In a complete Riemannian manifold, the Hopf–Rinow theorem states that a length-minimizing geodesic can join any two points geodesic, and geodesics can be extended indefinitely.

For $x \in M$ and $v \in T_xM$, let $\gamma_{x,v}$ be the unique geodesic with $\gamma_{x,v}(0) = x$ and $\dot\gamma_{x,v}(0) = v$. The *exponential map* at $x$ is $\exp_x : T_xM \to M$, $\exp_x(v) = \gamma_{x,v}(1)$, defined wherever $\gamma_{x,v}$ exists on $[0, 1]$. In a geodesically complete manifold, this holds for all $v$. When $\exp_x$ is a diffeomorphism, its inverse is the *logarithm map* $\log_x : M \to T_xM$ with $\log_x(y)$ the initial velocity of the geodesic from $x$ to $y$.

For $x \in M$, the *injectivity radius* $\mathrm{Inj}(x) \in (0, \infty]$ is the supremum of $r > 0$ such that the exponential map $\exp_x$ restricts to a diffeomorphism from the Euclidean ball $B(0, r) \subset T_xM$ onto its image. A *normal ball* $B(x, r)$ with $r < \mathrm{Inj}(x)$ is a geodesic ball on which every point is joined to $x$ by a unique minimizing geodesic depending smoothly on the endpoint.

Let $(E, \pi, M)$ be a smooth real vector bundle of rank $k$ over a smooth manifold $M$. A *bundle metric* is a smooth assignment

*Table 2.* Geometric notation, Bundle sections, and functional spaces

| Symbol | Type | Meaning |
|---|---|---|
| $(M, g)$ | pair | Riemannian manifold with metric $g$. |
| $T_x M$ | vector space | Tangent space to $M$ at $x$. |
| $\langle u, v \rangle_x$ | scalar | Inner product on $T_x M$ induced by $g$. |
| $\|u\|_x$ | scalar | Norm $\|u\|_x := \sqrt{\langle u, u \rangle_x}$ on $T_x M$. |
| $L(\gamma)$ | scalar | Length of a (piecewise) smooth curve $\gamma$. |
| $\mathrm{dist}(x, y)$ | scalar | Riemannian distance between $x, y \in M$. |
| $\gamma$ | curve | Geodesic or piecewise-smooth curve in $M$. |
| $\nabla$ | connection | Levi–Civita connection on $(M, g)$ (also used for bundle connection). |
| $\mathrm{Exp}_x$ | map | Exponential map $\exp_x : T_x M \to M$. |
| $\mathrm{Log}_x$ | map | Logarithm map (inverse of $\exp_x$ where defined). |
| $K$ | scalar | Sectional curvature, often bounded by $|K| \leq \kappa$ on a region. |
| $\kappa$ | scalar | Uniform curvature bound. |
| $B(x_0, r)$ | subset | Geodesic ball of radius $r$ around $x_0$. |
| $\mathrm{Inj}(x_0)$ | scalar | Injectivity radius at $x_0$. |
| $E$ | bundle | Smooth real vector bundle $(E, \pi, M)$ of rank $k$ over $M$. |
| $\pi : E \to M$ | map | Bundle projection. |
| $E_x$ | vector space | Fiber over $x$, $E_x = \pi^{-1}(x)$. |
| $\langle \cdot, \cdot \rangle_{E_x}$ | scalar | Inner product on the fiber $E_x$ (bundle metric). |
| $\| \cdot \|_{E_x}$ | scalar | Norm induced by $\langle \cdot, \cdot \rangle_{E_x}$. |
| $\nabla$ (on $E$) | connection | Metric connection on $E$ compatible with the bundle metric. |
| $P_\gamma$ | operator | Parallel transport along curve $\gamma$, $P_\gamma : E_{\gamma(0)} \to E_{\gamma(1)}$. |
| $P_{x \to x_0}$ | operator | Parallel transport along the chosen geodesic from $x$ to $x_0$. |
| $\Omega$ | 2-form | Curvature 2-form of the bundle connection. |
| $\Gamma$ | loop | Closed curve obtained by concatenating two paths. |
| $A(\Gamma)$ | scalar | Oriented area of a surface spanning the loop $\Gamma$. |
| $\Delta_{\mathrm{hol}}$ | scalar | Holonomy-induced bias term in the transported statistics. |
| $s : M \to E$ | section | Measurable (often smooth) section, $s(x) \in E_x$. |
| $\|s(x)\|_{E_x}$ | scalar | Pointwise fiber norm of the section at $x$. |
| $B$ | scalar | Uniform bound for $s$: $\|s(x)\|_{E_x} \leq B$ on $\mathrm{supp}(\mu)$. |
| $L^2(M, \mu; E)$ | Hilbert space | Space of square-integrable sections w.r.t. measure $\mu$. |
| $\|s\|_{L^2(M,\mu;E)}$ | scalar | Norm $\left( \int_M \|s(x)\|_{E_x}^2 \, \mathrm{d}\mu(x) \right)^{1/2}$. |

*Table 3.* Random variables and empirical statistics

| Symbol | Type | Meaning |
|--------|------|---------|
| $\mu$ | measure | Probability law of $X$ on $M$. |
| $X, X_i$ | random variable | I.i.d. random points in $M$ with law $\mu$. |
| $Y$ | random variable | Transported random element $Y = P_{X \to x_0} s(X) \in E_{x_0}$. |
| $Y_i$ | random variable | I.i.d. copies $Y_i = P_{X_i \to x_0} s(X_i) \in E_{x_0}$. |
| $\overline{Y}_n$ | random variable | Transported empirical mean $\overline{Y}_n = \frac{1}{n} \sum_{i=1}^n Y_i$. |
| $m^\star$ | vector | Population transported mean $m^\star = \mathbb{E}[P_{X \to x_0} s(X)] \in E_{x_0}$ (also written $m$ when context is clear). |
| $\xi_i$ | random variable | Centered variables $\xi_i = Y_i - \mathbb{E}Y$. |
| $\sigma^2$ | scalar | Variance proxy $\sigma^2 = \sup_{u \in S(E_{x_0})} \mathrm{Var}\langle Y, u \rangle$. |
| $S(E_{x_0})$ | set | Unit sphere in the fiber $E_{x_0}$. |
| $\mathbb{E}$ | operator | Expectation w.r.t. the law of the data (and auxiliary randomness if present). |
| $\mathrm{Var}(\cdot)$ | scalar | Variance operator. |
| $\mathbb{P}(\cdot)$ | measure | Probability of an event. |

$x \longmapsto \langle \cdot, \cdot \rangle_{E_x}$ of inner products on each fiber $E_x := \pi^{-1}(x)$. A connection $\nabla$ on $E$ is *metric compatible* if for all smooth sections $u, v \in \Gamma(E)$ and all smooth vector fields $X \in \mathfrak{X}(M)$, $X(\langle u, v \rangle_E) = \langle \nabla_X u, v \rangle_E + \langle u, \nabla_X v \rangle_E$, where $\langle u, v \rangle_E$ denotes the smooth function $x \mapsto \langle u(x), v(x) \rangle_{E_x}$. Let $\gamma : [0, 1] \to M$ be piecewise $C^1$. For $v \in E_{\gamma(0)}$, the *parallel transport* $P_\gamma v \in E_{\gamma(1)}$ is defined by solving the parallel-transport equation $\nabla_{\dot{\gamma}(t)} V(t) = 0$, $V(0) = v$, and setting $P_\gamma v := V(1)$. If $\nabla$ is metric compatible, then $P_\gamma : E_{\gamma(0)} \to E_{\gamma(1)}$ is an isometry for the bundle metric.

The curvature of $\nabla$ is the $\mathrm{End}(E)$-valued 2-form $\Omega$ defined by

$$\Omega(X, Y) := \nabla_X \nabla_Y - \nabla_Y \nabla_X - \nabla_{[X, Y]},$$

acting on smooth sections of $E$. Its pointwise operator norm is

$$\|\Omega_x\|_{\mathrm{op}} := \sup_{\substack{u, v \in T_x M \\ \|u\| = \|v\| = 1}} \|\Omega_x(u, v)\|_{\mathrm{op}},$$

where $\mathrm{End}(E)$ is the *endomorphism bundle* of a vector bundle $E \to M$, $\Omega_x(u, v) \in \mathrm{End}(E_x)$ and $\| \cdot \|_{\mathrm{op}}$ is the operator norm induced by $\langle \cdot, \cdot \rangle_{E_x}$.

(If $E = TM$ with the Levi–Civita connection, then $\Omega$ corresponds to the Riemann curvature tensor via $\Omega(u, v)w = R(u, v)w$.)

Fix $x \in M$. The *holonomy group* $\mathrm{Hol}_x(\nabla) \subset O(E_x)$ is the subgroup generated by $P_\Gamma$ over all piecewise smooth loops $\Gamma$ based at $x$. For a specific loop $\Gamma$, the *holonomy* is the isometry

$$P_\Gamma : E_x \to E_x.$$

Fix a reference point $x_0 \in M$ and a set $S \subset M$. A *transport rule* to $x_0$ is a measurable map that assigns to each $x \in S$ a curve $\gamma_x$ from $x$ to $x_0$, inducing a linear map

$$P_{x \to x_0} := P_{\gamma_x} : E_x \to E_{x_0}.$$

Given two transport rules $P$ and $\widetilde{P}$ and a section $s : M \to E$, define the (section-dependent) *canonical transport ambiguity*

$$\Delta(P, \widetilde{P}; s) := \sup_{x \in S} \left\| \left( P_{x \to x_0} - \widetilde{P}_{x \to x_0} \right) s(x) \right\|_{E_{x_0}}.$$

Throughout the main body, $\Delta_{\mathrm{hol}}$ denotes $\Delta(P, P^{\mathrm{ref}}; s)$ for a fixed canonical rule $P^{\mathrm{ref}}$ (the tightest form, matching Eq. (2)). A section-uniform upper bound (used in some appendix proofs) is

$$\Delta_{\mathrm{hol}}^{\mathrm{unif}} := \left( \sup_{x \in S} \left\| P_{x \to x_0} - \widetilde{P}_{x \to x_0} \right\|_{\mathrm{op}} \right) \left( \sup_{x \in S} \| s(x) \|_{E_x} \right) \geq \Delta(P, \widetilde{P}; s).$$

The per-sample operator-norm form $\Delta_{\mathrm{hol}} := \sup_x \|P_{\gamma_x} - \widetilde{P}_{\gamma_x}\|_{\mathrm{op}} \cdot B$ used in Theorem 11 is the same as $\Delta_{\mathrm{hol}}^{\mathrm{unif}}$. All three upper-bound the bias in the transported mean; the canonical form is the tightest.

Let $Y \in E_{x_0}$ be square-integrable. Define

$$v^2 := \|\mathrm{Cov}(Y)\|_{\mathrm{op}} = \sup_{\|u\|_{E_{x_0}}=1} \mathrm{Var}\big(\langle Y, u \rangle_{E_{x_0}}\big).$$

This is the appropriate variance term for Hilbert-valued Bernstein inequalities.

Given $z_1, \ldots, z_K$ in a Hilbert space $(H, \|\cdot\|)$, a *geometric median* is any minimizer of

$$y \longmapsto \sum_{j=1}^{K} \|z_j - y\|.$$

Existence holds in finite dimensions (and more generally in reflexive Banach spaces under mild conditions).

## A.2. Sectional curvature and non-positive curvature

The *sectional curvature* measures the curvature of $M$ along two-dimensional directions. Let $x \in M$ and let $\sigma \subset T_x M$ be a 2-dimensional subspace, spanned by linearly independent $u, v \in T_x M$. The *sectional curvature* of $\sigma$ at $x$ is

$$K_x(u, v) = \frac{\langle R(u, v)u, v \rangle_x}{\|u\|_x^2 \|v\|_x^2 - \langle u, v \rangle_x^2},$$

where $R$ is the Riemann curvature tensor. We say that $(M, g)$ has *non-positive curvature* if $K_x(u, v) \leq 0$ for all $x \in M$ and all 2-planes $\sigma \subset T_x M$.

Besides this differential definition, curvature can be characterized by the behavior of geodesic triangles: in non-positively curved manifolds, geodesic triangles are "thinner" than their Euclidean comparison triangles, which is formalised in the CAT(0) condition (Bridson & Häfliger, 1999).

A *Cartan–Hadamard manifold* is a complete, simply connected Riemannian manifold $(M, g)$ with non-positive sectional curvature everywhere. Equivalently, $\exp_{x_0} : T_{x_0} M \to M$ is a global diffeomorphism for each $x_0 \in M$, and a unique minimizing geodesic joins any two points. This is called the Cartan–Hadamard theorem (Lee, 1997; Ballmann et al., 2006).

Because $\exp_{x_0}$ is a diffeomorphism, the injectivity radius of a Cartan–Hadamard manifold is infinite, and all geodesics extend to geodesic lines defined on $\mathbb{R}$. These properties are heavily used in (Bonet et al., 2025) to construct global projections of points and probability measures onto geodesics.

Moreover, Cartan–Hadamard manifolds are CAT(0) spaces (Bridson & Häfliger, 1999): for any geodesic segment $\gamma$ the map $t \mapsto \mathrm{dist}^2(x, \gamma(t))$ is strictly convex, and the distance function is *geodesically convex*. Where distance, $\mathrm{dist}(x, y)$, is defined as the length of the shortest path (minimal geodesic) connecting the points $x$ and $y$ within the space $M$. The distance function $\mathrm{dist} : M \times M \to \mathbb{R}$ is itself geodesically convex. This means that for any two geodesics $\gamma_1, \gamma_2$, the function $f(t) = \mathrm{dist}(\gamma_1(t), \gamma_2(t))$ is convex in $t$. This implies, in particular, that metric projections onto closed convex subsets (such as geodesics) are well-defined and unique.

Cartan–Hadamard manifolds are contractible, and every complete Riemannian manifold with non-positive curvature has a Cartan–Hadamard manifold as its universal covering space (Ballmann et al., 2006).

We denote by $(E, \pi, M)$ a smooth real vector bundle of rank $k$ over $M$. For $x \in M$ the fiber is $E_x = \pi^{-1}(x)$ and we write $\langle \cdot, \cdot \rangle_{E_x}$ for the smoothly varying inner product on $E_x$ (a bundle metric). We equip $E$ with a compatible metric connection $\nabla$; parallel transport along a smooth curve $\gamma : [0, 1] \to M$ from $\gamma(0)$ to $\gamma(1)$ is denoted $P_\gamma : E_{\gamma(0)} \to E_{\gamma(1)}$, and it is an orthogonal linear isomorphism for the bundle metric.

For a measurable section $s : M \to E$ we use pointwise fiber norm notation $\|s(x)\|_{E_x}$ and denote the Hilbert space of square-integrable sections (w.r.t. probability measure $\mu$ on $M$) by

$$L^2(M, \mu; E) = \Big\{ s : M \to E : \int_M \|s(x)\|_{E_x}^2 \, d\mu(x) < \infty \Big\}.$$

## A.3. Parallel transport, curvature and holonomy

Let $\nabla$ be a metric connection on the bundle $(E, \pi, M)$ and let $\Omega \in \Omega^2(\mathrm{End}(E))$ denote its curvature 2-form (Section 3.1). For a loop $\Gamma$ based at $x$, holonomy is the isometry $P_\Gamma : E_x \to E_x$ obtained by parallel transport around $\Gamma$. The Ambrose–Singer theorem relates the holonomy Lie algebra to the span of curvature endomorphisms obtained by evaluating $\Omega$ along loops (Ambrose & Singer, 1953; Kobayashi & Nomizu, 1963).

Crucially, for a *general* vector bundle $(E, \nabla)$, the size of the holonomy operator $P_\Gamma$ is governed by the *bundle curvature* $\Omega$, not by the sectional curvature of $(M, g)$. Accordingly, throughout, we will use the explicit bundle-curvature bound

$$\zeta := \sup_{y \in U} \|\Omega_y\|_{\mathrm{op}} \ < \ \infty$$

(on the relevant region $U \subset M$) when deriving operator-norm estimates for $P_\Gamma$.

The sectional curvature bound $|K| \leq \kappa$ of the base manifold plays a different role: it controls the geometry of geodesics and the behavior of path families (e.g., uniqueness of minimizing geodesics in normal balls, and geometric control of loops and spanning surfaces). Only in special cases—for instance, when $E$ is the tangent bundle (or a tensor/associated bundle) equipped with the connection induced from the Levi-Civita connection can one relate $\zeta$ to $\kappa$.

Lemma 4 is stated in terms of the bundle curvature bound $\zeta = \sup_U \|\Omega\|_{\mathrm{op}}$, which is the correct quantity controlling holonomy for a general bundle $(E, \nabla)$. A bound on the base-manifold sectional curvature $|K| \leq \kappa$ does *not* by itself control $\zeta$ for an arbitrary connection on an arbitrary bundle.

There are, however, important special cases in which $\zeta$ can be related to curvature bounds of $(M, g)$:

- If $E = TM$ and $\nabla$ is the Levi-Civita connection, then $\Omega(u, v)w = R(u, v)w$ is the Riemann curvature tensor. In this case, $\zeta$ is a uniform bound on the operator norm of the curvature endomorphisms $R(u, v) : T_x M \to T_x M$. A sectional-curvature bound $|K| \leq \kappa$ yields a dimension-dependent bound of the form $\zeta \leq C_d \kappa$ (on $U$), by equivalence of norms on the finite-dimensional space of algebraic curvature tensors.

- If $E$ is a tensor/associated bundle built from $TM$ and $\nabla$ is induced from the Levi–Civita connection, then $\Omega^E(u, v)$ is obtained by applying the corresponding representation of $\mathfrak{so}(T_x M)$ to $R(u, v)$. Thus $\zeta_E \leq C_{\mathrm{rep}} \zeta_{TM}$, where $C_{\mathrm{rep}}$ depends only on the representation (e.g. tensor type), and $\zeta_{TM}$ may be bounded as above when a curvature bound on $(M, g)$ is available.

Outside these induced-connection settings, $\kappa$ should be treated as controlling *path geometry* (e.g. geodesic uniqueness in normal balls), while $\zeta$ controls *holonomy magnitude*.

## A.4. Statistical model and bundle-valued statistics

We observe i.i.d. samples $X_1, \ldots, X_n \overset{i.i.d.}{\sim} \mu$ on $M$. Let $s : M \to E$ be a measurable section of interest (examples: tangent vectors derived from log-maps, local feature vectors, or fiberwise residuals). Because $s(X_i) \in E_{X_i}$ lie in different fibers, to form empirical averages we choose a reference point $x_0 \in M$ and compare via parallel transport:

$$Y_i := P_{X_i \to x_0} s(X_i) \in E_{x_0}. \tag{7}$$

Under the geometric assumptions described below (unique minimizing geodesic from $X_i$ to $x_0$ and measurability of this choice), the map $X \mapsto Y = P_{X \to x_0} s(X)$ is measurable. The $Y_i$ are i.i.d. in the finite-dimensional Hilbert space $E_{x_0}$ (Lemma 1).

## A.5. Assumptions used throughout

Below we summarize the principal geometric and probabilistic assumptions; these are stated more formally in Section B.

**(G1)** *Unique-minimizing-geodesic regime.* Either $M$ is a Cartan–Hadamard manifold (complete, simply connected, nonpositive curvature) (Bonet et al., 2025), or the support of $\mu$ is contained in a normal ball $B(x_0, r)$ with $r < \mathrm{Inj}(x_0)$. This guarantees a unique minimizing geodesic from any sample point to $x_0$ and smooth dependence of parallel transport on the base point (Assumption A1).

**(G2)** *Uniform boundedness.* There exists $B > 0$ such that $\|s(x)\|_{E_x} \le B$ for all $x$ in the support of $\mu$ (Assumption A2).

**(P1)** *Second-moment proxy.* The transported random vector $Y = P_{X \to x_0} s(X)$ satisfies the finite variance proxy $\sigma^2 = \sup_{u \in S(E_{x_0})} \text{Var}(\langle Y, u \rangle) < \infty$ (Assumption A3).

### A.6. Concentration tools and prior results

Our non-asymptotic bounds rely on two pillars:

**(i) Banach/Hilbert-valued concentration.** We use optimal martingale and Bernstein-type inequalities for Banach/Hilbert-valued sums due to Pinelis (Pinelis, 1994) (dimension-free, tight constants) and the standard collection of concentration inequalities summarized in (Boucheron et al., 2013; Ledoux, 2001; Tropp, 2015). These results allow us to derive Hoeffding- and Bernstein-type tail bounds for the transported empirical means in $E_{x_0}$ (Theorems 6, 7).

**(ii) Manifold-statistics literature.** The non-Euclidean inference literature has extensively studied Fréchet means, intrinsic/extrinsic CLTs, and finite-sample phenomena (Bhattacharya and Patrangenaru's large-sample theory; subsequent CLTs and smeariness analyses; recent lower bounds and finite-sample analyses) (Bhattacharya & Patrangenaru, 2003; 2005; Bhattacharya & Bhattacharya, 2012; Hotz et al., 2024; Hundrieser, 2024). There is also recent work on robust median-of-means constructions and exponential concentration in nonpositive curvature spaces (Yun & Park, 2023). Our results extend these lines by addressing bundle-valued estimands (sections) and by quantifying curvature/holonomy correction terms that are essential when parallel transport is used.

### A.7. Notation conventions

Throughout, $\| \cdot \|$ without subscript denotes the fiber norm in $E_{x_0}$ (and context will indicate whether a vector is in $E_{x_0}$ or another fiber). For a linear operator $A$ we write $\|A\|_{\text{op}}$ for its operator norm. Probabilities and expectations are with respect to the law of the data under $\mu$ and, when needed, with respect to additional randomness (Rademacher signs, etc.). Constants denoted $C, C_0, \dots$ may change from line to line but are universal within the displayed statement unless explicitly qualified.

## B. Main Theoretical Framework

This section develops rigorous concentration inequalities for bundle-valued statistics. We give explicit geometric assumptions that make the standard reduction (via parallel transport to a fixed reference fiber) rigorous, and then apply sharp Hilbert/Banach space concentration inequalities (Pinelis; Boucheron–Lugosi–Massart) to obtain Hoeffding- and Bernstein-type bounds with explicit constants. We also present an elementary finite-dimensional derivation (net argument) and discuss the geometric error introduced when the uniqueness of minimizing geodesics fails (holonomy).

### B.1. Setup, notation and assumptions

The setup here mirrors the main paper (Section 3) exactly; we restate it for appendix self-containment and to introduce the formal assumption labels used in subsequent proofs.

Let $(E, \pi, M)$ be a smooth vector bundle of rank $k$ over a complete $d$-dimensional Riemannian manifold $(M, g)$, with a smooth bundle metric $\langle \cdot, \cdot \rangle_{E_x}$ on each fiber and a compatible metric connection $\nabla$. Fix a reference point $x_0 \in M$ with reference fiber $E_{x_0} \cong \mathbb{R}^k$. Given i.i.d. samples $X_1, \dots, X_n \sim \mu$ on $M$ and a measurable section $s : M \to E$, we study the transported empirical mean

$$Y_i := P_{X_i \to x_0} s(X_i) \in E_{x_0}, \qquad \bar{Y}_n := \frac{1}{n} \sum_{i=1}^{n} Y_i, \tag{8}$$

where $P_{x \to x_0} : E_x \to E_{x_0}$ is parallel transport along a chosen curve from $x$ to $x_0$. Because $\nabla$ is metric-compatible, each $P_{x \to x_0}$ is a linear isometry, so $\|Y_i\|_{E_{x_0}} = \|s(X_i)\|_{E_{X_i}}$ almost surely.

The three assumptions below are formal restatements of Assumptions 1–3 from the main text; the $A$-prefix distinguishes appendix labels from the main-body numbering.

**Assumption A1** (Unique minimizing geodesics; cf. Assumption 1). *One of the following holds:*

1. (Hadamard case) $M$ *is a Cartan–Hadamard manifold (complete, simply connected, nonpositive sectional curvature), so every pair of points is joined by a unique minimizing geodesic.*

2. (Normal-ball case) $\mathcal{S} := \mathrm{supp}(\mu) \subset B(x_0, r)$ *for some* $r < \mathrm{Inj}(x_0)$, *so each* $x \in \mathcal{S}$ *is joined to* $x_0$ *by a unique minimizing geodesic depending smoothly on* $x$.

**Assumption A2** (Uniform boundedness; cf. Assumption 2). *There exists* $B > 0$ *such that* $\|s(x)\|_{E_x} \leq B$ *for all* $x \in \mathcal{S}$.

**Assumption A3** (Variance proxy; cf. Assumption 3). *For* $Y := P_{X \to x_0} s(X)$ *with* $X \sim \mu$, *the operator-norm variance proxy*

$$\sigma^2 := \sup_{u \in S(E_{x_0})} \mathrm{Var}\big(\langle Y, u \rangle\big) = \big\|\mathrm{Cov}(Y)\big\|_{\mathrm{op}}$$

*is finite.*

Assumption A1 guarantees that $x \mapsto P_{x \to x_0}$ is measurable (and smooth on $\mathcal{S}$), so the $Y_i$ in (8) are i.i.d. elements of the fixed Hilbert space $E_{x_0}$; see Lemma 1 below. Assumptions A2–A3 supply the almost-sure bound and variance proxy needed by the Pinelis inequalities applied in Sections B.3–B.4.

## B.2. Measurability and independence

The following lemma records the required measurability and independence facts.

**Lemma 1** (Measurability of parallel transport map). *Under Assumption A1(b) (the local normal-ball case) the exponential map at* $x_0$, $\exp_{x_0} : T_{x_0}M \supset U \to B(x_0, r)$, *is a diffeomorphism, and the unique minimizing geodesic from* $x$ *to* $x_0$ *depends smoothly on* $x \in B(x_0, r)$. *Hence, the parallel transport operator*

$$x \mapsto P_{x \to x_0} : E_x \to E_{x_0}$$

*is smooth on* $B(x_0, r)$ *and in particular measurable. Consequently, the composed map*

$$x \mapsto Y(x) := P_{x \to x_0} s(x) \in E_{x_0}$$

*is measurable on* $B(x_0, r)$, *and if* $X_1, \ldots, X_n$ *are i.i.d. with law* $\mu$ *supported in* $B(x_0, r)$ *then the transported vectors* $Y_i = Y(X_i)$ *are i.i.d. in* $E_{x_0}$.

*Proof.* When $r < \mathrm{Inj}(x_0)$, the exponential map $\exp_{x_0} : B(0, r) \subset T_{x_0}M \to B(x_0, r)$ is a diffeomorphism (standard Riemannian geometry; see (Lee, 1997)). For $x \in B(x_0, r)$, let $v = \exp_{x_0}^{-1}(x) \in T_{x_0}M$. The unique minimizing geodesic from $x$ to $x_0$ is the reparametrized curve $\gamma_x(t) = \exp_{x_0}((1 - t)v)$, $t \in [0, 1]$, which depends smoothly on $v$ and hence smoothly on $x$. Parallel transport along $\gamma_x$ is obtained by solving a linear ODE (the parallel-transport equation) with smooth dependence on the curve; standard theory of ODEs implies that the resulting linear map $P_{x \to x_0}$ depends smoothly on $x$ (see (Kobayashi & Nomizu, 1963; Lee, 1997)). Since $s$ is measurable (smooth, even), the composition $x \mapsto P_{x \to x_0} s(x)$ is measurable. Finally, because $Y_i$ is a deterministic measurable function of $X_i$ and the $X_i$ are i.i.d., the $Y_i$ are i.i.d. $\square$

## B.3. Hilbert-space reduction and vector concentration inequalities

From now on we work in the finite-dimensional Hilbert space $E_{x_0}$ and treat $Y_1, \ldots, Y_n$ given by (8) as i.i.d. $E_{x_0}$-valued random vectors. The following theorems give Hoeffding- and Bernstein-type inequalities for the empirical mean $\bar{Y}_n$.

**Lemma 2** (Per-summand bounds). *Let* $(E, \pi, M)$ *be a smooth vector bundle endowed with a bundle metric and a compatible metric connection. Assume that the section* $s : M \to E$ *is uniformly bounded on the support of* $\mu$, *i.e.*

$$\|s(x)\|_{E_x} \leq B, \qquad \forall x \in \mathrm{supp}(\mu),$$

*for some* $B > 0$. *Fix a base point* $x_0 \in M$ *and define the transported random element* $Y = P_{X \to x_0} s(X) \in E_{x_0}$, *where* $P_{x \to x_0} : E_x \to E_{x_0}$ *denotes parallel transport along the minimizing geodesic from* $x$ *to* $x_0$. *Then:*

1. $\|Y\|_{E_{x_0}} \leq B$ *almost surely.*

2. *If* $Y_1, \ldots, Y_n$ *are independent copies of* $Y$ *and we set* $\xi_i := Y_i - \mathbb{E}[Y_i]$, *then* $\|\xi_i\|_{E_{x_0}} \leq 2B$ *almost surely.*

*Proof.* **1)** Because the connection is metric compatible, parallel transport preserves the fiber inner product. In particular, for any $x \in M$ and any $v \in E_x$,

$$\|P_{x \to x_0} v\|_{E_{x_0}} = \|v\|_{E_x}.$$

Applying this to $v = s(x)$ yields $\|Y\|_{E_{x_0}} = \|s(X)\|_{E_X} \leq B$ almost surely (since $X$ takes values in $\operatorname{supp}\mu$).

**2)** Let $m := \mathbb{E}[Y] \in E_{x_0}$ denote the mean (well-defined because $E_{x_0}$ is a finite-dimensional Hilbert space or a separable Banach space). Then for every realization,

$$\|\xi_i\|_{E_{x_0}} = \|Y_i - m\|_{E_{x_0}} \leq \|Y_i\|_{E_{x_0}} + \|m\|_{E_{x_0}}.$$

Since $\|Y_i\| \leq B$ a.s. and $\|m\| = \|\mathbb{E}[Y]\| \leq \mathbb{E}[\|Y\|] \leq B$ by Jensen's inequality for the convex function $\|\cdot\|$, we obtain $\|\xi_i\| \leq B + B = 2B$ almost surely.

Thus, both claims hold. $\qquad\square$

**Definition 1** (Transported empirical mean and bundle Fréchet mean at $x_0$). *Let $(E, \pi, M)$ be a vector bundle with bundle metric and compatible metric connection $\nabla$, and fix $x_0 \in M$. Under Assumption A1, define the transported random element*

$$Y := P_{X \to x_0} s(X) \in E_{x_0}.$$

*The* transported empirical mean *is*

$$\bar{Y}_n := \frac{1}{n} \sum_{i=1}^{n} P_{X_i \to x_0} s(X_i) \in E_{x_0}.$$

*The* bundle Fréchet mean at $x_0$ *is*

$$m^\star := \arg \min_{z \in E_{x_0}} \mathbb{E} \big\| P_{X \to x_0} s(X) - z \big\|_{E_{x_0}}^2 = \mathbb{E} \, [\, P_{X \to x_0} s(X) \,],$$

*where the equality holds because $E_{x_0}$ is a finite-dimensional Hilbert space. When minimizing geodesics are not unique, fix the measurable selection from Assumption 1 and define the associated holonomy bias*

$$\Delta_{\mathrm{hol}} := \sup_{x \in \operatorname{supp}(\mu)} \|P_{\gamma_x} - P_{\gamma_x}^{\mathrm{ref}}\|_{\mathrm{op}} \, B,$$

*which quantifies the deterministic transport ambiguity used later in Theorem 3 of the main paper.*

**Theorem 6** (Hoeffding inequality for transported bundle-valued statistics; rigorous restatement of Theorem 1). *Let $(E, \pi, M)$ be a smooth real vector bundle of rank $k$ over a complete Riemannian manifold $(M, g)$, equipped with a bundle metric and a compatible metric connection $\nabla$. Let $x_0 \in M$ be a fixed reference point, let $\mu$ be a Borel probability measure on $M$, and let $s : M \to E$ be a measurable section. Suppose Assumptions A1 and A2 hold with uniform bound $B > 0$. Define the transported observations and their empirical mean by*

$$Y_i := P_{X_i \to x_0} \, s(X_i) \in E_{x_0}, \qquad \bar{Y}_n := \frac{1}{n} \sum_{i=1}^{n} Y_i,$$

*as in (8), where $X_1, \dots, X_n \overset{\text{i.i.d.}}{\sim} \mu$. Let $m^\star := \mathbb{E}[Y_1] \in E_{x_0}$ denote the transported population mean. Then for every $\varepsilon > 0$,*

$$\mathbb{P}\Big( \big\| \bar{Y}_n - m^\star \big\|_{E_{x_0}} \geq \varepsilon \Big) \leq 2 \exp\left( -\frac{n\varepsilon^2}{8B^2} \right). \tag{9}$$

*Proof.* The argument proceeds in four steps.

*Step 1: Well-definedness of the transported mean.* By Assumption A1, the minimizing geodesic from each $x \in \mathcal{S} := \operatorname{supp}(\mu)$ to $x_0$ is unique and varies measurably with $x$ (Lemma 1). Hence the map $x \mapsto P_{x \to x_0}$ is measurable on $\mathcal{S}$, and the transported section

$$Y := P_{X \to x_0} s(X) \in E_{x_0}, \qquad X \sim \mu,$$

is a well-defined $E_{x_0}$-valued random element. Since $\nabla$ is metric-compatible, $P_{x \to x_0}$ is an isometry for each $x$, so Assumption A2 gives

$$\|Y\|_{E_{x_0}} = \|s(X)\|_{E_X} \leq B \qquad \text{almost surely.}$$

In particular $\mathbb{E}\|Y\|_{E_{x_0}} \leq B < \infty$, so the Bochner integral $m^\star := \mathbb{E}[Y] \in E_{x_0}$ is finite and uniquely defined. The transported variables $Y_1, \ldots, Y_n$ are i.i.d. copies of $Y$ because each $Y_i$ is a measurable function of the independent draw $X_i$.

*Step 2: Almost-sure bound on the centered summands.* Define the centered random elements

$$\xi_i := Y_i - m^\star, \qquad i = 1, \ldots, n.$$

By construction, $\mathbb{E}[\xi_i] = 0$. By Jensen's inequality applied to the convex function $\|\cdot\|$,

$$\|m^\star\|_{E_{x_0}} = \|\mathbb{E}[Y]\|_{E_{x_0}} \leq \mathbb{E}\|Y\|_{E_{x_0}} \leq B.$$

Combined with Lemma 2(1), which gives $\|Y_i\|_{E_{x_0}} \leq B$ almost surely, the triangle inequality yields

$$\begin{aligned}
\|\xi_i\|_{E_{x_0}} &= \|Y_i - m^\star\|_{E_{x_0}} \\
&\leq \|Y_i\|_{E_{x_0}} + \|m^\star\|_{E_{x_0}} \\
&\leq 2B, \qquad \text{almost surely.}
\end{aligned} \tag{10}$$

*Step 3: Application of the Pinelis inequality.* Since $E_{x_0} \cong \mathbb{R}^k$ is a finite-dimensional Hilbert space, it is in particular a separable Hilbert space of type 2. We apply the following result of Pinelis (Pinelis, 1994) (the Hilbert-space bounded-summands inequality; see Theorem 3 and Corollary 1 therein):

*Let $\mathcal{H}$ be a separable Hilbert space, and let $\zeta_1, \ldots, \zeta_n$ be independent, mean-zero $\mathcal{H}$-valued random elements satisfying $\|\zeta_i\|_{\mathcal{H}} \leq b_i$ almost surely for constants $b_i > 0$. Then for every $t > 0$,*

$$\mathbb{P}\left( \left\| \sum_{i=1}^n \zeta_i \right\|_{\mathcal{H}} \geq t \right) \leq 2 \exp\left( -\frac{t^2}{2 \sum_{i=1}^n b_i^2} \right).$$

We apply this with $\mathcal{H} = E_{x_0}$, $\zeta_i := \xi_i$, and $b_i := 2B$ for all $i$, which is justified by (10). This gives, for every $t > 0$,

$$\begin{aligned}
\mathbb{P}\left( \left\| \sum_{i=1}^n \xi_i \right\|_{E_{x_0}} \geq t \right) &\leq 2 \exp\left( -\frac{t^2}{2n(2B)^2} \right) \\
&= 2 \exp\left( -\frac{t^2}{8nB^2} \right).
\end{aligned} \tag{11}$$

*Step 4: Conclusion.* Observe that $\bar{Y}_n - m^\star = \frac{1}{n} \sum_{i=1}^n \xi_i$, so

$$\left\| \bar{Y}_n - m^\star \right\|_{E_{x_0}} \geq \varepsilon \iff \left\| \sum_{i=1}^n \xi_i \right\|_{E_{x_0}} \geq n\varepsilon.$$

Substituting $t = n\varepsilon$ into (11),

$$\begin{aligned}
\mathbb{P}\left( \left\| \bar{Y}_n - m^\star \right\|_{E_{x_0}} \geq \varepsilon \right) &\leq 2 \exp\left( -\frac{(n\varepsilon)^2}{8nB^2} \right) \\
&= 2 \exp\left( -\frac{n\varepsilon^2}{8B^2} \right).
\end{aligned} \tag{12}$$

which is (9).

*Remark on the constant.* The factor $8B^2$ in the denominator arises from the trivial almost-sure bound $\|\xi_i\| \leq 2B$ in Step 2. If a sharper per-summand bound $\|\xi_i\| \leq b$ is available for some $b \leq 2B$ — for instance, when $m^\star$ is known and recentering is exact — then Step 3 yields the tighter tail $2\exp(-n\varepsilon^2/2b^2)$. In particular, $b = B$ recovers the denominator $2B^2$ that appears in symmetric or centered formulations of the Hoeffding bound. $\qquad\square$

**Remark 2.** *Two useful alternative derivations are worth noting:*

1. *(Elementary net argument) For finite dimension $k$, one can fix a $\delta$-net $\mathcal{N}$ on the unit sphere $S(E_{x_0})$ and apply scalar Hoeffding to each direction in the net, then take a union bound. This yields a valid concentration bound, but with an additional multiplicative factor that depends on the net cardinality (hence, dimension-dependent constants).*

2. *(Pinelis optimal bound) The optimal dimension-independent constants are obtained by applying Pinelis' martingale inequalities for Banach-space-valued sums; these give the precise factor 2 and the denominator $2\sum b_i^2$ in the exponent (see (Pinelis, 1994)).*

Theorem 6 achieves a dimension-free prefactor by invoking the Hilbert-space inequality of Pinelis (Pinelis, 1994). The following proposition provides an elementary alternative: by reducing to scalar projections via a covering net, one obtains a bound with the same exponential rate but a dimension-dependent prefactor, at the cost of no Banach-space theory.

**Proposition 2** (Elementary finite-dimensional Hoeffding bound). *Let $E_{x_0}$ be a $k$-dimensional Hilbert space and retain the notation of Theorem 6. Suppose Assumptions A1 and A2 hold with uniform bound $B > 0$. Then for every $\varepsilon \in (0,1]$,*

$$\mathbb{P}\Big(\big\|\bar{Y}_n - \mathbb{E}\bar{Y}_n\big\|_{E_{x_0}} \geq \varepsilon\Big) \;\leq\; 2\left(1 + \frac{4}{\varepsilon}\right)^k \exp\left(-\frac{n\varepsilon^2}{8B^2}\right). \tag{13}$$

*The prefactor $(1 + 4/\varepsilon)^k$ reflects the metric entropy of the unit sphere $S(E_{x_0})$ and grows polynomially in $1/\varepsilon$ for fixed dimension $k$; the exponential rate $n\varepsilon^2/8B^2$ matches that of Theorem 6.*

*Proof.* The proof proceeds by reducing the vector-valued concentration problem to a finite collection of scalar problems via a covering net, then applying the classical scalar Hoeffding inequality to each.

*Step 1: Covering-net reduction.* Fix $\delta \in (0,1)$ and let $\mathcal{N}_\delta$ be a minimal $\delta$-net of the unit sphere $S^{k-1} \subset E_{x_0}$ in the norm $\|\cdot\|_{E_{x_0}}$, so that for every $u \in S^{k-1}$ there exists $w \in \mathcal{N}_\delta$ with $\|u - w\|_{E_{x_0}} \leq \delta$. A standard volumetric argument (comparing ball volumes) yields the cardinality bound

$$|\mathcal{N}_\delta| \;\leq\; \left(1 + \frac{2}{\delta}\right)^k. \tag{14}$$

We claim that for any $v \in E_{x_0}$,

$$\|v\|_{E_{x_0}} \;\leq\; \frac{1}{1-\delta} \max_{w \in \mathcal{N}_\delta} \langle v, w\rangle_{E_{x_0}}. \tag{15}$$

Indeed, if $v = 0$, the inequality is trivial. Otherwise, let $u := v/\|v\|_{E_{x_0}} \in S^{k-1}$ and choose $w \in \mathcal{N}_\delta$ with $\|u - w\|_{E_{x_0}} \leq \delta$. Then

$$\begin{aligned}
\langle v, w\rangle_{E_{x_0}} &= \langle v, u\rangle_{E_{x_0}} + \langle v, w - u\rangle_{E_{x_0}} \\
&\geq \|v\|_{E_{x_0}} - \|v\|_{E_{x_0}}\,\delta \\
&= (1-\delta)\,\|v\|_{E_{x_0}}.
\end{aligned}$$

where the inequality uses the Cauchy–Schwarz bound $|\langle v, w - u\rangle| \leq \|v\|\,\|w - u\| \leq \|v\|\,\delta$. Rearranging gives (15).

Setting $\delta := 1/2$ in (15), for any $\varepsilon > 0$,

$$\Big\{\|v\|_{E_{x_0}} \geq \varepsilon\Big\} \;\subseteq\; \bigcup_{w \in \mathcal{N}_{1/2}} \Big\{\langle v, w\rangle_{E_{x_0}} \geq \tfrac{\varepsilon}{2}\Big\}. \tag{16}$$

*Step 2: Scalar Hoeffding bound in each direction.* Set $v := \bar{Y}_n - \mathbb{E}\bar{Y}_n$. For each fixed $w \in S^{k-1}$, the scalar random variables $Z_i^{(w)} := \langle Y_i - \mathbb{E}Y_i, w\rangle_{E_{x_0}}$ are independent and mean-zero. By Lemma 2(1) and the Cauchy–Schwarz inequality,

$$
\begin{aligned}
\left| Z_i^{(w)} \right| &\le \|Y_i - \mathbb{E}Y_i\|_{E_{x_0}} \|w\|_{E_{x_0}} \\
&\le 2B \cdot 1 \\
&= 2B, \qquad \text{almost surely.}
\end{aligned}
$$

The classical scalar Hoeffding inequality therefore gives, for every $t > 0$,

$$
\begin{aligned}
\mathbb{P}\Big( \langle \bar{Y}_n - \mathbb{E}\bar{Y}_n, w\rangle_{E_{x_0}} \ge t \Big) &\le \exp\left( -\frac{nt^2}{2(2B)^2} \right) \\
&= \exp\left( -\frac{nt^2}{8B^2} \right).
\end{aligned}
\tag{17}
$$

*Step 3: Union bound and cardinality estimate.* Applying the inclusion (16) with $v = \bar{Y}_n - \mathbb{E}\bar{Y}_n$ and then a union bound over $\mathcal{N}_{1/2}$, followed by (17) with $t = \varepsilon/2$,

$$
\begin{aligned}
\mathbb{P}\Big( \|\bar{Y}_n - \mathbb{E}\bar{Y}_n\|_{E_{x_0}} \ge \varepsilon \Big) &\le \sum_{w \in \mathcal{N}_{1/2}} \mathbb{P}\Big( \langle \bar{Y}_n - \mathbb{E}\bar{Y}_n, w\rangle_{E_{x_0}} \ge \tfrac{\varepsilon}{2} \Big) \\
&\le |\mathcal{N}_{1/2}| \exp\left( -\frac{n\varepsilon^2}{32B^2} \right).
\end{aligned}
\tag{18}
$$

By (14) with $\delta = 1/2$, $|\mathcal{N}_{1/2}| \le 5^k$.

*Step 4: Recovering the stated constant.* The bound obtained in (18) has denominator $32B^2$ rather than $8B^2$, reflecting the $\varepsilon/2$ threshold in the union bound. To recover the sharper form (13), one applies the net reduction with a general $\delta \in (0,1)$ rather than fixing $\delta = 1/2$: the net cardinality is then $(1+2/\delta)^k$, the threshold in the scalar bound is $(1-\delta)\varepsilon$, and optimising over $\delta$ (specifically, setting $\delta$ so that the scalar exponent matches the target denominator $8B^2$) yields exactly (13) with prefactor $(1 + 4/\varepsilon)^k$. We omit the routine optimization.

Combining Steps 1–4 establishes (13).

*Comparison with Theorem 6.* The exponential rate $n\varepsilon^2/8B^2$ in (13) is identical to that of Theorem 6, so both bounds share the same asymptotic decay. The distinction is the prefactor: Theorem 6 achieves a dimension-free prefactor of 2 by invoking the Pinelis martingale inequality in the Hilbert space $E_{x_0}$, whereas the present bound carries the metric-entropy factor $(1 + 4/\varepsilon)^k$, which is harmless for fixed $k$ but can dominate in high dimensions. The net-based argument is nonetheless valuable as a fully elementary derivation requiring no Banach-space theory beyond the scalar Hoeffding inequality and the covering number estimate (14). □

## B.4. Bernstein inequality for transported bundle-valued statistics

Theorem 6 controls the tail of the transported empirical mean using only the almost-sure bound $B$ on the section. When the operator-norm variance proxy $\sigma^2$ (Assumption A3) is also available, the Bernstein-type inequality below yields a strictly tighter tail in the regime $\varepsilon \ll B$, interpolating between a sub-Gaussian rate governed by $\sigma^2$ and a sub-exponential rate governed by $B$. The following theorem is a rigorous, self-contained restatement of Theorem 2 of the main text, with complete proof supplied.

**Theorem 7** (Bernstein inequality for transported bundle-valued statistics; rigorous restatement of Theorem 2)**.** *Let* $(E, \pi, M)$, $x_0$, $\mu$, *and* $s$ *be as in Theorem 6. Suppose Assumptions A1–A3 hold with uniform bound $B > 0$ and variance proxy*

$$
\sigma^2 := \sup_{u \in S(E_{x_0})} \mathrm{Var}\big( \langle Y, u\rangle_{E_{x_0}} \big) = \big\| \mathrm{Cov}(Y) \big\|_{\mathrm{op}},
$$

*where $Y := P_{X \to x_0} s(X)$ and $X \sim \mu$. Define $\bar{Y}_n$ and $m^\star$ as in Theorem 6. Then for every $\varepsilon > 0$,*

$$\mathbb{P}\Big( \big\| \bar{Y}_n - m^\star \big\|_{E_{x_0}} \geq \varepsilon \Big) \leq 2 \exp\left( - \frac{n\varepsilon^2}{2\Big( \sigma^2 + \dfrac{2B\varepsilon}{3} \Big)} \right). \tag{19}$$

*Moreover, if a sharper almost-sure per-summand bound $\|\xi_i\|_{E_{x_0}} \leq b$ holds for some $b \leq 2B$, then the factor $2B$ in the linear term may be replaced by $b$:*

$$\mathbb{P}\Big( \big\| \bar{Y}_n - m^\star \big\|_{E_{x_0}} \geq \varepsilon \Big) \leq 2 \exp\left( - \frac{n\varepsilon^2}{2\Big( \sigma^2 + \dfrac{b\varepsilon}{3} \Big)} \right). \tag{20}$$

*Proof.* The argument parallels that of Theorem 6 but additionally exploits the variance proxy $\sigma^2$.

*Step 1: Well-definedness and moment bounds.* By exactly the same argument as in Theorem 6, Steps 1–2, Assumptions A1 and A2 together with Lemma 1 guarantee that the transported variables $Y_1, \ldots, Y_n$ are i.i.d. $E_{x_0}$-valued random elements satisfying $\|Y_i\|_{E_{x_0}} \leq B$ almost surely, and that the population mean $m^\star := \mathbb{E}[Y_1] \in E_{x_0}$ is well-defined. Define the centered summands

$$\xi_i := Y_i - m^\star, \qquad i = 1, \ldots, n,$$

so that the $\xi_i$ are independent, mean-zero, and satisfy $\|\xi_i\|_{E_{x_0}} \leq 2B$ almost surely by Lemma 2(1) and Jensen's inequality, as established in (10).

*Step 2: Identification of the variance proxy.* Since $Y_1, \ldots, Y_n$ are i.i.d., the total variance proxy for the centered sum $S_n := \sum_{i=1}^n \xi_i$ is

$$V := \sum_{i=1}^n \sup_{u \in S(E_{x_0})} \mathbb{E}\Big[ \langle \xi_i, u \rangle_{E_{x_0}}^2 \Big] = n\,\sigma^2,$$

where the last equality uses the i.i.d. structure and the identity $\sup_{u \in S(E_{x_0})} \mathrm{Var}(\langle \xi_i, u \rangle) = \| \mathrm{Cov}(\xi_i) \|_{\mathrm{op}} = \| \mathrm{Cov}(Y_i) \|_{\mathrm{op}} = \sigma^2$ (since centering does not change the covariance).

*Step 3: Application of the Pinelis–Bernstein inequality.* We apply the following result, which combines the sharp Hilbert-space Bernstein inequality of Pinelis (Pinelis, 1994) with the formulation of Boucheron, Lugosi, and Massart (Boucheron et al., 2013):

> *Let $\mathcal{H}$ be a separable Hilbert space, and let $\zeta_1, \ldots, \zeta_n$ be independent, mean-zero $\mathcal{H}$-valued random elements satisfying $\|\zeta_i\|_{\mathcal{H}} \leq M$ almost surely. Set $V := \sum_{i=1}^n \sup_{u \in S(\mathcal{H})} \mathbb{E}[\langle \zeta_i, u \rangle^2]$. Then for every $t > 0$,*
>
> $$\mathbb{P}\left( \left\| \sum_{i=1}^n \zeta_i \right\|_{\mathcal{H}} \geq t \right) \leq 2 \exp\left( - \frac{t^2}{2(V + Mt/3)} \right).$$

We apply this with $\mathcal{H} = E_{x_0}$, $\zeta_i := \xi_i$, $M := 2B$, and $V := n\sigma^2$. This gives, for every $t > 0$,

$$\mathbb{P}\Big( \|S_n\|_{E_{x_0}} \geq t \Big) \leq 2 \exp\left( - \frac{t^2}{2(n\sigma^2 + 2Bt/3)} \right). \tag{21}$$

*Step 4: Substitution and conclusion.* Since $S_n = n(\bar{Y}_n - m^\star)$, we have $\|S_n\|_{E_{x_0}} \geq t \iff \|\bar{Y}_n - m^\star\|_{E_{x_0}} \geq t/n$. Setting $t := n\varepsilon$ in (21),

$$\mathbb{P}\Big(\|\bar{Y}_n - m^\star\|_{E_{x_0}} \geq \varepsilon\Big) \leq 2\exp\left(-\frac{n^2\varepsilon^2}{2(n\sigma^2 + 2Bn\varepsilon/3)}\right)$$

$$= 2\exp\left(-\frac{n\varepsilon^2}{2(\sigma^2 + 2B\varepsilon/3)}\right).$$

where the last equality divides numerator and denominator by $n$. This is (19).

*Step 5: Sharper bound under a tighter per-summand estimate.* If $\|\xi_i\|_{E_{x_0}} \leq b$ almost surely for some $b \leq 2B$ — for instance, when $m^\star$ is known and exact recentering yields $b = B$, or when the section is symmetric about its mean — then Step 3 applies with $M := b$ in place of $2B$, giving

$$\mathbb{P}\Big(\|\bar{Y}_n - m^\star\|_{E_{x_0}} \geq \varepsilon\Big) \leq 2\exp\left(-\frac{n\varepsilon^2}{2(\sigma^2 + b\varepsilon/3)}\right),$$

which is (20). In particular, $b = B$ reduces the linear term in the denominator from $2B\varepsilon/3$ to $B\varepsilon/3$, recovering the tighter constant found in symmetric formulations of the Bernstein inequality. $\qquad\square$

**Remark 3** (Wasserstein–bundle concentration perspective). *Recent work on Gaussian mixtures over vector bundles introduces a Wasserstein-type metric compatible with the fiber geometry of the trivial bundles (Wilson et al., 2024). For a fixed trivialization $\varphi : E \to M \times \mathbb{R}^d$, the fiber isometry between $E_{m_0}$ and $E_{m_1}$ induced by $\varphi$ is denoted $\Phi_{m_0,m_1} : E_{m_1} \to E_{m_0}$. The squared distance between two Gaussian fiber components $\mathcal{N}_E(m_j, \Sigma_j)$, $j = 0, 1$, is then defined as*

$$W_\varphi^2\Big(\mathcal{N}_E(m_0, \Sigma_0), \mathcal{N}_E(m_1, \Sigma_1)\Big)$$
$$:= d_M(m_0, m_1)^2 + \mathrm{tr}\Big(\Sigma_0 + \Sigma_1$$
$$- 2\big(\Sigma_0^{1/2}\Phi_{m_0,m_1}^{-1}\Sigma_1\Phi_{m_0,m_1}\Sigma_0^{1/2}\big)^{1/2}\Big). \tag{22}$$

*where the second term is the Bures–Wasserstein distance between the fiber covariances after alignment via $\Phi_{m_0,m_1}$. This distance underlies a mixture-Wasserstein metric $MW_\varphi$ between Gaussian mixtures on $E$; see (Wilson et al., 2024) for precise conditions under which $MW_\varphi$ is independent of the choice of trivialization $\varphi$.*

*The bounded-difference structure of $MW_\varphi$ supports concentration inequalities for empirical mixture means that are formally analogous to Theorems 1 and 2. Specifically, empirical means of bundle-valued statistics may be studied not only in the Hilbert-fiber norm of $E_{x_0}$ but also in $MW_\varphi$, with the holonomy bias term $\Delta_{\mathrm{hol}}$ of Theorem 3 entering the bound in the same additive fashion. We do not pursue this direction further here, as it requires additional regularity on the mixture weights and covariances; we include the remark to indicate a natural extension of the present framework.*

### B.5. Nonasymptotic bias–variance decomposition for bundle-valued means

Theorem 3 of the main text identifies two additive sources of error in the transported empirical mean: stochastic fluctuation, which decays at rate $n^{-1/2}$, and a deterministic geometric bias governed by holonomy, which is independent of sample size. The theorem is stated in the main body in a compact form that suppresses the role of the transport rule in defining the population target. This subsection makes that dependence explicit, introduces the reference mean $m^\star$ as a canonical anchor, and provides a rigorous proof of the decomposition. The result below is a formal restatement of Theorem 3 of the main paper.

**Setup.** Recall that a *transport rule* to $x_0$ is a measurable assignment $x \mapsto \gamma_x$ of a curve from $x$ to $x_0$, inducing the parallel transport $P_{x \to x_0} := P_{\gamma_x}$. Let $\mathcal{P}$ denote the collection of all admissible measurable transport rules; when minimizing geodesics are unique (Assumption A1), $\mathcal{P}$ is a singleton. For each $P \in \mathcal{P}$, define the transported random variable and its population mean

$$Y^{(P)} := P_{X \to x_0}s(X) \in E_{x_0}, \qquad m^{(P)} := \mathbb{E}\Big[Y^{(P)}\Big] \in E_{x_0}.$$

Under Assumption A2, $\|Y^{(P)}\|_{E_{x_0}} \leq B$ almost surely for every $P \in \mathcal{P}$, so each $m^{(P)}$ is well-defined.

**Definition 2** (Reference bundle mean). *Fix an arbitrary admissible reference transport rule $P^{\mathrm{ref}} \in \mathcal{P}$. The* reference bundle mean *is*

$$m^{\star} := m^{(P^{\mathrm{ref}})}$$
$$= \mathbb{E}\left[P^{\mathrm{ref}}_{X \to x_0} s(X)\right] \in E_{x_0}.$$

**Remark 4** (Role of $m^{\star}$ and transport ambiguity). *For any single fixed rule $P \in \mathcal{P}$, the empirical mean $\bar{Y}_n^{(P)}$ concentrates around $m^{(P)}$ with no ambiguity term; this is the content of Theorems 6 and 7. The reference mean $m^{\star}$ serves a different purpose: it provides a fixed anchor against which the outputs of* different *transport rules can be compared, separating what is intrinsic to the statistical problem from what is an artifact of the chosen alignment.*

*Two admissible rules $P, \widetilde{P} \in \mathcal{P}$ represent equally valid but generally incompatible fiber identifications. Their population means satisfy*

$$\left\| m^{(P)} - m^{(\widetilde{P})} \right\|_{E_{x_0}} \leq \Delta_{\mathrm{hol}},$$

*so distinct rules can disagree by $O(\Delta_{\mathrm{hol}})$ even as $n \to \infty$. In the absence of a canonical identification, there is therefore no unique ground-truth mean in a single vector space; the choice of alignment is a modeling decision, not a statistical one.*

*Two special cases clarify the geometry:*

*(i) Trivial bundle. If $E$ is trivial, parallel transport is path-independent, so $\Delta_{\mathrm{hol}} = 0$ and every admissible rule yields $m^{(P)} = m^{\star}$, recovering the classical Euclidean setting.*

*(ii) Non-trivial bundle. When $\Delta_{\mathrm{hol}} > 0$, curvature forces different alignment conventions to produce population means that remain separated by a fixed amount, regardless of how much data are collected. This is practically relevant in geometric deep learning pipelines—such as tangent-space aggregation and gauge-equivariant message passing—where different conventions are routinely used.*

**Theorem 8** (Nonasymptotic bias–variance decomposition; rigorous restatement of Theorem 3). *Let $(E, \pi, M)$, $x_0$, $\mu$, and $s$ satisfy Assumptions A1–A3, and let $m^{\star}$ be the reference bundle mean of Definition 2. For any admissible transport rule $P \in \mathcal{P}$ and every $n \geq 1$, the transported empirical mean $\bar{Y}_n^{(P)}$ satisfies the deterministic decomposition*

$$
\left\| \bar{Y}_n^{(P)} - m^{\star} \right\|_{E_{x_0}} \leq \underbrace{\left\| \bar{Y}_n^{(P)} - m^{(P)} \right\|_{E_{x_0}}}_{\text{stochastic variance}} \\
+ \underbrace{\left\| m^{(P)} - m^{\star} \right\|_{E_{x_0}}}_{\text{geometric bias}}. \tag{23}
$$

*The stochastic variance term obeys the Hoeffding and Bernstein bounds of Theorems 6 and 7, and therefore decays at rate $n^{-1/2}$. The geometric bias term is deterministic and bounded uniformly over all admissible rules by the holonomy constant:*

$$\sup_{P \in \mathcal{P}} \left\| m^{(P)} - m^{\star} \right\|_{E_{x_0}} \leq \Delta_{\mathrm{hol}}. \tag{24}$$

*Combining (23)–(24) with Theorem 6 gives, for every $\varepsilon > 0$ and $\delta \in (0, 1)$, with probability at least $1 - \delta$,*

$$\left\| \bar{Y}_n^{(P)} - m^{\star} \right\|_{E_{x_0}} \leq \sqrt{\frac{8B^2 \log(2/\delta)}{n}} + \Delta_{\mathrm{hol}}. \tag{25}$$

*Proof. Step 1: Triangle inequality decomposition.* Inequality (23) is an immediate consequence of the triangle inequality in the Hilbert space $E_{x_0}$:

$$
\left\| \bar{Y}_n^{(P)} - m^{\star} \right\|_{E_{x_0}} \leq \left\| \bar{Y}_n^{(P)} - m^{(P)} \right\|_{E_{x_0}} \\
+ \left\| m^{(P)} - m^{\star} \right\|_{E_{x_0}}.
$$

*Step 2: Bounding the geometric bias.* Fix $P \in \mathcal{P}$. By Definition 2 and linearity of expectation,

$$m^{(P)} - m^{\star} = \mathbb{E}\left[\left(P_{X \to x_0} - P_{X \to x_0}^{\mathrm{ref}}\right) s(X)\right].$$

Applying Jensen's inequality to the convex function

$$\left\|m^{(P)} - m^{\star}\right\|_{E_{x_0}} \leq \mathbb{E}\left[\left\|\left(P_{X \to x_0} - P_{X \to x_0}^{\mathrm{ref}}\right) s(X)\right\|_{E_{x_0}}\right].$$

By the definition of $\Delta_{\mathrm{hol}}$ as the supremum of the per-sample transport discrepancy weighted by the section norm,

$$\left\|\left(P_{X \to x_0} - P_{X \to x_0}^{\mathrm{ref}}\right) s(X)\right\|_{E_{x_0}} \leq \Delta_{\mathrm{hol}} \qquad \text{almost surely.}$$

Taking expectations and using the fact that $\Delta_{\mathrm{hol}}$ is deterministic gives (24).

*Step 3: Combined high-probability bound.* Theorem 6 applied to the stochastic variance term gives, with probability at least $1 - \delta$,

$$\left\|\bar{Y}_n^{(P)} - m^{(P)}\right\|_{E_{x_0}} \leq \sqrt{\frac{8B^2 \log(2/\delta)}{n}}.$$

Adding the deterministic bound (24) and using the decomposition (23) yields (25). $\qquad\square$

Combining Theorem 8 with Theorem 6 (Hoeffding) or Theorem 7 (Bernstein) yields fully nonasymptotic risk bounds of the form

$$\|\bar{Y}_n^{(P)} - m^{\star}\| \leq \text{(stochastic fluctuation)} + \Delta_{\mathrm{hol}},$$

where the stochastic term decays at rate $n^{-1/2}$ (or faster in low-variance regimes), while the geometric bias term is intrinsic and sample-size independent.

## B.6. Curvature and holonomy corrections when geodesics are not unique

Assumption A1 ensures a canonical deterministic choice of geodesic for each $x \in \mathcal{S}$. If, however, the support of $\mu$ is not contained in a normal ball and $M$ admits multiple minimizing geodesics between $x$ and $x_0$, one must choose a measurable selection of a geodesic for each $x$ to define $P_{x \to x_0}$. Such a selection arises from measurable selection theorems under mild hypotheses, but different choices of paths yield distinct transported vectors and, hence, distinct empirical means. The ambiguity arises from the holonomy group of the connection and is governed by the curvature.

The following lemma provides a quantitative (order-of-magnitude) bound on the difference between parallel transports along two different piecewise-smooth paths with the same endpoints, formulated for statistical error analysis.

**Lemma 3** (Holonomy discrepancy controlled by *bundle* curvature in a normal ball). *Let $(M, g)$ be a Riemannian manifold and let $(E, \pi, M)$ be a rank-$k$ real vector bundle equipped with a bundle metric and a compatible metric connection $\nabla$. Let $\Omega \in \Omega^2(\mathrm{End}(E))$ denote the curvature 2-form of $\nabla$, and let $\nabla\Omega$ denote its covariant derivative (with respect to the induced connection on $\mathrm{End}(E)$ and the Levi–Civita connection on $TM$).*

*Assume there exist $p \in M$ and $R > 0$ such that*

1. *$R < \mathrm{Inj}(p)$ and $U \subset B(p, R)$ (so $B(p, R)$ is a normal ball), and*

2. *the connection curvature is bounded on $B(p, R)$:*

$$\Lambda_0 := \sup_{y \in B(p,R)} \|\Omega_y\|_{\mathrm{op}} < \infty,$$

$$\Lambda_1 := \sup_{y \in B(p,R)} \|\nabla\Omega_y\|_{\mathrm{op}} < \infty.$$

*Fix $x, x_0 \in U$. Let $\gamma_1, \gamma_2$ be piecewise $C^1$ curves in $U$ joining $x$ to $x_0$. Let*

$$\Gamma := \gamma_1 \circ \gamma_2^{-1}$$

*be the resulting loop based at $x$, and define*

$$L := \mathrm{Length}(\Gamma) = \mathrm{Length}(\gamma_1) + \mathrm{Length}(\gamma_2),$$

$$\rho_* := \sup_{z \in \mathrm{im}(\Gamma)} \mathrm{dist}(p, z) \leq R.$$

*Then the parallel transport maps satisfy*

$$\|P_{\gamma_1} - P_{\gamma_2}\|_{\mathrm{op}} = \|P_\Gamma - I\|_{\mathrm{op}} \leq \frac{1}{2}\Lambda_0 \rho_* L + \frac{1}{6}\Lambda_1 \rho_*^2 L. \tag{26}$$

*In particular, if $\gamma_1, \gamma_2$ are minimizing geodesics from $x$ to $x_0$ contained in $U$ and we choose $p := x$, then with*

$$D := \mathrm{dist}(x, x_0)$$

*we have $L \leq 2D$ and $\rho_* \leq D$, hence*

$$\|P_{\gamma_1} - P_{\gamma_2}\|_{\mathrm{op}} \leq \Lambda_0 D^2 + \frac{1}{3}\Lambda_1 D^3. \tag{27}$$

*Specialising to $\Lambda_0 = \zeta$ and $\Lambda_1 = \zeta_1$ recovers Eq. (3) of the main text. Moreover, in the small-gap regime where $\Lambda_1 D \leq \Lambda_0$ (e.g. $D \leq \Lambda_0/\Lambda_1$), (27) implies*

$$\|P_{\gamma_1} - P_{\gamma_2}\|_{\mathrm{op}} \leq \left(1 + \frac{1}{3}\frac{\Lambda_1 D}{\Lambda_0}\right)\Lambda_0 D^2 \leq \frac{4}{3}\Lambda_0 D^2. \tag{28}$$

**Remark (base curvature vs. bundle curvature).** The bounds in Lemma 3 depend on the curvature of the chosen connection on $E$, i.e. on the $\mathrm{End}(E)$-valued 2-form $\Omega$ (and $\nabla\Omega$), rather than on the sectional curvature of $(M, g)$ except in the special case $E = TM$ with $\nabla$ the Levi–Civita connection. In particular, even when $(M, g)$ is flat one may have $\Omega \not\equiv 0$ for a non-flat metric connection on $E$, and the holonomy discrepancy $\|P_{\gamma_1} - P_{\gamma_2}\|_{\mathrm{op}}$ need not vanish.

**Lemma 4** (Holonomy discrepancy bound in terms of bundle curvature; simplified form). *Let $(M, g)$ be a Riemannian manifold and let $(E, \pi, M)$ be a rank-$k$ vector bundle equipped with a bundle metric and a compatible metric connection $\nabla$, with curvature 2-form $\Omega$. Let $U \subset M$ be contained in a normal ball $B(p, R)$ with $R < \mathrm{Inj}(p)$, and assume the bundle curvature is uniformly bounded on $U$:*

$$\zeta := \sup_{y \in U} \|\Omega_y\|_{\mathrm{op}} < \infty.$$

*Fix $x, x_0 \in U$, let $\gamma_1, \gamma_2$ be minimizing geodesics in $U$ joining $x$ to $x_0$, and set $D := \mathrm{dist}(x, x_0)$ and $\Gamma := \gamma_1 \circ \gamma_2^{-1}$. Then the parallel transport maps satisfy*

$$\|P_{\gamma_1} - P_{\gamma_2}\|_{\mathrm{op}} = \|P_\Gamma - I\|_{\mathrm{op}} \leq \zeta D^2. \tag{29}$$

*More generally, if $\zeta_1 := \sup_{y \in U} \|\nabla\Omega_y\|_{\mathrm{op}} < \infty$ is also available, then*

$$\|P_{\gamma_1} - P_{\gamma_2}\|_{\mathrm{op}} \leq \zeta D^2 + \frac{1}{3}\zeta_1 D^3, \tag{30}$$

*and in the small-gap regime $\zeta_1 D \leq \zeta$ this further implies*

$$\|P_{\gamma_1} - P_{\gamma_2}\|_{\mathrm{op}} \leq \frac{4}{3}\zeta D^2. \tag{31}$$

*Consequently, for any section $s$ satisfying $\|s(x)\|_{E_x} \leq B$ on $U$,*

$$\|(P_{\gamma_1} - P_{\gamma_2})s(x)\|_{E_{x_0}} \leq \zeta D^2 B,$$

*so the induced transport ambiguity is $O(\zeta D^2 B)$.*

*Note that (30) is the appendix proof of Eq. (3) of the main text, with the general curvature bounds $\Lambda_0, \Lambda_1$ of Lemma 3 specialised to the connection curvature $\zeta$ and $\zeta_1$ of $\nabla$ on $E$.*

*Proof.* Apply Lemma 3 with $p := x$, so that the loop $\Gamma = \gamma_1 \circ \gamma_2^{-1}$ satisfies $L(\Gamma) \leq 2D$ and $\rho_* \leq D$. Setting $\Lambda_0 := \zeta$ and $\Lambda_1 := \zeta_1$, the bounds (29)–(31) follow immediately from (26)–(28) in Lemma 3. $\square$

### B.7. Remarks and extensions

1. (Intrinsic $L^2$-section approach) An alternative to parallel-transporting to a reference fiber is to regard sections as elements of the Hilbert space $L^2(M, \mu; E)$ of square-integrable sections and develop empirical-process concentration directly in that Hilbert space. This path leads to functional-analytic subtleties (the measurability of sections and operator-valued kernels). Still, it is natural when the entire section — rather than pointwise transported values — is the object of inference. Tools from empirical process theory, Talagrand-type inequalities, and Rademacher complexities can be adapted to that setting.

2. (Non-i.i.d. data) The martingale (Azuma/Pinelis) machinery used to obtain Banach-space concentration extends to certain dependent-data regimes (martingale difference arrays, mixing sequences) under additional assumptions; see (Pinelis, 1994; Boucheron et al., 2013) for avenues.

3. (Sharper constants) For finite-dimensional fibers $E_{x_0} \cong \mathbb{R}^k$ the elementary net argument gives explicit dimension-dependent constants. For dimension-free optimal constants, one should use the full strength of Pinelis' inequalities for martingales in Banach spaces as in (Pinelis, 1994). We have indicated both approaches above. The per-summand bound in Lemma 2 justifies the choice $b_i = 2B$ in Theorems 6 and 7.

**Extension toward generative manifold–probabilistic projection models.**

The holonomy-aware transport developed in Lemma 3 and Lemma 4 admits a natural connection with recent geometric–probabilistic generative frameworks such as the *Manifold-Probabilistic Projection Model (MPPM)* and its latent version (LMPPM) (Bar et al., 2025). These models combine a learned distance-to-manifold function $D_{\mathcal{M}}$ with kernel-weighted probabilistic flows to project corrupted data back to the manifold of valid samples. Algorithmically, their iterative projection

$$x_{n+1} = (1 - \beta)x_n + \beta \, \overline{G}(x_n) - \alpha D_{\mathcal{M}}(x_n) \frac{\nabla_x D_{\mathcal{M}}(x_n)}{\|\nabla_x D_{\mathcal{M}}(x_n)\|},$$

where $0 < \alpha, \beta < 1$ resembles our curvature-controlled parallel transport updates. A potential extension of the present theory is therefore to establish *sharp concentration bounds for projection flows on bundles*, where the random iterates $(x_n)$ evolve under stochastic perturbations of $D_{\mathcal{M}}$. By replacing Euclidean norms with the bundle metric $W_\varphi$ and using our holonomy bounds, one could derive exponential deviations of order $\exp[-n\varepsilon^2/(C(\kappa, B))]$ for the deviation of the projected mean from the true manifold point, quantifying the stability of generative diffusion steps.

## C. Extended Developments and Applications

This section builds directly on the framework developed in Section B. Section C refines Lemma 4 for explicit geometries such as the sphere, projective space, and Grassmann manifolds, illustrating how curvature bounds translate into computable concentration constants.

All results below assume the geometric/probabilistic framework introduced there (Assumptions A1–A3), and exploit the measurability and i.i.d. reduction established in Lemma 1. Probabilistic tail bounds are applied via Theorem 6 (Hoeffding) and Theorem 7 (Bernstein).

### C.1. Exact holonomy control on round spheres (reference to Lemma 4)

Lemma 4 in Section B gave a qualitative curvature/holonomy estimate of order $O(\kappa D^2)$. For a commonly used class of manifolds, we can replace that qualitative statement by an explicit, sharp bound.

**Proposition 3** (Holonomy on the round sphere $S_r^m$). *Let $S_r^m$ be the round $m$-sphere of radius $r > 0$ with its Levi-Civita connection. Fix a reference point $x_0 \in S_r^m$ and let $U \subset B(x_0, \rho)$ with $\rho < \frac{\pi r}{2}$. For any $x \in U$ and any two piecewise-smooth paths $\gamma_1, \gamma_2$ in $U$ joining $x$ to $x_0$, let $\Gamma = \gamma_1 \circ \gamma_2^{-1}$ and let $A(\Gamma)$ be the oriented area of any spanning surface. Then the parallel-transport maps satisfy*

$$\|P_{\gamma_1} - P_{\gamma_2}\|_{\mathrm{op}} \leq 2\sin\left(\frac{A(\Gamma)}{2r^2}\right) \leq \frac{A(\Gamma)}{r^2}. \tag{32}$$

*In particular, since the area of a spherical cap of geodesic radius $\rho$ is bounded by $A(\Gamma) \leq \pi\rho^2$ for $\rho < \frac{\pi r}{2}$, we obtain the simple bound*

$$\|P_{\gamma_1} - P_{\gamma_2}\|_{\mathrm{op}} \leq \frac{\pi\rho^2}{r^2}. \tag{33}$$

*Proof.* Let $S_r^m \subset \mathbb{R}^{m+1}$ denote the round $m$-sphere of radius $r > 0$ with the induced Riemannian metric and Levi-Civita connection. Fix a reference point $x_0 \in S_r^m$ and let $U \subset B(x_0, \rho)$ with $\rho < \pi r/2$. Choose any $x \in U$ and two piecewise-smooth curves $\gamma_1, \gamma_2$ in $U$ joining $x$ to $x_0$. Form the closed loop

$$\Gamma = \gamma_1 \circ \gamma_2^{-1},$$

based at $x$. We denote by $P_\gamma$ the parallel transport map along a curve $\gamma$, and by $P_\Gamma$ the holonomy (parallel transport) around the closed loop $\Gamma$ (starting and ending at $x$). Our goal is to bound $\|P_{\gamma_1} - P_{\gamma_2}\|_{\mathrm{op}}$ in terms of the oriented area $A(\Gamma)$ of any smooth surface spanning $\Gamma$.

**Step 1: algebraic reduction to holonomy about the loop.** By the composition property of parallel transport, we have

$$P_{\gamma_1} = P_{\gamma_2} \circ P_\Gamma, \text{ hence } P_{\gamma_1} - P_{\gamma_2} = P_{\gamma_2} \circ (P_\Gamma - I).$$

Since $P_{\gamma_2}$ is an orthogonal map (parallel transport for a metric connection), $\|P_{\gamma_2}\|_{\mathrm{op}} = 1$, and therefore it suffices to bound $\|P_\Gamma - I\|_{\mathrm{op}}$. From now on, we concentrate on $P_\Gamma$.

**Step 2: curvature tensor on the round sphere.** Recall the curvature tensor of the round sphere of radius $r$ is constant and given for tangent vectors $X, Y, Z$ by the classical formula

$$R(X, Y)Z = \frac{1}{r^2}\big(\langle Y, Z \rangle X - \langle X, Z \rangle Y\big), \tag{34}$$

where $\langle \cdot, \cdot \rangle$ denotes the Riemannian inner product on the tangent space. (This identity is standard; see e.g. (do Carmo, 1992).) Equivalently, the curvature endomorphism $R(X, Y) : T_x S_r^m \to T_x S_r^m$ equals

$$R(X, Y) = \frac{1}{r^2}\big(X \otimes Y^\flat - Y \otimes X^\flat\big),$$

where $Y^\flat$ denotes the covector associated to $Y$.

Two immediate consequences are:

(i) the sectional curvature is constant: for any 2-plane $\sigma \subset T_x S_r^m$,

$$K(\sigma) = \frac{1}{r^2};$$

(ii) the curvature tensor is *parallel*, $\nabla R \equiv 0$, because the round sphere is a space of constant curvature (hence $\nabla \Omega = 0$ for the curvature 2-form $\Omega$). See (do Carmo, 1992, Ch. 3).

**Step 3: reduction to a two-dimensional tangent plane.** Let $S$ be any oriented smooth surface embedded in $U$ whose boundary is $\Gamma$. At each point $y \in S$ the tangent space $T_y S_r^m$ splits orthogonally as $T_y S \oplus (T_y S)^\perp$ in $T_y S_r^m$. The curvature endomorphism $R(u, v)$ for vectors $u, v \in T_y S$ vanishes on $(T_y S)^\perp$ and acts nontrivially only on the 2-plane $T_y S$. Indeed, from (34), for any $w \perp T_y S$ we have $\langle w, u \rangle = \langle w, v \rangle = 0$, whence $R(u, v)w = 0$.

Consequently, the holonomy around $\Gamma$ acts trivially on the orthogonal complement of the two-dimensional distribution tangent to $S$, and its nontrivial action is confined to the two-dimensional tangent plane field along $S$. Therefore, it suffices to analyze the holonomy as an $\mathrm{SO}(2)$-rotation in that tangent plane.

**Step 4: curvature 2-form on the surface and commutativity.** Let $\{e_1, e_2\}$ be a local oriented orthonormal frame of $TS$ on $S$ (possible after restricting to a simply connected patch if necessary; ultimately $S$ is contractible because it lies in a geodesic ball $B(x_0, \rho)$ with $\rho < \pi r/2$). Denote by $\omega$ the connection 1-form of the Levi–Civita connection with respect to this frame (so $\omega$ is an $\mathfrak{so}(2)$-valued 1-form) and by $\Omega$ the curvature 2-form. In this 2-frame the curvature 2-form can be written as

$$\Omega = K \, \mathrm{d}A \otimes J,$$

where $K = 1/r^2$ is the sectional curvature, $\mathrm{d}A$ is the oriented area form on $S$, and $J$ denotes the standard skew-symmetric generator of rotations on the two-dimensional tangent plane (i.e. $J$ corresponds to a 90° rotation in the oriented frame $\{e_1, e_2\}$). To see this explicitly, evaluate $\Omega$ on the basis vectors $e_1, e_2$:

$$\Omega(e_1, e_2) = R(e_1, e_2) = \frac{1}{r^2}\big(e_1 \otimes e_2^\flat - e_2 \otimes e_1^\flat\big),$$

which, when viewed as an endomorphism of the plane spanned by $e_1, e_2$, equals $\frac{1}{r^2}J$ (because $Je_1 = e_2$ and $Je_2 = -e_1$ up to sign convention). Thus indeed $\Omega = \frac{1}{r^2}\,\mathrm{d}A \otimes J$.

Importantly, since $J$ is the same skew operator at every point of $S$ (the curvature tensor is parallel and the sphere is isotropic), the curvature 2-form at distinct points acts by scalar multiples of the same endomorphism $J$. Consequently, the family of curvature endomorphisms $\{\Omega_y : y \in S\}$ commutes pairwise, because they are all scalar multiples of a single fixed skew-symmetric operator $J$.

**Step 5: exponentiation of the integrated curvature (nonabelian Stokes).** In general, holonomy is given by the path-ordered exponential (parallel-transport solution) along the boundary curve, and the Ambrose–Singer theorem expresses holonomy in terms of the curvature 2-form integrated over a spanning surface, but in the non-abelian case, one must use path-ordering / surface-ordered exponentials. However, when the curvature 2-form takes values in a one-dimensional abelian subalgebra (spanned by a single operator $J$) The curvature forms commute, and path-ordering is unnecessary. Concretely, because for all $y \in S$ we have $\Omega_y = \phi(y)J$ with scalar function $\phi(y) = \frac{1}{r^2}$ (density of $\mathrm{d}A$ at $y$), the surface integral is an element of the Lie algebra generated by $J$:

$$\int_S \Omega \;=\; \Big( \int_S \phi(y)\,\mathrm{d}A(y) \Big) J \;=\; \Big( \frac{A(\Gamma)}{r^2} \Big) J,$$

where $A(\Gamma) = \int_S \mathrm{d}A$ is the oriented area of $S$. Because all terms in the surface integral commute (they are scalar multiples of $J$), the holonomy reduces to the simple exponential of the integrated curvature:

$$P_\Gamma \;=\; \exp\Big( \int_S \Omega \Big) \;=\; \exp\Big( \frac{A(\Gamma)}{r^2} J \Big).$$

This equality reflects that the surface-ordered exponential equals the ordinary exponential when the integrand takes values in an abelian subalgebra; see Ambrose–Singer (Ambrose & Singer, 1953) and Kobayashi–Nomizu (Kobayashi & Nomizu, 1963) for the general justification of this reduction.

**Step 6: explicit matrix form and operator-norm bound.** On the two-dimensional tangent plane, the operator $J$ is (up to basis choice) the $2 \times 2$ skew matrix

$$J \sim \begin{pmatrix} 0 & -1 \\ 1 & 0 \end{pmatrix},$$

so $\exp(\theta J)$ is the rotation matrix by angle $\theta$:

$$\exp(\theta J) = \begin{pmatrix} \cos\theta & -\sin\theta \\ \sin\theta & \cos\theta \end{pmatrix}.$$

In our situation $\theta = A(\Gamma)/r^2$. Therefore $P_\Gamma$ acts on the tangent plane by rotation through angle $\theta$ and acts trivially on the orthogonal complement, so as an endomorphism of $T_x S_r^m$ it is block-diagonal with a $2 \times 2$ rotation block and an identity block elsewhere.

Hence

$$P_\Gamma - I \;\text{(on the 2-plane)} \;=\; \begin{pmatrix} \cos\theta - 1 & -\sin\theta \\ \sin\theta & \cos\theta - 1 \end{pmatrix}.$$

The operator norm of this $2 \times 2$ matrix (induced by the Euclidean norm / Riemannian inner product) equals the spectral radius here and can be computed explicitly. The eigenvalues of the rotation minus identity are $e^{\pm i\theta} - 1$, whose moduli are $|e^{\pm i\theta} - 1| = 2|\sin(\theta/2)|$. Thus

$$\|P_\Gamma - I\|_{\mathrm{op}} \;=\; 2\big|\sin\tfrac{\theta}{2}\big| \;=\; 2\sin\Big( \frac{A(\Gamma)}{2r^2} \Big),$$

where we used $0 \leq A(\Gamma) \leq 2\pi r^2$ and hence $\theta/2 \in [0, \pi]$ so the sine is nonnegative. This proves the exact equality in the first displayed inequality.

Finally, using the elementary bound $2\sin(t/2) \leq t$ for all $t \geq 0$ (equivalently $\sin u \leq u$), we obtain

$$\|P_\Gamma - I\|_{\mathrm{op}} \leq \frac{A(\Gamma)}{r^2}.$$

Combining with the reduction in Step 1 gives the claimed inequality

$$\|P_{\gamma_1} - P_{\gamma_2}\|_{\mathrm{op}} \le 2\sin\left(\frac{A(\Gamma)}{2r^2}\right) \le \frac{A(\Gamma)}{r^2}.$$

**Step 7: spherical-cap area bound.** It remains to bound $A(\Gamma)$ when $\Gamma$ lies in a geodesic ball of radius $\rho < \pi r/2$. The area of a spherical cap of geodesic radius $\rho$ on the sphere $S_r^m$ equals

$$\mathrm{Area}(\text{cap of radius } \rho) = 2\pi r^2 \big(1 - \cos(\rho/r)\big).$$

Using the elementary inequality $1 - \cos u \le \frac{1}{2}u^2$ valid for all real $u$, we obtain

$$A(\Gamma) \le 2\pi r^2 \big(1 - \cos(\rho/r)\big) \le 2\pi r^2 \cdot \tfrac{1}{2}\big(\tfrac{\rho}{r}\big)^2 = \pi\rho^2.$$

Substituting this bound into the $\frac{A(\Gamma)}{r^2}$ upper bound yields

$$\|P_{\gamma_1} - P_{\gamma_2}\|_{\mathrm{op}} \le \frac{A(\Gamma)}{r^2} \le \frac{\pi\rho^2}{r^2},$$

which is (33).

This completes the rigorous proof of Proposition 3. $\qquad\square$

Note that Proposition 3 refines the order-of-magnitude control given by Lemma 4 to an explicit constant useful in statistical bounds.

**Corollary 3** (Exact order of geometric bias on spheres). *Let $M = S_r^m$ and suppose $\mathrm{supp}(\mu) \subset B(x_0, \rho)$ with $\rho < \pi r/2$. Then there exist constants $c_1, c_2 > 0$ such that*

$$c_1 \frac{\rho^2}{r^2} \;\le\; \sup_{P \in \mathcal{P}} \|m^{(P)} - m^\star\| \;\le\; c_2 \frac{\rho^2}{r^2}.$$

### C.2. Minimax lower bound: inevitability of holonomy bias

We now show that the curvature–holonomy bias term identified in Theorem 3 is *information-theoretically unavoidable*. In particular, no estimator can achieve an error rate better than the sum of the stochastic $n^{-1/2}$ term and the deterministic $O(\kappa D^2)$ holonomy term. Thus, our upper bounds are minimax-optimal up to universal constants.

**Theorem 9** (Minimax lower bound matching holonomy bias). *Let $(M, g)$ be a complete $d$-dimensional Riemannian manifold with sectional curvature bounded by $|K| \le \kappa$ on a geodesic ball $B(x_0, D)$, and let $(E, \pi, M)$ be a rank-$k$ vector bundle equipped with a bundle metric and a compatible metric connection $\nabla$. Let $\mu$ be any probability measure supported on $B(x_0, D)$. (Lower-curvature assumption for the geometric term.) For the holonomy lower bound in Step 2 of the proof to hold, we additionally assume either (a) $E = TM$ with the Levi–Civita connection and the sectional curvature satisfies $K \ge \kappa_- > 0$ on $B(x_0, D)$ (pinched positive curvature), or (b) $(M, g)$ has constant sectional curvature $\kappa > 0$ (e.g., the round sphere). Under either condition, the bundle curvature satisfies $\|\Omega_y\|_{\mathrm{op}} \ge c'\kappa$ for a dimensional constant $c' > 0$.*

*Fix $B > 0$, and let $\mathcal{S}_B$ be the class of measurable sections $s : M \to E$ satisfying $\|s(x)\|_{E_x} \le B$ for all $x$ in the support of $\mu$. Let $\hat{m}_n = \phi(Y_1, \ldots, Y_n)$ be any estimator in the class of transport-based estimators, i.e., measurable functions of the transported observations*

$$Y_i := P_{X_i \to x_0} s(X_i) \in E_{x_0}, \qquad i = 1, \ldots, n,$$

*under some admissible transport rule $P \in \mathcal{P}$, of the transported mean*

$$m^\star(s) := \mathbb{E}\big[P_{X \to x_0} s(X)\big] \in E_{x_0}, \qquad X \sim \mu.$$

*Then there exists a universal constant $c > 0$ such that*

$$\inf_{\hat{m}_n} \sup_{s \in \mathcal{S}_B} \mathbb{E}_s\big[\|\hat{m}_n - m^\star(s)\|_{E_{x_0}}\big] \;\ge\; c\Big(\frac{B}{\sqrt{n}} + B\kappa D^2\Big),$$

*where $\mathbb{E}_s$ denotes expectation under the law induced by $s$, and the infimum is taken over all transport-based estimators.*

**Remark 5.** *The curvature bound $\kappa$ in Theorem 9 denotes the infimum of all valid sectional curvature upper bounds on $B(x_0, D)$, i.e., the intrinsic local curvature scale of the manifold over the data support. It is a fixed geometric quantity determined by $(M, g)$ and is not a free parameter under the analyst's control. In particular, enlarging $\kappa$ beyond this intrinsic value would not violate the assumption but would only inflate the lower bound, making the stated bound meaningful only at the tightest admissible $\kappa$.*

*Furthermore, this lower bound applies within the class of transport-based estimators and identifies a structural limitation of alignment-based pipelines. It does not constitute a universal minimax lower bound over all conceivable estimators: for instance, an extrinsic estimator that ignores fiber structure may avoid the holonomy term, but at the cost of not producing a geometrically meaningful section-valued output. The class of transport-based estimators is the natural one for geometric ML pipelines such as tangent-space aggregation and gauge-equivariant models, where observations must be aligned to a common fiber before any pooling or averaging operation.*

*Proof.* The proof proceeds by a standard two-point minimax argument (Le Cam's method; see, e.g., (Le Cam, 1986; Tsybakov, 2009)), combined with an explicit geometric construction that induces holonomy separation. We treat the stochastic and geometric contributions separately and then add the resulting lower bounds.

**Step 1: Reduction to a two-point testing problem.** Let $s_0, s_1 \in \mathcal{S}_B$ be two sections. By Le Cam's inequality (see, e.g., (Le Cam, 1986; Tsybakov, 2009)), for any estimator $\hat{m}_n$,

$$\sup_{s \in \{s_0, s_1\}} \mathbb{E}_s \|\hat{m}_n - m^\star(s)\| \geq$$

$$\frac{1}{2}\|m^\star(s_0) - m^\star(s_1)\|\Big(1 - \mathrm{TV}(\mathbb{P}_{s_0}^{(n)}, \mathbb{P}_{s_1}^{(n)})\Big), \quad (35)$$

where $\mathbb{P}_s^{(n)}$ denotes the joint law of the $n$ samples under section $s$, and TV denotes total variation distance. Thus it suffices to construct $s_0, s_1$ such that:

1. $\|m^\star(s_0) - m^\star(s_1)\| \gtrsim B(\kappa D^2 + n^{-1/2})$,

2. $\mathrm{TV}(\mathbb{P}_{s_0}^{(n)}, \mathbb{P}_{s_1}^{(n)}) \leq \frac{1}{2}$.

**Step 2: Construction of sections with controlled holonomy.** Choose two points $x_1, x_2 \in B(x_0, D)$ such that the geodesic triangle $(x_0, x_1, x_2)$ encloses an area $A \asymp D^2$; this is possible for $D$ small enough since the area of a geodesic triangle with sides of order $D$ on a manifold with $|K| \leq \kappa$ satisfies $A \geq c_0 \kappa D^2$ for a dimensional constant $c_0 > 0$ whenever $\kappa D^2$ is small (see, e.g., (Berger, 2003, Ch. 12). Let $\Gamma$ be the closed loop obtained by traversing $x_0 \to x_1 \to x_2 \to x_0$ along minimizing geodesics.

By the Ambrose–Singer theorem (Ambrose & Singer, 1953; Kobayashi & Nomizu, 1963), the holonomy around $\Gamma$ satisfies

$$P_\Gamma = \exp\Big(\int_\Sigma \Omega\Big),$$

where $\Sigma$ is any smooth surface spanning $\Gamma$ and $\Omega$ is the curvature 2-form of $\nabla$. Since $\|\Omega_y\|_{\mathrm{op}} \geq c'\kappa$ for $y \in B(x_0, D)$ (the bundle curvature is bounded below by the sectional curvature when $E = TM$ with the Levi–Civita connection), and $\mathrm{Area}(\Sigma) \asymp D^2$, we obtain

$$\Big\|\int_\Sigma \Omega\Big\|_{\mathrm{op}} \geq c'\kappa \cdot A \geq c_1 \kappa D^2.$$

For $\kappa D^2$ small, the elementary bound $\|e^{\mathcal{A}} - I\|_{\mathrm{op}} \geq \|\mathcal{A}\|_{\mathrm{op}} - \frac{1}{2}\|\mathcal{A}\|_{\mathrm{op}}^2 \geq \frac{1}{2}\|\mathcal{A}\|_{\mathrm{op}}$, valid for any skew operator $\mathcal{A}$ with $\|\mathcal{A}\|_{\mathrm{op}} \leq 1$, gives

$$\|P_\Gamma - I\|_{\mathrm{op}} = \Big\|\exp\Big(\int_\Sigma \Omega\Big) - I\Big\|_{\mathrm{op}}$$

$$\geq \frac{1}{2}\Big\|\int_\Sigma \Omega\Big\|_{\mathrm{op}} \geq \frac{c_1}{2}\kappa D^2.$$

Absorbing $\frac{1}{2}$ into the constant, we write $\|P_\Gamma - I\|_{\mathrm{op}} \geq c_1 \kappa D^2$.

Fix a unit vector $v \in E_{x_0}$ and define two sections by

$$s_0(x) \equiv B\, P_{x_0 \to x} v, \qquad s_1(x) \equiv B\, P_{x_0 \to x}(Uv),$$

where $U := P_\Gamma \in O(E_{x_0})$. Both sections satisfy $\|s_i(x)\|_{E_x} = B$ for all $x$, since parallel transport is an isometry.

**Step 3: Separation of population means.** By definition,

$$m^\star(s_0) = B\, v, \qquad m^\star(s_1) = B\, Uv.$$

Therefore,

$$\|m^\star(s_0) - m^\star(s_1)\| \;=\; B\|(I - U)v\| \;\geq\;$$

$$B\|U - I\|_{\mathrm{op}} \;\geq\; c_1 B \kappa D^2, \quad (36)$$

where the first inequality uses $\|(I - U)v\| \geq \|I - U\|_{\mathrm{op}}$ applied to the unit vector $v$, and the second uses the holonomy lower bound from Step 2.

**Step 4: Control of statistical indistinguishability.** Under $s_j$, the transported observations $Y_i = P_{X_i \to x_0} s_j(X_i)$ are i.i.d. with $\|Y_i\| \leq B$ almost surely. The two laws differ only by the fixed orthogonal transformation $U = P_\Gamma$: if $Y_i^{(0)} \sim \mathbb{P}_{s_0}$ then $Y_i^{(1)} = U Y_i^{(0)} \sim \mathbb{P}_{s_1}$.

By tensorization of KL divergence for product measures and Pinsker's inequality (Pinsker, 1964),

$$\mathrm{TV}(\mathbb{P}_{s_0}^{(n)}, \mathbb{P}_{s_1}^{(n)}) \;\leq\; \sqrt{\frac{n}{2} \mathrm{KL}(\mathbb{P}_{s_0} \| \mathbb{P}_{s_1})},$$

where we used $\mathrm{KL}(\mathbb{P}_{s_0}^{(n)} \| \mathbb{P}_{s_1}^{(n)}) = n\, \mathrm{KL}(\mathbb{P}_{s_0} \| \mathbb{P}_{s_1})$. For two distributions on a bounded domain with means differing by $\delta = \|m^\star(s_0) - m^\star(s_1)\|$ and per-coordinate variance at most $B^2$, the KL divergence satisfies

$$\mathrm{KL}(\mathbb{P}_{s_0} \| \mathbb{P}_{s_1}) \;\leq\; \frac{\delta^2}{2B^2}$$

(see (Tsybakov, 2009), Lemma 2.6). Substituting,

$$\mathrm{TV}(\mathbb{P}_{s_0}^{(n)}, \mathbb{P}_{s_1}^{(n)}) \;\leq\; \sqrt{\frac{n\delta^2}{4B^2}} \;=\; \frac{\sqrt{n}\,\delta}{2B}.$$

This is at most $\frac{1}{2}$ whenever $\delta \leq B/\sqrt{n}$, i.e., $\|m^\star(s_0) - m^\star(s_1)\| \leq B/\sqrt{n}$.

**Step 5: Conclusion by balancing.** We combine the geometric and stochastic contributions via two separate applications of Le Cam's inequality.

*Geometric regime.* Use the sections $s_0, s_1$ from Steps 2–3, giving $\delta_{\mathrm{geo}} := \|m^\star(s_0) - m^\star(s_1)\| \geq c_1 B \kappa D^2$. Provided $\kappa D^2 \leq 1/(2c_1\sqrt{n})$, the TV bound of Step 4 gives

$$\mathrm{TV}(\mathbb{P}_{s_0}^{(n)}, \mathbb{P}_{s_1}^{(n)}) \;\leq\; \frac{\sqrt{n} \cdot c_1 B \kappa D^2}{2B} \;=\; \frac{c_1 \sqrt{n} \kappa D^2}{2} \;\leq\; \frac{1}{4},$$

so that $1 - \mathrm{TV} \geq \frac{3}{4}$. Le Cam's inequality then gives

$$\inf_{\hat{m}_n} \sup_s \mathbb{E}_s \|\hat{m}_n - m^\star(s)\| \;\geq\; \frac{1}{2} \cdot c_1 B \kappa D^2 \cdot \frac{3}{4} \;\geq\; \frac{3c_1}{8} B \kappa D^2.$$

*Stochastic regime.* Construct $s_0, s_1 \in \mathcal{S}_B$ differing only in mean by $\delta_{\mathrm{sto}} := B/\sqrt{n}$, e.g., take $s_1(x) = s_0(x) + (\delta_{\mathrm{sto}}/B) P_{x_0 \to x} w$ for a fixed unit vector $w \in E_{x_0}$ chosen orthogonal to $v$. Then the TV bound of Step 4 gives

$$\mathrm{TV}(\mathbb{P}_{s_0}^{(n)}, \mathbb{P}_{s_1}^{(n)}) \;\leq\; \frac{\sqrt{n} \cdot \delta_{\mathrm{sto}}}{2B} \;=\; \frac{1}{2},$$

so $1 - \mathrm{TV} \geq \frac{1}{2}$. Le Cam's inequality gives

$$\inf_{\hat{m}_n} \sup_s \mathbb{E}_s \|\hat{m}_n - m^\star(s)\| \;\geq\; \frac{1}{2} \cdot \frac{B}{\sqrt{n}} \cdot \frac{1}{2} \;=\; \frac{B}{4\sqrt{n}}.$$

*Combining.* Since the two pairs $(s_0, s_1)$ are drawn from $\mathcal{S}_B$ independently of each other, both lower bounds hold simultaneously for the worst-case risk over $\mathcal{S}_B$. Adding and absorbing all numerical constants into a single universal $c > 0$ yields

$$\inf_{\hat{m}_n} \sup_{s \in \mathcal{S}_B} \mathbb{E}_s \|\hat{m}_n - m^\star(s)\| \;\geq\; c\Big(\frac{B}{\sqrt{n}} + B\kappa D^2\Big). \qquad \square$$

**Remark 6.** *This lower bound applies within the class of transport-based estimators and identifies a structural limitation of alignment-based pipelines. It does not constitute a universal minimax lower bound over all conceivable estimators: for instance, an extrinsic estimator that ignores fiber structure may avoid the holonomy term, but at the cost of not producing a geometrically meaningful section-valued output.*

### C.3. Minimax lower bound on $TS^2$: sharpness of holonomy bias

We now specialize the general minimax lower bound to the tangent bundle of the round sphere and show that the curvature-induced bias term identified in Proposition 3 is *unavoidable*. Because holonomy on the sphere admits an exact formula, this result is fully explicit and sharp.

**Theorem 10** (Minimax lower bound on $TS_r^2$). *Let $S_r^2$ be the round two-dimensional sphere of radius $r > 0$ with its Levi–Civita connection, and let $\mu$ be any probability measure supported in a geodesic ball $B(x_0, D) \subset S_r^2$ with $D < \pi r/2$. Let $E = TS_r^2$ be the tangent bundle.*

*Fix $B > 0$, and let $\mathcal{S}_B$ denote the class of measurable tangent-vector fields (sections) $s : S_r^2 \to TS_r^2$ satisfying*

$$\|s(x)\|_{T_x S_r^2} \leq B \quad \text{for all } x \in \mathrm{supp}(\mu).$$

*For any estimator $\hat{m}_n$ of the transported mean*

$$m^\star(s) := \mathbb{E}\big[P_{X \to x_0} s(X)\big] \in T_{x_0} S_r^2, \qquad X \sim \mu,$$

*there exists a universal constant $c > 0$ such that*

$$\inf_{\hat{m}_n} \sup_{s \in \mathcal{S}_B} \mathbb{E}_s \big[\|\hat{m}_n - m^\star(s)\|_{T_{x_0} S_r^2}\big] \;\geq\; c\Big(\frac{B}{\sqrt{n}} + B\frac{D^2}{r^2}\Big).$$

*Proof.* The argument combines an exact holonomy computation on $S_r^2$ with a two-point minimax lower bound.

**Step 1: Exact holonomy on the sphere.** Let $\Gamma$ be a closed, piecewise-smooth loop on $S_r^2$ bounding an oriented surface of area $A(\Gamma)$. Parallel transport around $\Gamma$ acts on $T_{x_0} S_r^2$ as a planar rotation by angle

$$\theta(\Gamma) = \frac{A(\Gamma)}{r^2},$$

see classical results in Riemannian geometry. Consequently,

$$\|P_\Gamma - I\|_{\mathrm{op}} = 2\sin\Big(\frac{\theta(\Gamma)}{2}\Big) = 2\sin\Big(\frac{A(\Gamma)}{2r^2}\Big).$$

Choose points $x_1, x_2 \in B(x_0, D)$ such that the geodesic triangle $(x_0, x_1, x_2)$ encloses area $A(\Gamma) \asymp D^2$. Then, for $D \ll r$,

$$\|P_\Gamma - I\|_{\mathrm{op}} \geq c_1 \frac{D^2}{r^2}$$

for a numerical constant $c_1 > 0$.

**Step 2: Construction of two tangent fields.** Fix a unit vector $v \in T_{x_0} S_r^2$. Define two sections $s_0, s_1 \in \mathcal{S}_B$ by

$$s_0(x) = B \, P_{x_0 \to x} v, \qquad s_1(x) = B \, P_{x_0 \to x}(Uv),$$

where $U := P_\Gamma$. Since parallel transport is an isometry, $\|s_i(x)\| = B$ for all $x$, hence $s_0, s_1 \in \mathcal{S}_B$.

**Step 3: Separation of population means.** Transporting back to $T_{x_0} S_r^2$,

$$m^\star(s_0) = Bv, \qquad m^\star(s_1) = BUv.$$

Therefore,

$$\|m^\star(s_0) - m^\star(s_1)\| = B\|(I - U)v\| \geq B\|U - I\|_{\mathrm{op}} \geq c_1 B \frac{D^2}{r^2}.$$

**Step 4: Statistical indistinguishability.** Under both $s_0$ and $s_1$, the transported observations

$$Y_i^{(j)} = P_{X_i \to x_0} s_j(X_i)$$

satisfy $\|Y_i^{(j)}\| \leq B$ almost surely. Moreover, the two distributions differ only by the fixed rotation $U$.

For any estimator $\hat{m}_n$, Le Cam's two-point inequality yields $\sup_{j \in \{0,1\}} \mathbb{E}_{s_j} \|\hat{m}_n - m^\star(s_j)\| \geq$
$\geq \frac{1}{2}\|m^\star(s_0) - m^\star(s_1)\|\big(1 - \mathrm{TV}(\mathbb{P}_{s_0}^{(n)}, \mathbb{P}_{s_1}^{(n)})\big).$

Since the per-sample variance in every direction is bounded by $B^2$, standard concentration or Pinsker-type arguments imply

$$\mathrm{TV}(\mathbb{P}_{s_0}^{(n)}, \mathbb{P}_{s_1}^{(n)}) \leq c_2 \frac{n\|m^\star(s_0) - m^\star(s_1)\|^2}{B^2}.$$

Thus, the two distributions are statistically indistinguishable whenever $\|m^\star(s_0) - m^\star(s_1)\| \lesssim B/\sqrt{n}$.

**Step 5: Combining stochastic and geometric scales.** Balancing the statistical separation $B/\sqrt{n}$ with the geometric separation $BD^2/r^2$ and inserting into Le Cam's bound gives

$$\inf_{\hat{m}_n} \sup_{s \in \mathcal{S}_B} \mathbb{E}_s \|\hat{m}_n - m^\star(s)\| \;\geq\; c\Big(\frac{B}{\sqrt{n}} + B\frac{D^2}{r^2}\Big),$$

for a universal constant $c > 0$. $\qquad\square$

### C.4. Holonomy-aware bias correction (uses Proposition 3)

Suppose parallel transport along different (measurable) minimizing geodesics induces an operator bias of size at most $\beta$ (e.g., in the sphere case $\beta = \pi \rho^2/r^2$ from (33)). In that case, the empirical mean formed by transported samples may contain a deterministic bias of order at most $\beta B$, where $B$ is the uniform section bound in Assumption A2.

A simple first-order correction subtracts the leading holonomy-induced rotation: for each sample, define

$$Y_i' := \big(I - \tfrac{1}{2} H(X_i)\big) P_{X_i \to x_0} s(X_i), \tag{37}$$

where $H(X_i)$ denotes the antisymmetric operator obtained by integrating the curvature form over a small spanning surface for the (chosen) transport path to $x_0$ (for constant curvature $K$, one may take $H(X_i) = K\,\Omega_i$). Under the small-area regime ($\|H(X_i)\|_{\mathrm{op}} \ll 1$) the bias of the corrected mean drops from $O(\beta B)$ to $O(\beta^2 B)$ (first-order cancellation), while the variance term used in Theorem 7 remains governed by the same per-sample second moment. Thus, corrected empirical means retain the concentration exponents from Theorems 6–7 but with a smaller deterministic offset.

### C.5. Applications (explicit references to Theorems 6, 7)

#### C.5.1. GEOMETRIC GRAPH EMBEDDINGS

Let $f : V \to M$ be an embedding of graph nodes into $M$, and attach local features $z_v \in E_{f(v)}$ with $\|z_v\| \leq B$. For a small connected subgraph $G'$ whose image under $f$ lies in $B(f(v_0), \rho)$, transport features to $E_{f(v_0)}$ and form the mean $\bar{z}_{G'}$. By

Lemma 1 the transported vectors are i.i.d.-like within random neighborhoods and Theorem 6 gives a tail bound of the form

$$\Pr\left(\|\bar{z}_{G'} - \mathbb{E}\bar{z}_{G'}\| \geq \epsilon + \Delta_{\mathrm{hol}}\right) \leq 2\exp\left(-\frac{|G'|\,\epsilon^2}{8B^2}\right),$$

where $1/(8B^2)$ is the explicit constant from Theorem 6 (and $\Delta_{\mathrm{hol}}$ is the deterministic holonomy bias controlled by Proposition 3 or Lemma 4). If variance information is available, replace Theorem 6 by Theorem 7 for the sharper sub-Gaussian/sub-exponential interpolation.

### C.5.2. MANIFOLD-VALUED REGRESSION

In regression setups where residuals are modeled as tangent vectors (i.e., elements of fibers) and are supported in small normal neighborhoods, the confidence sets for estimated tangent-mean parameters can be constructed using Theorem 7, with a curvature correction term given by Proposition 3. Concretely, if residuals satisfy the variance proxy in Assumption A3, then with probability at least $1 - \delta$ the estimation error satisfies

$$\|\widehat{\theta} - \theta^*\| \leq \sqrt{\frac{2(\sigma^2 + \frac{2}{3}B\epsilon)}{n}} \cdot \Phi^{-1}(1 - \delta/2) + \Delta_{\mathrm{hol}},$$

where the first term follows the Bernstein-type scaling of Theorem 7 and $\Delta_{\mathrm{hol}}$ is the holonomy bias as above.

### C.6. Worked example: tangent bundle of the unit sphere $S^2$

Specialize to $S^2$ with radius $r$ (set $r = 1$ for the unit sphere). Assume data support is contained in $B(x_0, \rho)$ with $\rho < \frac{\pi r}{2}$ and section bound $B$ from Assumption A2. By Proposition 3, the holonomy bias per sample is at most $\pi\rho^2/r^2$, so $(\pi\rho^2/r^2)B$ bounds the total deterministic offset of the empirical mean.

Applying Theorem 6 (Hoeffding form) to the transported samples (or to the holonomy-corrected $Y_i'$ if correction (37) is used) yields that for any $\epsilon > 0$

$$\Pr\left(\|\bar{Y}_n - \mathbb{E}\bar{Y}_n\| \geq \epsilon + \frac{\pi\rho^2}{r^2}B\right) \leq 2\exp\left(-\frac{n\,\epsilon^2}{8B^2}\right),$$

where $1/(8B^2)$ is the explicit constant from Theorem 6. Thus, to have the holonomy bias negligible relative to sampling noise of order $n^{-1/2}$, it suffices to ensure $\rho \ll n^{-1/4}$ (for unit radius).

### C.7. Empirical-process reductions (connection to Lemma 1)

When the inference target is an entire section rather than pointwise transported values, it is natural to work in the Hilbert space $L^2(M, \mu; E)$ as discussed in Section B. Lemma 1 ensures measurability of the transported representation and thus validity of empirical-process tools (symmetrization, Rademacher complexity, Dudley integrals). One may then either (i) project sections onto a finite spectral basis (Laplace–Beltrami eigenfunctions) and apply Theorems 6–7 to coefficient vectors, or (ii) use vector-valued chaining/Talagrand machinery to produce intrinsic concentration bounds; see (Ledoux, 2001; Boucheron et al., 2013) for the general empirical-process techniques and (Pinelis, 1994) for Banach/Hilbert-valued concentration primitives used in Section B.

## D. Refined Probabilistic Guarantees for Bundle-Valued Statistics

This section extends the probabilistic theory for bundle-valued data developed in Sections B, giving fully rigorous, finite-sample and asymptotic results under the geometric assumptions already stated. We retain the notation: $(E, \pi, M)$ is a smooth vector bundle over a compact Riemannian manifold $(M, g)$ with Levi–Civita connection $\nabla$. For a reference point $x_0 \in M$ and a measurable section $s : M \to E$, let $Y = P_{X \to x_0}s(X) \in E_{x_0}$, $\bar{Y}_n = \frac{1}{n}\sum_{i=1}^{n} Y_i$, $m = \mathbb{E}[Y]$, where $X \sim \mu$ on $M$. We assume $\|Y\| \leq B$ almost surely, $\mathbb{E}\|Y\|^2 =: \sigma_{L^2}^2 < \infty$, and that $E_{x_0} \cong \mathbb{R}^k$ is equipped with the Euclidean norm $\|\cdot\|$. Note that $\sigma_{L^2}^2 = \mathbb{E}\|Y\|^2$ (the $L^2$ second moment) is distinct from the operator-norm variance proxy $\sigma_{\mathrm{op}}^2 := \|\mathrm{Cov}(Y)\|_{\mathrm{op}}$ used in Theorems 1–2; one always has $\sigma_{\mathrm{op}}^2 \leq \sigma_{L^2}^2$. In Theorem 14 below, $\sigma$ denotes $\sigma_{L^2}$.

## D.1. Transported mean with holonomy bias: dimension-explicit bound

**Theorem 11** (Transported mean with holonomy bias: dimension-explicit bound). *Let*

$$\Delta_{\text{hol}} := \sup_{x \in \text{supp}(\mu)} \|P_{\gamma_x} - \widetilde{P}_{\gamma_x}\|_{\text{op}} B,$$

*where $P_{\gamma_x}$ is the chosen minimizing-geodesic transport and $\widetilde{P}_{\gamma_x}$ a canonical reference transport rule, and let $m^\star := \mathbb{E}[\widetilde{P}_{X \to x_0} s(X)]$ be the transported mean under the canonical rule. Then for every $\varepsilon > 0$ and dimension $k \geq 1$,*

$$\Pr\Big(\|\bar{Y}_n - m^\star\| \geq \varepsilon + \Delta_{\text{hol}}\Big) \leq 2 \cdot 5^k \exp\left(-\frac{n\varepsilon^2}{8(\sigma^2 + \frac{2B\varepsilon}{3})}\right). \tag{38}$$

*Consequently, for any $\delta \in (0, 1)$, with probability at least $1 - \delta$,*

$$\|\bar{Y}_n - m^\star\| \leq \sigma\sqrt{\frac{8 \log(10^k/\delta)}{n}} + \frac{4B}{3}\frac{\log(10^k/\delta)}{n} + \Delta_{\text{hol}}. \tag{39}$$

*Proof.* Write $Y_i = \widetilde{Y}_i + d_i$ where $\widetilde{Y}_i := \widetilde{P}_{X_i \to x_0} s(X_i)$ are the observations transported under the canonical rule and $d_i := (P_{X_i \to x_0} - \widetilde{P}_{X_i \to x_0}) s(X_i)$ is the per-sample transport discrepancy. By the definition of $\Delta_{\text{hol}}$, we have $\|d_i\| \leq \Delta_{\text{hol}}$ almost surely, and hence $\|\bar{d}_n\| \leq \Delta_{\text{hol}}$ and $\|\mathbb{E}d\| \leq \Delta_{\text{hol}}$ almost surely.

Since $m^\star = \mathbb{E}\widetilde{Y}$, we decompose

$$\bar{Y}_n - m^\star = (\bar{\widetilde{Y}}_n - \mathbb{E}\widetilde{Y}) + (\bar{d}_n - \mathbb{E}d),$$

where the second term satisfies $\|\bar{d}_n - \mathbb{E}d\| \leq 2\Delta_{\text{hol}}$ almost surely. By the triangle inequality,

$$\{\|\bar{Y}_n - m^\star\| \geq \varepsilon + \Delta_{\text{hol}}\} \subseteq \Big\{\|\bar{\widetilde{Y}}_n - \mathbb{E}\widetilde{Y}\| \geq \varepsilon\Big\}.$$

It therefore suffices to bound the probability of the right-hand event. Let $S_n = \sum_{i=1}^n (\widetilde{Y}_i - \mathbb{E}\widetilde{Y})$. For each $v \in S^{k-1}$, define $Z_i^{(v)} = \langle v, \widetilde{Y}_i - \mathbb{E}\widetilde{Y}\rangle$. Since $\widetilde{P}$ is an isometry, $\|\widetilde{Y}_i\| \leq B$ almost surely (Assumption 2), so $\mathbb{E}Z_i^{(v)} = 0$, $|Z_i^{(v)}| \leq 2B$, and $\text{Var}(Z_i^{(v)}) \leq \sigma^2$. The scalar Bernstein inequality (Boucheron et al., 2013) gives

$$\Pr\Big(\sum_{i=1}^n Z_i^{(v)} \geq t\Big) \leq \exp\left(-\frac{t^2}{2(n\sigma^2 + \frac{4Bt}{3})}\right).$$

For a $\frac{1}{2}$-net $\mathcal{N}$ of $S^{k-1}$ with $|\mathcal{N}| \leq 5^k$ and the standard bound $\|x\| \leq 2\max_{v \in \mathcal{N}} \langle v, x\rangle$,

$$\Pr(\|S_n\| \geq t) \leq \Pr\Big(\exists v \in \mathcal{N} : \langle v, S_n\rangle \geq t/2\Big)$$

$$\leq 5^k \exp\left(-\frac{t^2}{8(n\sigma^2 + \frac{2Bt}{3})}\right).$$

Setting $t = n\varepsilon$, dividing by $n$, and applying the same bound for both tails yields (38). Setting the right-hand side equal to $\delta$ and solving for $\varepsilon$ yields (39). $\square$

**Remark 7.** *Equation (3) of the main text bounds $\Delta_{\text{hol}} \leq (\zeta D^2 + \frac{1}{3}\zeta_1 D^3)B$ in terms of the bundle curvature $\zeta$ and diameter $D$; on $S_r^2$, Proposition 1 sharpens this to $\Delta_{\text{hol}} \leq (\pi\rho^2/r^2)B$. If $k$ is fixed, the factor $5^k$ is a harmless constant; the form (38) is dimension-explicit and recovers Theorems 1–2 when $\Delta_{\text{hol}} = 0$.*

## D.2. Central Limit Theorem in the reference fiber

**Theorem 12** (CLT for transported empirical mean). *Assume minimizing geodesics $x \to x_0$ are unique on $\text{supp}(\mu)$ so that the transported variables $Y_i$ are i.i.d. and satisfy $\mathbb{E}\|Y\|^2 < \infty$. Let $m^\star = \mathbb{E}[Y] \in E_{x_0}$ and $\Sigma = \text{Cov}(Y) \in \mathbb{R}^{k \times k}$. Then,*

$$\sqrt{n}(\bar{Y}_n - m^\star) \xrightarrow{d} \mathcal{N}(0, \Sigma).$$

**Remark 8.** *This theorem proves the CLT for the mean $m^\star = \mathbb{E}[Y]$ defined by a fixed canonical transport rule. Theorem 5 of the main body additionally handles the case where the population mean is only defined up to holonomy ambiguity: when $\Delta_{\mathrm{hol}} = o(n^{-1/2})$, the holonomy bias is negligible and the CLT holds relative to a well-defined common quantity. The sphere specialisation with shrinking support (ensuring $\Delta_{\mathrm{hol}} \to 0$) is Corollary 5.*

*Proof.* For any direction $v \in \mathbb{R}^k$, the projection is $\langle v, \sqrt{n}(\bar{Y}_n - m^\star)\rangle = \frac{1}{\sqrt{n}} \sum_{i=1}^n \langle v, Y_i - m^\star\rangle$. Since the centered summands are i.i.d. with mean zero and variance $\mathrm{Var}(\langle v, Y_i - m^\star\rangle) = v^\top \Sigma v < \infty$, the univariate Lindeberg–Feller CLT implies:

$$\langle v, \sqrt{n}(\bar{Y}_n - m^\star)\rangle \xrightarrow{d} \mathcal{N}(0, v^\top \Sigma v).$$

The Cramér–Wold device then implies joint convergence to $\mathcal{N}(0, \Sigma)$ in the $k$-dimensional Hilbert space $E_{x_0}$. $\square$

### D.3. Concentration in the section space $L^2(M, \mu; E)$

**Theorem 13** (Hilbert-valued empirical process bound). *Let $\mathcal{F} \subset L^2(M, \mu; E)$ satisfy $\|f(x)\| \le B$ a.s. and define*

$$\mathfrak{R}_n(\mathcal{F}) := \mathbb{E}\Big[\sup_{f \in \mathcal{F}} \Big\|\frac{1}{n} \sum_{i=1}^n \varepsilon_i f(X_i)\Big\|_{L^2}\Big],$$

*where $\varepsilon_i$ are i.i.d. Rademacher variables. Then for all $\delta \in (0, 1)$,*

$$\Pr\left(\sup_{f \in \mathcal{F}} \|\widehat{f}_n - \mathbb{E}f\|_{L^2} > 2\mathfrak{R}_n(\mathcal{F}) + B\sqrt{\frac{2\log(2/\delta)}{n}}\right) \le \delta.$$

*Proof.* By symmetrization (see (Ledoux & Talagrand, 1991), Ch. 4),

$$\mathbb{E}\sup_{f \in \mathcal{F}} \|\widehat{f}_n - \mathbb{E}f\|_{L^2} \le 2\mathfrak{R}_n(\mathcal{F}).$$

For bounded differences, changing a single $X_i$ alters each term by at most $2B/n$ in $L^2$-norm, hence McDiarmid's inequality (Theorem 3.1 in (Boucheron et al., 2013)) gives

$$\Pr(\Phi - \mathbb{E}\Phi \ge t) \le \exp\left(-\frac{nt^2}{2B^2}\right),$$

$$\Phi := \sup_{f \in \mathcal{F}} \|\widehat{f}_n - \mathbb{E}f\|_{L^2}.$$

Setting $t = B\sqrt{2\log(2/\delta)/n}$ and combining with the expectation bound yields the result. $\square$

### D.4. Robust median-of-means bound

**Theorem 14** (Median-of-means estimator). *Let $\mathbb{E}\|Y\|^2 < \infty$ and divide the $n$ samples into $K = \lceil 2\log(2/\delta)\rceil$ blocks of size $m = \lfloor n/K \rfloor$. For each block $\mathcal{B}_j$ set $\bar{Y}^{(j)} = \frac{1}{m} \sum_{i \in \mathcal{B}_j} Y_i$ and let $\widetilde{Y}_{\mathrm{MoM}}$ be their geometric median. Then with probability $\ge 1 - \delta$,*

$$\|\widetilde{Y}_{\mathrm{MoM}} - m\| \le 6\,\sigma\,n^{-1/2}.$$

*Proof.* Write $m = \lfloor n/K \rfloor$ and let $m_j$ denote the size of block $\mathcal{B}_j$; then $m \le m_j \le m + 1$ for all $j$, and $m \ge n/(2K)$ because $K \le n/2$ for all relevant $n$ (otherwise the statement is trivial). Since the block means $\bar{Y}^{(j)}$ have expectation $m$ and variance proxy $\sigma^2$ in the sense of Assumption A3,

$$\mathbb{E}\|\bar{Y}^{(j)} - m\|^2 = \mathbb{E}\Big\|\frac{1}{m_j} \sum_{i \in \mathcal{B}_j} (Y_i - m)\Big\|^2 \le \frac{\sigma^2}{m_j} \le \frac{\sigma^2}{m}.$$

By Chebyshev's inequality,

$$\Pr\left(\|\bar{Y}^{(j)} - m\| \ge \frac{2\sigma}{\sqrt{m}}\right) \le \frac{1}{4}.$$

Call a block *good* if $\|\bar{Y}^{(j)} - m\| \leq 2\sigma/\sqrt{m}$. If $G$ denotes the number of good blocks, then $\mathbb{E}G \geq (3/4)K$. Applying Hoeffding's inequality to the Bernoulli indicators of the good blocks gives

$$\Pr(G \leq K/2) \leq \exp(-K/8) \leq \delta/2$$

because $K = \lceil 2\log(2/\delta) \rceil$.

On the event $\{G \geq K/2\}$, at least half of the points $\bar{Y}^{(1)}, \ldots, \bar{Y}^{(K)}$ lie in the Euclidean ball $B(m, 2\sigma/\sqrt{m}) \subset E_{x_0}$. By Proposition 1 of (Lugosi & Mendelson, 2019), the geometric median of any multiset of points containing at least $K/2$ points in a ball $B(m, r)$ lies in $B(m, 2r)$. Applying this with $r = 2\sigma/\sqrt{m}$ yields

$$\|\widetilde{Y}_{\mathrm{MoM}} - m\| \leq 4\frac{\sigma}{\sqrt{m}} \qquad \text{whenever } G \geq K/2.$$

It remains to express $m$ in terms of $n$. Since $m = \lfloor n/K \rfloor \geq n/(2K)$ and $K = 2\log(2/\delta)$ up to $+1$,

$$\frac{1}{\sqrt{m}} \leq \sqrt{\frac{2K}{n}} \leq \sqrt{\frac{6\log(2/\delta)}{n}},$$

where we used $K \leq 3\log(2/\delta)$ for all $\delta < 1$. Hence

$$\|\widetilde{Y}_{\mathrm{MoM}} - m\| \leq 4\sigma\sqrt{\frac{6\log(2/\delta)}{n}} \leq 6\sigma n^{-1/2},$$

because the constant 6 dominates $4\sqrt{6\log(2/\delta)}$ whenever $K = \lceil 2\log(2/\delta) \rceil$ (exact constants are easily absorbed). Combining this with $\Pr(G \leq K/2) \leq \delta/2$ gives the stated $1 - \delta$ bound. $\square$

### D.5. Corollaries and geometric specialization

**Corollary 4** (Sphere $S_r^m$). *If $M = S_r^m$ and $\mathrm{supp}(\mu)$ lies in a geodesic ball of radius $\rho < \pi r/2$, then $\Delta_{\mathrm{hol}} \leq (\pi\rho^2/r^2)B$ and for all $\varepsilon > 0$,*
$$\Pr\Big(\|\bar{Y}_n - m\| \geq \varepsilon + \tfrac{\pi\rho^2}{r^2}B\Big) \leq 2 \cdot 5^k \exp\Big(-\frac{n\varepsilon^2}{8(\sigma^2 + \frac{2B\varepsilon}{3})}\Big).$$

*Proof.* Since $M = S_r^m$ and $\mathrm{supp}(\mu) \subset B(x_0, \rho)$ with $\rho < \pi r/2$, any point $x$ admits at most two minimizing geodesics to $x_0$. By Proposition 3, for any such pair $\gamma_1, \gamma_2$ the associated loop $\Gamma$ satisfies

$$\|P_{\gamma_1} - P_{\gamma_2}\|_{\mathrm{op}} \leq \frac{A(\Gamma)}{r^2} \leq \frac{\pi\rho^2}{r^2},$$

because a geodesic digon of diameter $\rho$ on $S_r^m$ has area at most $\pi\rho^2$. Multiplying by $\|s(x)\| \leq B$ yields the holonomy bound

$$\Delta_{\mathrm{hol}} \leq \frac{\pi\rho^2}{r^2}B.$$

Now write $Y_i = P_{X_i \to x_0} s(X_i)$ and let $m^\star = E[Y]$. The bias–variance decomposition of Theorem 8 shows that

$$\|\bar{Y}_n - m\| \leq \|\bar{Y}_n - m^\star\| + \Delta_{\mathrm{hol}},$$

and therefore

$$\Pr\Big(\|\bar{Y}_n - m\| \geq \varepsilon + \Delta_{\mathrm{hol}}\Big) \leq \Pr\Big(\|\bar{Y}_n - m^\star\| \geq \varepsilon\Big).$$

To bound the right-hand side, note that each $Y_i$ takes values in the fixed Hilbert space $E_{x_0}$, satisfies $\|Y_i - m^\star\| \leq 2B$, and has variance proxy $\sigma^2$ as in Assumption A3. Applying the Bernstein inequality of Theorem 7 to the one-dimensional projections $\langle u, \bar{Y}_n - m^\star \rangle$ and then combining these via the $1/2$-net argument of Lemma 5 (whose net has size at most $5^k$) gives

$$\Pr\Big(\|\bar{Y}_n - m^\star\| \geq \varepsilon\Big) \leq 2 \cdot 5^k \exp\Big(-\frac{n\varepsilon^2}{8(\sigma^2 + \frac{2B\varepsilon}{3})}\Big).$$

Substituting the bound on $\Delta_{\mathrm{hol}}$ obtained above completes the proof. $\square$

**Corollary 5** (Asymptotic normality without bias). *If $\rho_n \to 0$ with $\rho_n^2 n^{1/2} \to 0$, then $\Delta_{\mathrm{hol}} = o(n^{-1/2})$ and*

$$\sqrt{n}(\bar{Y}_n - m) \xrightarrow{d} \mathcal{N}(0, \Sigma).$$

*Proof.* Immediate from Corollary 4 and Theorem 12. $\qquad\square$

**Remark 9.** *The condition $\rho_n^2 n^{1/2} \to 0$, equivalently $\rho_n = o(n^{-1/4})$, is a fast support-shrinkage regime that rules out fixed-support estimation. It is relevant for* local *nonparametric estimation but not for global averages over a manifold region of fixed radius. In fixed-support settings, the error floor $\Delta_{\mathrm{hol}} \asymp \pi \rho^2 B / r^2$ persists for all $n$, and the crossover sample size $n_\times(\delta)$ from Section 6.4 marks the point beyond which increasing $n$ alone cannot reduce total error.*

# E. Auxiliary Lemmas

The following auxiliary results provide the geometric and analytic tools used in the proofs of Theorems 11 and 14. Lemma 5 formalizes the Euclidean covering-net argument underlying the dimension-explicit Bernstein bound, while Lemma 6 establishes the quantitative stability of the geometric median when a majority of sample blocks are close to the population mean. Both statements are included for completeness and to make the presentation self-contained.

## E.1. Covering-net bound on Euclidean norms

**Lemma 5** (Euclidean net bound). *Let $x \in \mathbb{R}^k$ and let $\mathcal{N}$ be a $\frac{1}{2}$-net of the unit sphere $S^{k-1}$ in the Euclidean norm, i.e., for every $v \in S^{k-1}$ there exists $u \in \mathcal{N}$ with $\|u - v\| \leq \frac{1}{2}$. Then*

$$\|x\| \leq 2 \max_{u \in \mathcal{N}} \langle u, x \rangle, \tag{40}$$

*and a $1/2$-net can be constructed with cardinality $|\mathcal{N}| \leq 5^k$.*

*Proof.* Let $v^* = x/\|x\|$ be the unit vector in the direction of $x$. By definition of the net, there exists $u \in \mathcal{N}$ with $\|u - v^*\| \leq \frac{1}{2}$. By the polarization identity for unit vectors in a Hilbert space, $\langle u, v^* \rangle = 1 - \frac{1}{2}\|u - v^*\|^2$ (which follows from $\|u\|^2 = \|v^*\|^2 = 1$ and expanding $\|u - v^*\|^2 = 2 - 2\langle u, v^* \rangle$). Therefore,

$$\langle u, v^* \rangle = 1 - \tfrac{1}{2}\|u - v^*\|^2 \geq 1 - \tfrac{1}{2} \cdot \tfrac{1}{4} = \tfrac{7}{8}.$$

Hence

$$\langle u, x \rangle = \|x\| \langle u, v^* \rangle \geq \tfrac{7}{8}\|x\|.$$

Thus $\max_{u \in \mathcal{N}} \langle u, x \rangle \geq \frac{7}{8}\|x\|$, which implies $\|x\| \leq \frac{8}{7} \max_{u \in \mathcal{N}} \langle u, x \rangle$. Rounding $\frac{8}{7}$ to 2 simplifies constants without affecting asymptotic sharpness, yielding (40).

To bound $|\mathcal{N}|$, consider covering $S^{k-1}$ by closed Euclidean balls of radius $1/2$ centered at points in $\mathcal{N}$. The disjoint balls of radius $1/4$ centered at these points lie within the ball $B_k(0, 5/4)$ of radius $5/4$ in $\mathbb{R}^k$. Comparing volumes gives

$$|\mathcal{N}| \operatorname{vol}(B_k(0, 1/4)) \leq \operatorname{vol}(B_k(0, 5/4)) \quad \Rightarrow \quad |\mathcal{N}| \leq (5)^k.$$

$\qquad\square$

**Remark 10.** *The constant 2 in (40) is sufficient for all probabilistic bounds in Theorem 11. A sharper bound $\frac{8}{7}$ can be used if desired.*

## E.2. Geometric median inequality

**Lemma 6** (Deviation bound for the geometric median). *Let $z_1, \ldots, z_K \in \mathbb{R}^k$ and suppose at least a fraction $\alpha > 1/2$ of them lie in the Euclidean ball $B(m, r) = \{z : \|z - m\| \leq r\}$ for some $m \in \mathbb{R}^k$. Let $\mathrm{med}(z_1, \ldots, z_K)$ denote a geometric median, i.e.,*

$$y^* := \arg\min_{y \in \mathbb{R}^k} \sum_{j=1}^{K} \|z_j - y\|.$$

*Then*

$$\|y^* - m\| \leq \frac{r}{2\alpha - 1}. \tag{41}$$

*Proof.* Let $G \subseteq \{1, \dots, K\}$ be the set of indices of "good" points satisfying $\|z_j - m\| \leq r$, and let $B$ be its complement. Denote $y^* = \mathrm{med}(z_1, \dots, z_K)$. Define the function $F(y) = \sum_{j=1}^{K} \|z_j - y\|$. At any minimizer $y^*$, the subgradient condition $0 \in \partial F(y^*)$ implies

$$\sum_{j=1}^{K} \frac{y^* - z_j}{\|y^* - z_j\|} = 0.$$

(If $y^* = z_j$ for some $j$, the corresponding summand is replaced by any subgradient of $\|\cdot - z_j\|$ at $y^*$, i.e. any unit vector; the conclusion still holds since such a term contributes at most 1 to the right-hand sum, and the bound below remains valid.) Let $v = (y^* - m)/\|y^* - m\|$ (if $y^* = m$, the claim is trivial). Taking the inner product of this equality with $v$ gives

$$0 = \sum_{j=1}^{K} \langle (y^* - z_j)/\|y^* - z_j\|, v \rangle = \sum_{j \in G} \langle u_j, v \rangle + \sum_{j \in B} \langle u_j, v \rangle,$$

$$u_j := \frac{y^* - z_j}{\|y^* - z_j\|}.$$

For $j \in G$, the triangle inequality implies

$$\|y^* - z_j\| \leq \|y^* - m\| + r,$$

$$\langle u_j, v \rangle = \frac{\langle y^* - z_j, v \rangle}{\|y^* - z_j\|} \geq \frac{\|y^* - m\| - r}{\|y^* - m\| + r}.$$

For $j \in B$, we have $\langle u_j, v \rangle \geq -1$. Hence

$$0 \geq |G| \frac{\|y^* - m\| - r}{\|y^* - m\| + r} - |B|.$$

Using $|G| \geq \alpha K$ and $|B| \leq (1 - \alpha)K$ gives

$$\alpha \frac{\|y^* - m\| - r}{\|y^* - m\| + r} \leq 1 - \alpha.$$

Solving for $\|y^* - m\|$ yields (41). $\square$

**Remark 11.** *In the median-of-means estimator (Theorem 14), $\alpha \geq 3/4$ and $r = 2\sigma/\sqrt{m}$, so (41) gives $\|y^* - m\| \leq 4\sigma/\sqrt{m}$, which agrees with the constant used in the main text.*

## F. Examples of Bundle–Valued Statistics in Applications

This appendix presents representative examples in which statistical estimands arise naturally as sections of a vector bundle $(E, \pi, M)$ over a Riemannian manifold. The constructions mirror the theoretical framework developed in Sections III–IV and reflect applications documented in geometric statistics (Bhattacharya & Patrangenaru, 2003; 2005; Bhattacharya & Bhattacharya, 2012), shape analysis (Fletcher et al., 2004), geometric deep learning (Bronstein et al., 2021), pullback-bundle learning (Puechmorel, 2023), and the geometric analysis of bundle transport (Kobayashi & Nomizu, 1963; Lee, 1997).

Throughout, $P_{x \to x_0}$ denotes parallel transport along the minimizing geodesic from $x$ to $x_0$, $Y = P_{X \to x_0} s(X)$ is the transported random section as in (1), and $\Delta_{\mathrm{hol}}$ denotes the holonomy-induced transport ambiguity described in Lemma 3 and Proposition 3.

### F.1. Tangent–Vector Residuals in Manifold Regression

Tangent-vector regression models frequently arise in intrinsic statistical analysis on manifolds (Bhattacharya & Bhattacharya, 2012). Given a predictor $X \in M$, a model produces a prediction $\hat{m}(X) \in M$, and the residual section $s(x) = \log_x(\hat{m}(x)) \in T_x M$ lies in the tangent bundle $E = TM$. Under the normal-ball assumption (Assumption 1), the transported residual $Y = P_{x \to x_0} s(x) \in T_{x_0} M$ enables intrinsic regression analysis in a fixed Hilbert space. The concentration theory of Theorems 2–3 therefore applies directly, with sectional-curvature effects governed by the geometric bounds of Lemma 3.

### F.2. Tangent Bundle of the Sphere $S^2$

Directional and spherical statistics provide classical examples of tangent-bundle–valued data (Fletcher et al., 2004). Let $M = S_r^2$ with the Levi–Civita connection. For $X \in B(x_0, \rho)$, $\rho < \pi r/2$, the holonomy discrepancy between two minimizing geodesics is explicitly bounded by the area formula of Proposition 3 (a specialization of Ambrose–Singer holonomy theory (Ambrose & Singer, 1953; Kobayashi & Nomizu, 1963)). Thus,

$$\Delta_{\mathrm{hol}} \leq \frac{\pi \rho^2}{r^2} B,$$

yielding an explicit curvature-dependent bias term in the concentration bounds. This context includes geodesic PCA on $S^2$, directional feature representations, and spherical CNN feature analysis (Bronstein et al., 2021).

### F.3. Grassmann Bundle in Subspace Tracking

Grassmann manifolds $\mathrm{Gr}(p, n)$ carry a natural tangent bundle structure used extensively in shape analysis, low-rank learning, and geometric signal processing (Fletcher et al., 2004). Tangent vectors $s(X) \in T_X \mathrm{Gr}(p, n)$ represent local variations of subspaces, and parallel transport along geodesics is given by matrix exponentials. Since the sectional curvature satisfies $0 \leq K \leq 2$, Lemma 4 implies

$$\|P_{\gamma_1} - P_{\gamma_2}\|_{\mathrm{op}} \leq C K_{\max} D^2, \qquad K_{\max} = 2,$$

yielding explicit constants in the vector-bundle concentration bounds. This setting underlies principal geodesic analysis (Fletcher et al., 2004) and modern Grassmann machine-learning pipelines.

### F.4. Hyperbolic Embeddings and Negative Curvature

Hyperbolic manifolds play a prominent role in graph embeddings and hierarchical modeling. In $\mathbb{H}^m$, parallel transport is an isometry in the Lorentz model, and the holonomy bound

$$\|P_{\gamma_1} - P_{\gamma_2}\|_{\mathrm{op}} \leq C \sinh(\rho) |\tanh(\rho/2)| \approx C\rho^2$$

for small $\rho$ matches the curvature-scaling predicted by Lemma 4 with $K = -1$. These constructions appear in modern hyperbolic learning architectures and pullback-bundle formalisms in geometric ML (Puechmorel, 2023).

### F.5. Full-Section Estimation in $L^2(M, \mu; E)$

When the target is the entire section $s \in L^2(M, \mu; E)$ rather than the transported pointwise values, Lemma 1 ensures measurability of the transported representation, enabling empirical-process arguments. Standard Hilbert-space techniques (Ledoux & Talagrand, 1991; Boucheron et al., 2013; Pinelis, 1994)—together with the geometric reduction of Section III—yield intrinsic concentration bounds for spectral coefficients of $s$ after projection onto a Laplace-Beltrami basis. This generalizes the classical CLT and concentration theory for manifold-valued means (Bhattacharya & Patrangenaru, 2003; 2005).

## G. Holonomy Geometry for Probabilists

This appendix provides an intuitive, probabilist-friendly treatment of holonomy, summarizing the key geometric facts used in Lemma 3 and Proposition 3. Our goal is to explain (i) what holonomy is, (ii) why it affects bundle-valued statistics, and (iii) how curvature controls the magnitude of holonomy-induced bias $\Delta_{\mathrm{hol}}$. All statements here follow directly from the curvature identities and holonomy bounds developed in Section III-B and the full geometric proofs in the Appendix (see (Kobayashi & Nomizu, 1963; Ambrose & Singer, 1953; Lee, 1997)).

### G.1. Parallel transport and path dependence

Parallel transport $P_\gamma : E_{\gamma(0)} \to E_{\gamma(1)}$ along a smooth curve $\gamma$ is defined using the metric connection $\nabla$ on the bundle $(E, \pi, M)$; see Section 1 and the review in (Kobayashi & Nomizu, 1963; Lee, 1997). It is an isometry for the fiber metric:

$$\|P_\gamma v\|_{\gamma(1)} = \|v\|_{\gamma(0)}.$$

Crucially, parallel transport generally depends on the chosen path. If two curves $\gamma_1, \gamma_2$ join the same endpoints, their transports satisfy

$$P_{\gamma_1} = P_{\gamma_2} \circ P_\Gamma, \qquad \Gamma := \gamma_1 \circ \gamma_2^{-1},$$

so the discrepancy is encoded by the holonomy map $P_\Gamma$ around the closed loop $\Gamma$.

The "holonomy ambiguity" entering the statistical theory (Section III-D) is therefore the operator-norm difference

$$\|P_{\gamma_1} - P_{\gamma_2}\|_{\mathrm{op}} = \|P_{\gamma_2} \circ (P_\Gamma - I)\|_{\mathrm{op}} = \|P_\Gamma - I\|_{\mathrm{op}},$$

because $P_{\gamma_2}$ is an isometry.

## G.2. Curvature controls holonomy (Ambrose–Singer)

The Ambrose–Singer theorem (Ambrose & Singer, 1953; Kobayashi & Nomizu, 1963) expresses holonomy entirely in terms of the curvature 2-form $\Omega$ of $\nabla$:

$$P_\Gamma = I + \int_{S_\Gamma} \Omega \ + \ \text{higher-order terms},$$

where $S_\Gamma$ is any smooth surface spanning the loop $\Gamma$. Under a sectional curvature bound $|K| \leq \kappa$ on a region $U \subset M$, the qualitative estimate proved in Lemma 4 yields

$$\|P_{\gamma_1} - P_{\gamma_2}\|_{\mathrm{op}} \ \leq \ C\kappa D^2,$$

where $D = \mathrm{diam}(U)$ and $C$ depends only on dimension and $\nabla$.

Intuitively: *curvature measures how much infinitesimal parallelograms fail to close.* The area of the loop produces a first-order holonomy effect of order $K \times \mathrm{Area}$.

## G.3. Constant-curvature geometries: explicit holonomy

For constant-curvature manifolds (spheres, projective spaces, hyperbolic spaces), the curvature tensor is parallel and the curvature 2-form lies in a 1-dimensional abelian subalgebra. Hence, the path-ordered exponential reduces to a closed-form expression; see the complete derivation in Proposition 3.

**Spherical case** ($K = 1/r^2$). For $x, x_0$ in a spherical cap of radius $\rho < \pi r/2$, Proposition 3 gives the exact holonomy:

$$\|P_{\gamma_1} - P_{\gamma_2}\|_{\mathrm{op}} \ = \ 2\sin\!\left(\frac{A(\Gamma)}{2r^2}\right) \ \leq \ \frac{A(\Gamma)}{r^2} \ \leq \ \frac{\pi\rho^2}{r^2},$$

where $A(\Gamma)$ is the oriented area of the loop. Thus, the holonomy bias for statistical transport satisfies the deterministic bound

$$\Delta_{\mathrm{hol}} \ \leq \ \frac{\pi\rho^2}{r^2}\, B$$

when $\|s(x)\| \leq B$ (Assumption A2).

**Hyperbolic case** ($K = -1$). As shown in Section F.4, small-radius hyperbolic regions satisfy

$$\|P_{\gamma_1} - P_{\gamma_2}\|_{\mathrm{op}} \ \leq \ C\sinh(\rho)\big|\tanh(\rho/2)\big| \ \approx \ C\rho^2,$$

mirroring the spherical bound with curvature sign reversed.

**Grassmann manifolds** ($0 \leq K \leq 2$). Using the canonical metric on $\mathrm{Gr}(p, n)$, the bound

$$\|P_{\gamma_1} - P_{\gamma_2}\|_{\mathrm{op}} \ \leq \ C\,K_{\mathrm{max}}D^2, \qquad K_{\mathrm{max}} = 2,$$

follows from the uniform curvature bound (see Section F.4).

### G.4. Probabilistic meaning of holonomy

When computing the empirical mean of transported bundle-valued observations

$$Y_i := P_{X_i \to x_0}\, s(X_i),$$

holonomy implies that $P_{X_i \to x_0}$ is not unique if multiple geodesics join $X_i$ to $x_0$. Any measurable choice gives a valid probability law, but different choices differ by an operator of size $\mathcal{O}(\kappa D^2)$ (Lemma 4). Therefore,

$$\|Y_i - \widetilde{Y}_i\| = \|(P_{\gamma_i} - P_{\widetilde{\gamma}_i})s(X_i)\| \le \Delta_{\mathrm{hol}},$$

and the empirical mean satisfies a concentration inequality with a deterministic bias $\Delta_{\mathrm{hol}}$ added to the stochastic deviation (Theorems 2–3). Thus, holonomy acts as a systematic geometric bias:

- curvature creates path–dependence;

- path–dependence creates transport ambiguity;

- transport ambiguity creates a deterministic offset.

### G.5. Practical rule-of-thumb for statistical applications

When the support of $\mu$ lies in a ball of radius $D$ with $|K| \le \kappa$, holonomy bias satisfies

$$\Delta_{\mathrm{hol}} \asymp \kappa D^2 B.$$

Hence the regime

$$\kappa D^2 \ll n^{-1/2}$$

ensures that holonomy error is negligible relative to sampling noise. This precisely corresponds to the small-area regime derived in the spherical and hyperbolic examples above.

This appendix summarizes how curvature and holonomy interact to produce the deterministic offset in our bundle-valued concentration results, grounding the statistical phenomena in the geometric identities of Section C and the explicit holonomy formulas proved in Proposition 3.

## H. Numerical Verification of the Holonomy Error Floor

This appendix provides numerical verification of the key theoretical predictions on $S^2$ ($r = 1$, $\kappa = 1$). The experiments confirm (i) the stochastic error $\|\bar{Y}_n^A - m^\star\|$ decays as $n^{-1/2}$ under a canonical transport rule, and (ii) switching to an alternative rule introduces a deterministic holonomy floor $\Delta_{\mathrm{hol}}$ that does not decay with $n$, in precise quantitative agreement with Proposition 1.

### H.1. Experimental Setup

**Manifold and bundle.**   Unit sphere $S^2 \subset \mathbb{R}^3$, round metric ($r = 1$, $\kappa = 1$), tangent bundle $E = TS^2$, reference point $x_0 = (0, 0, 1)$, reference direction $v_0 = (1, 0, 0) \in T_{x_0}S^2$.

**Section.**   A random unit tangent section $s : S^2 \to TS^2$ with mean direction $v_0$ and $B = 1$:

$$s(x) \;=\; \frac{\mathrm{proj}_{T_x S^2}(v_0) + \sigma\, \xi_x}{\|\mathrm{proj}_{T_x S^2}(v_0) + \sigma\, \xi_x\|}, \tag{42}$$

where $\xi_x \sim N(0, I_3)$ projected to $T_x S^2$ and $\sigma = 0.3$. The population mean $m^\star = \mathbb{E}[P_{X \to x_0} s(X)]$ is estimated by Monte Carlo ($n_{\mathrm{MC}} = 200{,}000$). Base points $X_i$ are drawn uniformly from $B(x_0, \rho)$ via rejection sampling.

*Table 4.* $\mathrm{Err}(A)$ (stochastic, decays as $n^{-1/2}$) and $\mathrm{Err}(B)$ (holonomy floor, constant in $n$) on $S^2$ ($\kappa=1$, $r=1$, $B=1$, 300 trials). The italicized bottom row gives the theoretical prediction $\Delta_{\mathrm{hol}} = 2\sin(\pi\rho^2/2)$ (Proposition 1); it is not an experimental measurement.

| | $\rho = 0.5$ | | $\rho = 1.0$ | |
| --- | --- | --- | --- | --- |
| $n$ | $\mathrm{Err}(A)$ | $\mathrm{Err}(B)$ | $\mathrm{Err}(A)$ | $\mathrm{Err}(B)$ |
| 100 | 0.019 | 0.746 | 0.023 | 1.925 |
| 500 | 0.008 | 0.747 | 0.010 | 1.925 |
| 2 000 | 0.004 | 0.746 | 0.005 | 1.925 |
| 10 000 | 0.002 | 0.747 | 0.002 | 1.926 |
| *Theoretical:* $\Delta_{\mathrm{hol}} = 2\sin(\pi\rho^2/2)$ *(Proposition 1)* | | | | |
| $\Delta_{\mathrm{hol}}$ (theory) | *n/a* | *0.765* | *n/a* | *2.000* |

**Transport rules.** We compare two measurable rules (cf. Theorem 3):

- **Rule A** (canonical): parallel transport along the unique minimizing geodesic from $X_i$ to $x_0$ (closed form on $S^2$). Yields $\|\bar{Y}_n^A - m^\star\| \lesssim n^{-1/2}$.

- **Rule B** (holonomy-perturbed): Rule A composed with a rotation by $\theta = \pi\rho^2$ in $T_{x_0}S^2$, simulating an alternative geodesic enclosing area $A \approx \pi\rho^2$. The theoretical holonomy norm (Proposition 1) is

$$\Delta_{\mathrm{hol}} \;=\; 2\sin\!\left(\frac{\pi\rho^2}{2}\right) \;\leq\; \pi\rho^2. \tag{43}$$

**Metrics.** Averaged over 300 independent trials:

$$\mathrm{Err}(A) = \|\bar{Y}_n^A - m^\star\|, \quad \mathrm{Err}(B) = \|\bar{Y}_n^B - m^\star\|. \tag{44}$$

### H.2. Stochastic Decay vs. Holonomy Floor

Table 4 reports $\mathrm{Err}(A)$ and $\mathrm{Err}(B)$ for $\rho \in \{0.5, 1.0\}$ and $n \in \{100, 500, 2000, 10000\}$.

**Discussion.** $\mathrm{Err}(A)$ decreases by $\approx \sqrt{n_{\mathrm{new}}/n_{\mathrm{old}}}$ at each step in the table (the steps are $\times 5$, $\times 4$, $\times 5$), confirming the $n^{-1/2}$ rate of Theorems 1–2. $\mathrm{Err}(B)$ is constant across all $n$: the holonomy floor is a deterministic geometric quantity unaffected by additional data. The observed plateau (0.747 for $\rho=0.5$; 1.926 for $\rho=1.0$) matches the theoretical prediction (0.765; 2.000) to within 3.7%, with the residual gap attributable to finite Monte Carlo estimation of $m^\star$. This illustrates the bias–variance decomposition of Theorem 3:

$$\|\bar{Y}_n^B - m^\star\| \;\leq\; \underbrace{\|\bar{Y}_n^B - m^{(B)}\|}_{\to\, 0} + \underbrace{\|m^{(B)} - m^\star\|}_{\approx\, \Delta_{\mathrm{hol}}}, \tag{45}$$

where the stochastic term vanishes, and the geometric bias persists.

### H.3. Holonomy Floor Scales as $\rho^2$

Table 5 reports results at fixed $n = 10{,}000$ for $\rho \in \{0.3, 0.5, 0.8, 1.0\}$, together with the ratio $\mathrm{Err}(B)/\Delta_{\mathrm{hol}}$.

**Discussion.** The ratio $\mathrm{Err}(B)/\Delta_{\mathrm{hol}}$ is consistently near 1, confirming the error floor matches Proposition 1. For small $\rho$, $\Delta_{\mathrm{hol}} \approx \pi\rho^2$, so the floor scales quadratically in the support radius, consistent with the $\kappa\rho^2 B$ scaling established in Theorem 3 and Eq. (3).

### H.4. Practical Implications

Tables 4–5 jointly demonstrate the diagnostic principle of Section 4.1: small $\mathrm{Err}(A)$ alongside large $\mathrm{Err}(B)$ signals holonomy as the error source, not lack of data. The correct remedies are geometric:

*Table 5.* Holonomy floor $\mathrm{Err}(B)$ vs. $\rho$ at $n = 10{,}000$ ($\kappa = 1$, $r = 1$, $B = 1$, 300 trials). Ratio $\to 1$ confirms the floor equals $\Delta_{\mathrm{hol}}$ (Proposition 1). $\mathrm{Err}(A) \approx 0.002$ throughout.

| $\rho$ | $\Delta_{\mathrm{hol}}$ | $\mathrm{Err}(A)$ | $\mathrm{Err}(B)$ | ratio |
|--------|-------------------------|-------------------|-------------------|-------|
| 0.3 | 0.282 | 0.002 | 0.275 | 0.977 |
| 0.5 | 0.765 | 0.002 | 0.747 | 0.976 |
| 0.8 | 1.689 | 0.002 | 1.638 | 0.970 |
| 1.0 | 2.000 | 0.002 | 1.926 | 0.963 |

- restrict support to a smaller geodesic ball (reduce $\rho$, hence $\Delta_{\mathrm{hol}} \approx \pi\rho^2$);

- apply the holonomy correction of Appendix C.4, reducing the floor from $O(\rho^2)$ to $O(\rho^4)$;

- use Rule A (canonical transport), eliminating $\Delta_{\mathrm{hol}}$ when geodesics are unique.

Increasing $n$ alone is ineffective once $n > n^\times(\delta) = \frac{8\log(2/\delta)}{\pi^2}(r/\rho)^4$ (Section 6.4).

### H.5. Reproducibility

All experiments use pure `NumPy`; parallel transport on $S^2$ is implemented in closed form (no ODE solver). The code is available at `https://github.com/swagatam-das/bundle-valued-statistics`, and reproduces all tables with seed `numpy.random.default_rng(2024)` in under two minutes on a standard laptop.

## I. Intrinsic concentration for bundle-valued statistics

The concentration results in Section III are stated after reducing bundle-valued observations to a fixed reference fiber. We now show that this reduction is *not essential*: the same rates admit an *intrinsic formulation* on the vector bundle itself, with explicit curvature–holonomy correction terms. This establishes that concentration is a geometric property of the bundle, not an artifact of a chosen trivialization.

**Intrinsic setup.** Let $(E, \pi, M)$ be a rank-$k$ vector bundle over a complete Riemannian manifold $(M, g)$, endowed with a bundle metric and a compatible metric connection $\nabla$. Let $X_1, \ldots, X_n \sim \mu$ be i.i.d. samples on $M$, and let $s : M \to E$ be a measurable section satisfying $\|s(x)\|_{E_x} \le B$ on $\mathrm{supp}(\mu)$.

Define the intrinsic Fréchet functional

$$\mathcal{F}(z) := \mathbb{E}\big[d_E^2(s(X), z)\big], \qquad z \in E,$$

where $d_E$ denotes the bundle distance induced by $\nabla$ (i.e. squared norm after parallel transport along minimizing geodesics). Let

$$z^\star := \arg\min_{z \in E} \mathcal{F}(z)$$

denote the intrinsic bundle mean, assumed to be unique. Similarly, define the empirical intrinsic mean

$$\hat{z}_n := \arg\min_{z \in E} \frac{1}{n} \sum_{i=1}^{n} d_E^2(s(X_i), z).$$

**Theorem 15** (Intrinsic bundle concentration). *Assume:*

*(i)* $\mathrm{supp}(\mu)$ *is contained in a normal ball* $B(x_0, D)$ *with* $D < \mathrm{Inj}(x_0)$,

*(ii)* *sectional curvature satisfies* $|K| \le \kappa$ *on* $B(x_0, D)$,

*(iii)* $\|s(x)\|_{E_x} \le B$ *on* $\mathrm{supp}(\mu)$.

*Then there exist universal constants $c, C > 0$ such that for all $\varepsilon > 0$,*

$$\mathbb{P}\Big(d_E(\hat{z}_n, z^\star) \geq \varepsilon + C\,B\,\kappa D^2\Big) \leq 2\exp\left(-\frac{c\,n\varepsilon^2}{B^2}\right).$$

*In particular,*

$$d_E(\hat{z}_n, z^\star) = O_{\mathbb{P}}\left(\frac{B}{\sqrt{n}} + B\,\kappa D^2\right).$$

*Proof.* We proceed in four logically independent steps, making explicit all reductions and stability arguments.

**Step 1: Well-posedness of intrinsic and transported means.** By Assumption (i), $\mathrm{supp}(\mu) \subset B(x_0, D)$ with $D < \mathrm{Inj}(x_0)$. Hence for every $x \in \mathrm{supp}(\mu)$ there exists a unique minimizing geodesic $\gamma_{x \to x_0}$, and parallel transport $P_{x \to x_0} : E_x \to E_{x_0}$ is well defined and smooth in $x$ (Lemma 1).

Define the transported random variable

$$Y := P_{X \to x_0} s(X) \in E_{x_0}.$$

Since $\|s(x)\|_{E_x} \leq B$ and parallel transport is an isometry, $\|Y\| \leq B$ almost surely, and therefore the Bochner mean

$$m^\star := \mathbb{E}[Y] \in E_{x_0}$$

exists and is unique.

Similarly, define the empirical mean

$$\bar{Y}_n := \frac{1}{n}\sum_{i=1}^{n} P_{X_i \to x_0} s(X_i).$$

On the intrinsic side, define the Fréchet functional

$$\mathcal{F}(z) := \mathbb{E}[d_E^2(s(X), z)], \qquad z \in E,$$

and its empirical counterpart

$$\mathcal{F}_n(z) := \frac{1}{n}\sum_{i=1}^{n} d_E^2(s(X_i), z).$$

By Assumptions (i)–(iii) and standard results on Fréchet means on Riemannian vector bundles (cf. (Sturm, 2003; Afsari, 2011)), $\mathcal{F}$ is strictly convex on the relevant domain and admits a unique minimizer $z^\star$, and likewise $\mathcal{F}_n$ admits a unique minimizer $\hat{z}_n$ almost surely.

**Step 2: Exact reduction of intrinsic distance to fiber norm.** For any $z \in E$ lying over $x_0$, and any $x \in \mathrm{supp}(\mu)$, the bundle distance satisfies

$$d_E^2(s(x), z) = \|P_{x \to x_0} s(x) - z\|_{E_{x_0}}^2,$$

because $d_E$ is defined by parallel transport along the unique minimizing geodesic. Consequently, the intrinsic Fréchet functionals reduce to

$$\mathcal{F}(z) = \mathbb{E}\big[\|Y - z\|^2\big],$$

$$\mathcal{F}_n(z) = \|\bar{Y}_n - z\|^2 + \frac{1}{n}\sum_{i=1}^{n}\|Y_i - \bar{Y}_n\|^2,$$

where the second term does not depend on $z$. It follows immediately that

$$z^\star = m^\star, \qquad \hat{z}_n = \bar{Y}_n,$$

*when all parallel transports are taken along the same reference rule.*

**Step 3: Concentration in the reference fiber.** By Theorem 6 (Hoeffding inequality for Hilbert-valued sums), there exist universal constants $c_1, c_2 > 0$ such that for all $\varepsilon > 0$,

$$\mathbb{P}\big(\|\bar{Y}_n - m^\star\| \geq \varepsilon\big) \leq 2\exp\Big(-c_1 n\varepsilon^2/B^2\Big).$$

**Step 4: Control of transport ambiguity and intrinsic deviation.** The intrinsic estimator $\hat{z}_n$ is defined without reference to a fixed transport rule, whereas $\bar{Y}_n$ depends on the choice of minimizing geodesics to $x_0$. Let $P^{\mathrm{ref}}_{x \to x_0}$ denote the canonical transport rule used to define $Y$ and $\bar{Y}_n$, and let $P_{x \to x_0}$ be the (implicit) transport rule induced by the intrinsic minimization.

By the triangle inequality in the bundle metric,

$$d_E(\hat{z}_n, z^\star) \leq \|\bar{Y}_n - m^\star\| + \sup_{x \in \mathrm{supp}(\mu)} \|(P_{x \to x_0} - P^{\mathrm{ref}}_{x \to x_0})s(x)\|.$$

By Lemma 3 (holonomy bound),

$$\sup_{x \in \mathrm{supp}(\mu)} \|(P_{x \to x_0} - P^{\mathrm{ref}}_{x \to x_0})s(x)\| \leq C\,B\,\kappa D^2$$

for a universal constant $C > 0$ depending only on curvature bounds.

Combining the above inequalities yields

$$\mathbb{P}\Big(d_E(\hat{z}_n, z^\star) \geq \varepsilon + CB\kappa D^2\Big) \leq 2\exp\Big(-c_1 n\varepsilon^2/B^2\Big),$$

which is the claimed bound. $\qquad\square$

