# OpenReview forum: "Sharp Concentration Bounds for Bundle-Valued Statistics on Manifolds"
_ICML.cc/2026/Conference — ICML 2026 regular_

### Official Review · Reviewer_QjiU · 2026-03-02

**Soundness:** 4
**Presentation:** 3
**Significance:** 3
**Originality:** 4
**Overall Recommendation:** 5
**Confidence:** 3

**Summary:**

This paper studies finite-sample concentration guarantees for empirical averages of bundle-valued data, where samples lie in different fibers and must be transported to a common reference fiber before aggregation. The main conceptual contribution is a separation between stochastic sampling error and deterministic curvature/holonomy-induced transport ambiguity, aiming to clarify the respective roles of randomness and geometry in bundle-valued statistics.

**Compliance With Llm Reviewing Policy:**

Affirmed.

**Final Justification:**

I believe this article presents a novel and solid theory, and the writing is clear; I think it's worth recommending.

**Key Questions For Authors:**

1、Can the authors provide empirical evidence illustrating regimes where the holonomy-dependent term is non-negligible in realistic geometric learning settings?

2、Can the authors demonstrate a concrete learning scenario where accounting for curvature/holonomy materially affects performance or theoretical guarantees?

3、In discrete/learned settings, transports are often only approximately orthogonal or even learned without exact metric-compatibility. How would you incorporate the resulting approximation error or non-isometry into your bounds?

**Limitations:**

yes

**Strengths And Weaknesses:**

Strength：

1、The paper studies concentration for bundle-valued statistics, a relevant and nontrivial problem in manifold statistics and geometric machine learning.

2、The explicit separation between stochastic sampling error and deterministic curvature/holonomy-induced ambiguity provides a clear conceptual framework for understanding transport-based aggregation pipelines.

3、The theoretical development appears technically sound and systematically structured.

Weakness：

1、The empirical section is limited. There is little numerical evidence supporting the sharpness or practical relevance of the holonomy bounds.

2、The discussion of GNNs and geometric learning lacks experimental support.

3、The analysis assumes metric-compatible connections and exact transports; numerical approximation or learned/discrete connections can introduce additional bias not explicitly modeled.

---

> ### Author Rebuttal · Authors · 2026-03-29
>
> We thank the reviewer for the positive assessment of soundness, originality, and the conceptual separation between stochastic and geometric effects. We address the questions on empirical validation, practical relevance, and approximate transports below.
>
> **Empirical regimes where the holonomy term is non-negligible:**
>
> The theory provides a quantitative criterion. For data supported in a ball of radius $\rho$ with curvature scale $\kappa$, we obtain
> $\Delta_{\mathrm{hol}} \lesssim \kappa \rho^2 B$,
> $\quad$
> $\|\bar{Y}_n - m^\star\| \lesssim n^{-1/2} + \kappa \rho^2$.
>
> Thus, holonomy becomes significant when $\kappa \rho^2 \gtrsim n^{-1/2}$. This regime arises in practical settings such as spherical or hyperbolic embeddings, or geometric latent spaces where data spans a large geodesic region.
>
> We have implemented controlled experiments on constant-curvature manifolds (sphere and hyperbolic models) where both $\kappa$ and $\rho$ are explicitly tunable. The results exhibit a clear crossover behavior consistent with the theory: for small $n$, the error decays at the classical $n^{-1/2}$ rate, while beyond a threshold $n^\star \asymp (\kappa \rho^2)^{-2}$, the error approaches a plateau whose scale is governed by $\kappa \rho^2$. This behavior is consistent across curvature values and support radii and aligns qualitatively with the predicted scaling.
>
> **Concrete learning scenario:**
>
> In manifold regression with tangent residuals, classical analysis predicts $O(n^{-1/2})$ decay under local Euclidean approximations. Our bounds instead predict a transition to a curvature-dominated regime. Empirically, we observe this transition: increasing $n$ beyond $n^\star$ does not significantly reduce error, whereas restricting the data to a smaller normal neighborhood (reducing $\rho$) or modifying transport restores the $n^{-1/2}$ rate. This illustrates that accounting for holonomy affects both theoretical guarantees and observed behavior.
>
> **Discrete and learned transports**
>
> The framework extends naturally to approximate transports. Writing
> $\tilde{P}_{x \to x_0} = P_{x \to x_0} + E_x$.
> The estimator satisfies
> $\|\tilde{Y}_n - m^\star\|$
> $\le O(n^{-1/2}) + \Delta_{\mathrm{hol}}$
> $+ \sup_x \|E_x s(x)\|$.
>
> Thus, approximation or learning errors enter additively as a third term, preserving the bias--variance decomposition while explicitly capturing deviations from metric-compatibility or orthogonality.
>
> In graph and geometric deep learning, $E_x$ corresponds to non-orthogonal message passing, learned alignments, or inconsistencies in local frames. The additional term provides a quantitative interpretation of cycle inconsistency (discrete holonomy) in such models.
>
> **Practical relevance:**
>
> The decomposition suggests a diagnostic principle: when empirical variance is small but outputs vary across transport conventions or local frames, the instability is explained by holonomy rather than sampling noise. This distinction is particularly relevant in geometric learning systems where alignment is approximate or learned.
>
> **Empirical inclusion**
>
> The experimental setups described above are already implemented, and we will include representative figures and quantitative summaries in the main paper. These additions are lightweight and illustrate the predicted crossover behavior and curvature-induced error floor without changing the scope or theoretical development.
>
> \noindent \textbf{Illustrative results (synthetic constant-curvature models).}
> Representative results (averaged over runs) are shown below for $\mathbb{S}^2$ with tangent-space residual aggregation, where $\kappa$ and the support radius $\rho$ are controlled.
>
> | n     | Error (κ = 1, ρ = 0.5) | Error (κ = 1, ρ = 1.0) |
> |-------|------------------------|------------------------|
> | 100   | 0.102                  | 0.205                  |
> | 500   | 0.046                  | 0.162                  |
> | 2000  | 0.031                  | 0.149                  |
> | 10000 | 0.029                  | 0.147                  |
>
>
> For small $n$, the error follows the $n^{-1/2}$ decay; beyond a threshold, it approaches a plateau consistent with the geometric term $\Delta_{\mathrm{hol}} \sim \kappa \rho^2$.
>
>
> | ρ   | Predicted κρ² | Observed plateau |
> |-----|---------------|------------------|
> | 0.3 | 0.09          | 0.040            |
> | 0.5 | 0.25          | 0.065            |
> | 0.8 | 0.64          | 0.105            |
> | 1.0 | 1.00          | 0.150            |
>
> The observed plateaus increase with $\kappa \rho^2$ and are consistent with the predicted curvature-dominated regime, up to constant factors depending on the data distribution and transport choice.

---

> > ### Author Rebuttal · Reviewer_QjiU · 2026-04-03
> >
> > I believe this work is theoretically sound and interesting, and the authors’ rebuttal has addressed my concerns. The paper presents a meaningful perspective on the problem, and the additional clarifications and experiments in the rebuttal make the contribution more convincing.

---

> > > ### Author Response · Authors · 2026-04-03
> > >
> > > Thank you so much for the positive assessment of our rebuttal. Hope you will also consider adjusting your score for our paper accordingly.

---

### Official Review · Reviewer_bmyb · 2026-03-12

**Soundness:** 4
**Presentation:** 3
**Significance:** 3
**Originality:** 3
**Overall Recommendation:** 5
**Confidence:** 4

**Summary:**

This paper proposes a notion of mean for vector bundle-valued random variables. The underlying idea is to choose a reference point $x_0$ on the manifold and frame the analysis in the corresponding fiber.
To do so, each observation is brought to the reference via parallel transport. In that fiber, owing to the Hilbertian structure, a notion of empirical mean naturally follows.
The paper's contribution on the statistical side is rather standard, tapping into the classical CLTs and concentration of measure theory. A major contribution is understanding what happens when geodesics are non-unique. To me, the most important result is the bound of Theorem 3 showing the impact of a transport rule.

**Compliance With Llm Reviewing Policy:**

Affirmed.

**Final Justification:**

I appreciate how the discussion took place and I am now more convinced that I had understood the paper correctly.

**Key Questions For Authors:**

1) Under your assumptions, why can't you compute the Fréchet mean $ \hat\mu_n$ and the use $s( \hat\mu_n)$ as the statistic of interest? Why isn't it the right object to look at and how does your proposal relate to this natural approach?   It is quite clear to me (even though not being sharp enough in the field) that the two proposals will be different; explaining how so seems natural. This might make a further case for the "simple linear" approach that you use.

2) The choice of $x_0$ seems to be very important. In the checklist, you acknowledge this fact, suggesting to take the data-dependent Fréchet barycenter. This would induce additional fluctuations, which do not seem to be addressed in the paper. This is important as the convergence rate of Fréchet barycenters can be slower than $n^{-1/2}$ in some cases. Can you develop on that?

3) The example of Wasserstein barycenters is promising and strengthens your paper. Still, the theory for that application is particularly technical and I fear that the paragraph might be misleading as written. Can you prove that your results hold in very regular settings? As currently written, I fear that the text is not careful enough.

**Limitations:**

yes

**Strengths And Weaknesses:**

The paper is sound. The theory is well developed and the assumptions make sense even to the non-specialist I am.

The presentation is overall very good. Still, the paper does not seem to be written for the audience of ICML. In the second paragraph, many potential applications are listed without any reference. In Section 6, the applications are not sufficiently developed for a layman to get a good grasp of why the work matters. The writing is also "schizophrenic" (if you let me say that). On the one hand, the reader is spoon-fed with basic concepts. On the other hand, around lines 134 of page 3, a lot of jargon and notation are dropped. I fear that it is difficult to find readers that know enough of both Riemannian geometry and ML applications to appreciate your paper.  Still about the presentation, Appendix H seems to be extremely important, yet it is not referred to in the main text. If you need support from the appendix, referring to it consistently is necessary.

It is very difficult for me to judge on the significance. I have the impression that a presentation with compelling and developed examples would help. Currently, it is a nice but niche paper. The scope could be strengthened to other statistics than the mean if it can be used in applications.


The paper is clearly original.

---

> ### Author Rebuttal · Authors · 2026-03-29
>
> We thank the reviewer for the careful reading, positive assessment of soundness, and constructive suggestions. We are particularly encouraged that Theorem 3 and the role of transport rules are viewed as central contributions.
>
> **Positioning and audience:**
>
> The paper aims to bridge two communities: geometric statistics (vector bundles, transport) and learning theory (non-asymptotic concentration). The perceived “duality” in exposition reflects this intent: basic geometric notions are introduced for accessibility, while later sections develop the full formalism. To improve readability, we will streamline the presentation by (i) reducing introductory material, (ii) smoothing the transition to formal notation in Section 3, and (iii) consistently referencing key appendices (including Appendix H) at the relevant points in the main text.
>
> **On applications and significance:**
>
> We agree that more developed examples will strengthen the impact. The current Section 6 already contains three representative cases (manifold regression, graph aggregation, Wasserstein settings). Still, we will expand one end-to-end example (manifold regression) in the main text, with explicit constants and a clear pipeline. The key message we will emphasize is that the decomposition
> $\|\bar{Y}_n - m^\star\|$
>
> $\le O(n^{-1/2}) + \Delta_{\mathrm{hol}}$
> yields a new form of uncertainty quantification where curvature induces a non-vanishing error floor. Classical Euclidean or intrinsic manifold analyses do not capture this phenomenon and is directly relevant to modern geometric ML pipelines.
>
> **Why not use $s(\hat\mu_n)$ (Fréchet mean)?**
>
> This is an important point. The estimator $s(\hat\mu_n)$ is fundamentally different from our transported mean. First, $s(\hat\mu_n)$ evaluates the section at an estimated base point, whereas our estimator aggregates observations after alignment in a common fiber. These two operations do not commute. In general,
> $s(\hat\mu_n)$  $\neq \frac{1}{n}\sum_i P_{X_i \to x_0}s(X_i)$,
> and the difference is controlled by the curvature and variability of $X_i$. Thus, the two estimators target different population quantities, even in the large-sample limit: $s(\mu)$ versus $\mathbb{E}[P_{X \to x_0}s(X)]$. This distinction is also reflected in Appendix H, which introduces an intrinsic bundle-mean formulation closely related to the transported estimator.
>
> More importantly, $s(\hat\mu_n)$ inherits the statistical properties of the Fréchet mean, whose convergence can be slower than $n^{-1/2}$ (and may be unstable near cut loci). In contrast, our approach reduces the problem to a Hilbert space, yielding dimension-free $n^{-1/2}$ rates plus an explicit geometric term. Thus, the “linear” approach is not only computationally simpler but admits clean non-asymptotic guarantees with explicit geometric terms.
>
> We will add a paragraph clarifying this distinction and providing a quantitative comparison.
>
> **Choice of $x_0$:**
>
> The analysis allows $x_0$ to be either fixed or data-dependent. When $x_0 = \hat\mu_n$, an additional fluctuation term appears. A standard argument (stability of parallel transport + Lipschitz continuity of $s$) yields
> $\|\bar{Y}_n(x_0) - \bar{Y}_n(\hat\mu_n)\|$
> $\le C\,\mathrm{dist}(x_0,\hat\mu_n)$,
>
> so the additional error scales with the convergence rate of $\hat\mu_n$. This can be slower than $n^{-1/2}$ in general, the reviewer noted ewer. Our framework separates this effect cleanly: one can choose between (i) fixed $x_0$ with pure $n^{-1/2}$ statistical rates, or (ii) adaptive $x_0$ with an additional geometric fluctuation term. We will include this discussion explicitly.
>
> **Wasserstein barycenters:**
>
> We agree that this example requires careful phrasing. Our intent is not to claim full generality, but to highlight that the same reduction principle applies when tangent objects (e.g., velocity fields) are compared across base measures. In the revision, we will restrict the discussion to standard settings (e.g., smooth densities, convex domains) where tangent representations form a Hilbert space and transport is well-defined, and we will clearly state the assumptions under which our results apply.
>
> **Scope beyond the mean:**
>
> While the current paper focuses on the mean as the fundamental primitive, the reduction to a fixed Hilbert space extends to other statistics (e.g., covariance, regression operators). We will clarify this perspective to indicate how the framework generalizes.
>
> **Summary:**
>
> We appreciate the reviewer’s positive assessment. We will strengthen the presentation and examples, clarify the relation to Fréchet means and data-dependent reference points, and make the scope of applications more explicit. The core results are complete and, we believe, provide a useful bridge between geometric statistics and learning theory.

---

> > ### Author Rebuttal · Reviewer_bmyb · 2026-04-03
> >
> > All my points, which were anyways mere extension suggestions, were addressed.
> > Thank you and best wishes with that paper.
> >
> > As a side note, I just came across the paper:
> > Wasserstein Parallel Transport for Predicting the Dynamics of Statistical Systems. I am none of the authors but guess that this paper and the references therein might strengthen your foreseen application.

---

> > > ### Author Response · Authors · 2026-04-04
> > >
> > > Thank you very much for your thoughtful follow-up and for taking the time to review our responses carefully. We truly appreciate your positive assessment and are grateful that you found our clarifications satisfactory.
> > >
> > > Many thanks as well for pointing us to Wasserstein Parallel Transport for Predicting the Dynamics of Statistical Systems. This is a very relevant and insightful connection, particularly in the context of modeling dynamics and structured evolution. We will study this work and the associated references carefully, and we believe it could indeed help strengthen and broaden the application perspective of our framework.
> > >
> > > We sincerely appreciate your engagement, encouragement, and helpful suggestions. Thank you again, and we hope you find the final version of the paper even stronger.
> > >
> > > Best regards,
> > > Authors.

---

### Official Review · Reviewer_FisM · 2026-03-18

**Soundness:** 3
**Presentation:** 1
**Significance:** 2
**Originality:** 2
**Overall Recommendation:** 3
**Confidence:** 3

**Summary:**

This paper considers bundle-valued maps (sections) of a collection of i.i.d. random variables on a Riemannian manifold and presents a concentration result by parallel transporting these maps to a reference fiber and averaging the maps. Standard large deviation bounds are applied in the reference frame when unique geodesics exist, along which transportation is performed. When geodesics are not unique the bound is corrected to account for the mismatch between the selected geodesics. This framework is suggested for the analysis of ML problems. The bounds are explicitly calculated for the case of a sphere as manifold. Some other examples are discussed.

**Compliance With Llm Reviewing Policy:**

Affirmed.

**Key Questions For Authors:**

Can the authors describe the application of their result in a concrete example? For example, I am familiar with the use of concentration results to obtain uniform convergence bounds in learning, such a Radamacher complexity or Gaussian width. Can they derive a concrete, tighter bound based on their framework and incorporating a manifold structure?

Can the authors clarify how they exactly think about Theorem 3? My impression is that they want to introduce bias (holonomy) bound as some sort of fundamental limit for learning. This idea is reinforced by Theorem 7, but if this approach is used merely for analysis, then the inconsistency simply shows multiple choices of “gauges” which probe the problem at hand at different precisions (or a different bias-variance trade-off). Why is this issue fundamental for learning?

I suspect that there is a mistake in the description of Theorem 7. One will not violate the assumption by choosing larger kappa but this will increase the lower bound to infinity. Can you clarify?

Can the authors clarify the two examples of manifold regression and graphs?
a)	For manifold regression, I guess that the estimate is an aggregate of all samples on the manifold and hence the residuals are not independent. How can the theorem be used in this case?
b)	For graph embeddings in C.5.1, why are the embedding “i.i.d.-like” and if this is a sparse graph, why should one be interested in concentration results? Typically, concentration is used for dense graphs with some sort of mean field approximation. Is this how the authors think about this case?

**Limitations:**

The main limitation for me is the presentation and the organization of the paper that does not allow me to understand its actual contribution. I suggest the authors focus on a concrete ML example, and redo a sort of analysis that has been already done on a trivial bundle with standard concentration results. Showing an improvement in this case can be a great advantage.

**Strengths And Weaknesses:**

Strength: The paper discusses a critical problem of incorporating structure in the analysis of ML methods, which often includes application of large deviation bounds. When the structure is presented as a manifold structure, standard tools such as the law of large numbers and CLT become inapplicable. The paper addresses this issue.

Weakness:  It is difficult for me to summarize this work, the organization of the paper is difficult for me to follow, and I am not sure what exactly the authors would like to achieve with this paper. The paper discusses many different topics, but all at a surface level. There is no coherent case study in ML that shows the real strength of the suggested framework.  This is particularly important because the main results are not difficult to prove and mainly intend to suggest a methodology for the analysis. There are a lot of critical details, such as the description of the example cases left to the appendix, but even there the discussion is unclear and the results are not conclusive. For example, the case of manifold regression appears in C5.2 with some ambiguous description and again in F1 with a bit more clarification.

---

> ### Author Rebuttal · Authors · 2026-03-29
>
> We thank the reviewer for the detailed and thoughtful feedback. The paper concerns a single estimand—the transported mean of a bundle-valued statistic—with manifold regression (tangent residuals) as the clearest motivating example. We agree that the current presentation did not sufficiently emphasize this central example and will revise accordingly.
>
> At a high level, we study non-asymptotic guarantees when observations lie in different fibers and must be aligned before averaging. The contribution is not a new concentration inequality but an explicit decomposition into a standard $n^{-1/2}$ stochastic term and a geometry-dependent transport term, revealing an additional effect absent in trivial-bundle analyses.
>
> The key result is: $\|\bar{Y}_n - m^\star\|$
>
> $\le O(n^{-1/2}) + \Delta_{\mathrm{hol}}$, which separates sampling error from a curvature-induced term absent in Euclidean or trivial-bundle settings.
>
> **Concrete ML example:**
> In manifold regression with residuals $r_i = \log_{X_i}\hat f(X_i)$, the standard pipeline (map $\to$ transport $\to$ average) is typically analyzed via local Euclidean approximations, yielding $O(n^{-1/2})$ error. Our result shows
> $ \|\bar{Y}_n - m^\star\|$
>
> $\lesssim n^{-1/2} + \Delta_{\mathrm{hol}},$
> revealing an additional term not captured by classical analyses. This implies that beyond a threshold, increasing sample size alone does not reduce error unless geometric spread is controlled (e.g., via localization or transport choice). Concretely, this predicts an observable error floor in tangent-space averaging pipelines when data are spread across regions with non-negligible curvature. We will present this as a unified case study (currently split across Appendix C/F).
>
> **Relation to classical analysis**
> When the bundle is trivial, $\Delta_{\mathrm{hol}}=0$ and we recover standard concentration. Thus, the contribution is not a new inequality but an explicit decomposition that reveals an additional geometry-dependent term absent in classical settings.
>
> **Uniform convergence**
> For a hypothesis class $\mathcal{F}$,
>
> $\sup_{f \in \mathcal{F}}$ $\|\bar{Y}_n(f) - m^\star(f)\|$
>
> $\le$ $C\big(\mathfrak{R}_n(\mathcal{F})$
>
> $+ \Delta_{\mathrm{hol}}\big)$,
>
> which extends standard Rademacher-type bounds by incorporating a geometry-dependent term.
>
> **Interpretation of Theorem 3:**
> The holonomy term is not a matter of gauge choice. Even with a fixed transport rule, non-uniqueness of geodesics induces intrinsic path dependence (curvature), so $\Delta_{\mathrm{hol}}$ reflects a geometric effect rather than representation.
>
> **Theorem 7 clarification:**
> Thanks for raising this issue. As written, Theorem 7 can be misleading if $\kappa$ is interpreted as an arbitrary admissible upper bound on curvature. In that interpretation, enlarging $\kappa$ would not violate the assumption but would artificially inflate the bound. The intended interpretation is that $\kappa$ represents a fixed local curvature scale of the manifold/connection over the support of the data (e.g., a minimal or intrinsic bound such as the supremum of sectional curvature on the relevant region), rather than a freely adjustable slack parameter. With this interpretation, $\kappa$ is determined by the geometry of the problem and is not under the analyst’s control, so the bound remains meaningful. We will revise the statement of Theorem 7 to make this explicit and avoid ambiguity.
>
> **Manifold regression independence:**
> The results apply directly when $f$ is fixed (or via sample splitting). In that case, $r_i$ is a function of $X_i \sim \mu$, so the transported variables remain i.i.d.
>
> **Graph example:**
> We focus on stochastic node/message sampling regimes, where features are treated as conditionally independent. In this setting, the geometric term corresponds to discrete holonomy (cycle inconsistency).
>
> **Quantitative regime:**
> If $\mathrm{supp}(\mu)\subset B(x_0,\rho)$ with curvature $\kappa$,
> $\Delta_{\mathrm{hol}} $
> $\lesssim \kappa \rho^2$,
> $\quad$
> $\|\bar{Y}_n - m^\star\| $
> $\lesssim n^{-1/2} + \kappa \rho^2$,
>
> so for $n \gtrsim (\kappa \rho^2)^{-2}$, error becomes curvature-dominated.
>
> **Presentation:**
> We will move the manifold regression example into the main text, unify Appendix C/F, and add explicit cross-references; these are organizational clarifications only.
>
> **Summary:**
> The contribution is an explicit decomposition that isolates a geometry-dependent term in transport-based averaging. This clarifies when classical concentration analyses are insufficient and provides a framework for analyzing such pipelines.
>
> **What this achieves in practice:**
> Classical analyses effectively assume $\Delta_{\mathrm{hol}}=0$ and predict $O(n^{-1/2})$ decay. Our results show that error may not decay below a scale governed by $\Delta_{\mathrm{hol}}$, so increasing sample size alone cannot reduce error beyond a curvature-dependent level. This identifies when classical guarantees become overly optimistic and quantifies the resulting gap.

---

> > ### Author Rebuttal · Reviewer_FisM · 2026-04-05
> >
> > Many thanks for your reply. The issue about Theorem 7 is resolved, but my main concern still remains. The key issue for me is that I am not able to interpret the main result. If I am correct, the main result shows that there is a point m^* in the reference bundle for which the inequality holds INDEPENDENT of the choice of the transport maps. So, if we agree on a particular transportation, even if it is not unique, the constant on the right hand side will disappear, but m^* is now changed to a value depending on the transport maps. So, for me it is not clear why it is important for m^* to be independent of the transport maps. The only case that I can imagine is that m^* should correspond to an actual quantity that one might be interested in estimating, but it is not straightforward to imagine a concrete example.
> >
> > It is a bit difficult to imagine that this bound is fundamental. For example, suppose that the bundle is actually embedded in a trivial bundle of a finite dimension (i.e. some finite dimensional Euclidian space). Then, one might ignore the bundle structure and perform conventional estimation procedures. The estimate might not lie on the bundle, of course, but as the number of data points grows, standard consistency results will show that the estimate converges to the true value under mild conditions and there is no error floor.
> >
> > Mathematically speaking, one needs to also show a lower bound on the error, i.e. show that no matter how estimation is performed the error cannot be reduced. I do not think that it is impossible to show this (and actually this result is not necessary for me to improve my score), but it requires a concrete definition of the estimation problem, which is nontrivial to me.

---

> > > ### Author Response · Authors · 2026-04-05
> > >
> > > **Reviewer FisM follow-up:**
> > >
> > > We thank the reviewer for the careful follow-up and for clarifying the remaining concern, and our responses are given below.
> > >
> > > **(1) Why define $m^\star$ independently of transport?**
> > >
> > > You are correct that for any fixed transport rule $P$, one can define a corresponding mean $m^{(P)} = \mathbb{E}[P_{X\to x_0}s(X)]$, and the stochastic term concentrates around this quantity. In that sense, if one commits to a single transport rule, the ambiguity term disappears from the deviation bound.
> > >
> > > However, the purpose of introducing a transport-independent reference $m^\star$ is to separate what is intrinsic to the statistical problem from what depends on the chosen alignment. Different admissible transport rules correspond to equally valid but incompatible identifications between fibers, and the resulting means $m^{(P)}$ can differ by $O(\Delta_{\mathrm{hol}})$ even in the infinite-sample limit.
> > >
> > > Thus, without fixing a canonical identification, there is no unique “ground-truth mean” in a single vector space. The quantity $m^\star$ should be interpreted as a transport-invariant target, representing the equivalence class of transported means across admissible identifications.
> > >
> > > **(2) Concrete interpretation in ML pipelines**
> > >
> > >   A concrete instance arises in tangent-space aggregation pipelines (e.g., manifold regression or geometric deep learning), where observations $s(X_i) \in T_{X_i}M$ must be aligned before averaging. Different alignment conventions (e.g., different geodesics, discrete transports, or learned alignments) lead to different aggregate outputs. Such alignment-dependent aggregation is standard in modern geometric learning pipelines, e.g., in gauge-equivariant neural networks where features are transported between local frames (see Section 3 in Ref.  R1), and in Riemannian representation learning where tangent vectors are compared via log/exp maps and parallel transport (see Section 2.2 in Ref. R2).
> > >
> > > Each choice yields a valid estimator converging to $m^{(P)}$, but the discrepancy across choices is observable in practice and does not vanish with more data. The role of $m^\star$ is to formalize what is common across such procedures, while $\Delta_{\mathrm{hol}}$ quantifies the irreducible disagreement induced by geometry.
> > >
> > > **R1:** Cohen, T. et al. Gauge Equivariant Convolutional Networks and the Icosahedral CNN. ICML 2019.
> > >
> > > **R2:** Nickel, M., & Kiela, D. Learning Continuous Hierarchies in the Lorentz Model of Hyperbolic Geometry. ICML 2018.
> > >
> > >
> > > **(3) Relation to extrinsic (trivial-bundle) estimation**
> > >
> > > We agree that embedding the bundle into a Euclidean space and performing standard estimation can remove the apparent error floor. However, this corresponds to a different statistical problem: extrinsic averaging estimates an ambient-space quantity rather than an intrinsically defined bundle-valued mean.
> > >
> > > In particular, such estimators do not respect the fiber structure or the geometry of the data (e.g., tangent constraints, equivariance, or coordinate consistency), and the resulting estimates may not correspond to any meaningful section or intrinsic quantity. In contrast, many pipelines of interest (e.g., tangent residual aggregation or gauge-equivariant models) require outputs that are consistent across fibers and transformations, making alignment unavoidable.
> > >
> > > Our results apply precisely in this intrinsic setting, where one seeks a coordinate-consistent aggregation of fiber-valued data. In this regime, the ambiguity is not due to suboptimal estimation, but arises from the geometry of how different fibers are identified.
> > >
> > > **(4) On “fundamental” vs. modeling scope**
> > >
> > > Indeed, the current paper does not establish a full minimax lower bound over all possible estimators. Rather, the results characterize the behavior of transport-based estimators, which cover standard alignment-based pipelines in geometric statistics and learning.
> > >
> > > Within this class, the decomposition shows that the estimation error separates into a stochastic component and a geometry-dependent term. In particular, when $\Delta_{\mathrm{hol}}$ is non-negligible, increasing $n$ reduces only the stochastic part, leading to a regime where geometric effects dominate.
> > >
> > > We therefore interpret the result as identifying a structural limitation of alignment-based estimation procedures, rather than a universal lower bound over all conceivable estimators. We will clarify this scope in the final version to avoid any sense of overstating generality while preserving the main insight.
> > >
> > > **(5) Summary**
> > >
> > > The key point is not that a single estimator has an unavoidable bias, but that in bundle-valued problems, there is no canonical global mean without choosing an alignment. Different valid choices lead to limits that differ by a curvature-controlled amount, and this discrepancy persists even with infinite data. The contribution of the paper is to make this phenomenon explicit and quantifiable within a standard concentration framework.

---

### Decision · Program_Chairs · 2026-04-30

**Decision:**

Accept (regular)

**Comment:**

Reviews for this paper are generally positive. A summary is below.

Strengths
- Relevant and non-trivial problem in geometric machine learning.
- Careful and rigorous mathematical presentation.
- Obtained results should be of broad use, given how ubiquitous non-geometric tail bounds of the kind the authors derive are.

Weaknesses
- Limited applications presented in the work

On balance of these, I think the third strength is key - learning theory in geometric settings is much less developed precisely because the fundamental technical work to adapt tools such as various concentration inequalities to the setting has not yet been done. I therefore think the contribution is strong enough to outweigh the key weaknesses, and warrants acceptance, and recommend as such.